# OpenworldAUC: Towards Unified Evaluation and Optimization for Open-world Prompt Tuning

Cong Hua [1 2]   Qianqian Xu [1]   Zhiyong Yang [2]   Zitai Wang [1]   Shilong Bao [2]   Qingming Huang [2 3 1]

## Abstract

Prompt tuning adapts Vision-Language Models like CLIP to open-world tasks with minimal training costs. In this direction, one typical paradigm evaluates model performance **separately** on known classes (*i.e.*, base domain) and unseen classes (*i.e.*, new domain). However, real-world scenarios require models to handle inputs **without prior domain knowledge**. This practical challenge has spurred the development of **open-world prompt tuning**, which demands a unified evaluation of two stages: 1) detecting whether an input belongs to the base or new domain (**P1**), and 2) classifying the sample into its correct class (**P2**). What's more, as domain distributions are generally unknown, a proper metric should be insensitive to varying base/new sample ratios (**P3**). However, we find that current metrics, including HM, overall accuracy, and AUROC, fail to satisfy these three properties simultaneously. To bridge this gap, we propose OpenworldAUC, a unified metric that jointly assesses detection and classification through pairwise instance comparisons. To optimize OpenworldAUC effectively, we introduce **Gated Mixture-of-Prompts (GMoP)**, which employs domain-specific prompts and a gating mechanism to dynamically balance detection and classification. Theoretical guarantees ensure generalization of GMoP under practical conditions. Experiments on 15 benchmarks in open-world scenarios show GMoP achieves SOTA performance on OpenworldAUC and other metrics. We release the code at https://github.com/huacong/OpenworldAUC

*Table 1.* An overview of the existing metrics for OPT. ✗ means OverallAcc evaluates detection implicitly.

| Metrics | (P1)First-stage detection | (P2)Second-stage classification | (P3)Domain distribution insensitivity |
|---|---|---|---|
| HM | ✗ | ✓ | ✓ |
| OverallAcc | ✗ | ✓ | ✗ |
| AUROC | ✓ | ✗ | ✓ |
| OpenworldAUC | ✓ | ✓ | ✓ |

## 1. Introduction

Recent powerful Vision-Language Foundation Models (VLMs), such as CLIP (Radford et al., 2021), use **natural language** to align visual concepts, enabling them to infer new samples. To better utilize such abilities with minimal computational cost, Prompt Tuning (PT) has gained significant attention (Lester et al., 2021; Zhou et al., 2022b). PT allows the model to adapt to open-world scenarios with only a small set of learnable parameters for textual or visual prompts.

In this direction, a common setting is base-to-new (Zhou et al., 2022b; Yao et al., 2024; Zhou et al., 2022a; Khattak et al., 2023b; Zhang et al., 2024), where the model is trained on a set of known classes (*i.e.*, base domain) and evaluated **separately** on the base domain and the unseen classes (*i.e.*, new domain). This task implicitly assumes that the input samples are pre-labeled as belonging to either the base or new domain. However, this assumption may not hold in practice. To address this limitation, (Zhou et al., 2024) adopt a divide-and-conquer strategy, inducing a more practical setting named Open-world Prompt Tuning (OPT). In this setting, the pipeline has two stages: the first stage performs **base-to-new detection** to determine whether an input belongs to the base or new domain, and the second stage **classifies the sample into its correct class**.

In this complicated setting, how to evaluate model performance becomes challenging. We argue that an appropriate metric should comprehensively evaluate **(P1)** first-stage detection, **(P2)** second-stage classification, and also **(P3)** be insensitive to the domain distribution, *i.e.* the base/new ratio. Unfortunately, as summarized in Tab. 1, existing metrics do not satisfy the aforementioned properties simultaneously, making them inconsistent with the actual model performance. Concretely, the harmonic mean (HM) of BaseAcc and NewAcc, a common metric for the base-to-new task,

---

[1]Key Lab of Intelligent Info. Processing, Inst. of Comput. Tech., CAS, Beijing, China [2]School of Comput. Sci. and Tech., UCAS, Beijing, China [3]Key Lab of Big Data Mining and Knowl. Manag., UCAS, Beijing, China. Correspondence to: Qianqian Xu <xuqianqian@ict.ac.cn>, Qingming Huang <qmhuang@ucas.ac.cn>.

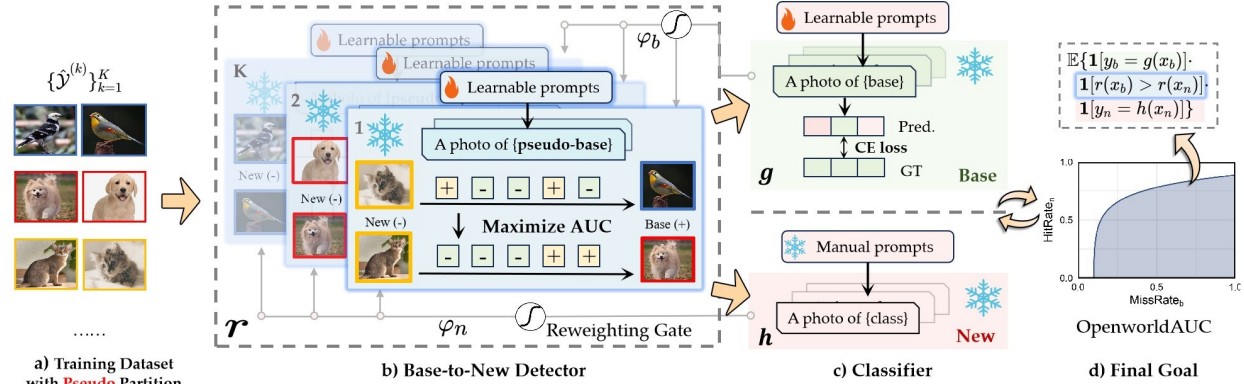

*Figure 1.* OpenworldAUC and its optimization framework. **a)** We first perform pseudo partition on the training dataset to simulate new domain. b) Based on this partition, we calculate the AUROC-like ranking loss to optimize the detector. **c)** Then, to optimize base-domain classifier, the CE loss is calculated on the original training set. Herein, a gating mechanism selects the correctly-classified samples to calculate the aforementioned ranking loss. **c)** For new domain classifier, we adopt a fixed hand-crafted prompt to avoid the overfitting on the base domain. **d)** Overall, each prompt fulfills its specific responsibility to jointly maximize the OpenworldAUC metric.

only evaluates second-stage performance but **ignores detection performance (P1)**. The empirical results in (Zhou et al., 2024) and our theoretical analysis in Sec. 3.1 both illustrate that models with the same HM can have a significant performance gap in the OPT setting. To alleviate this issue, (Zhou et al., 2024) turn to calculate the accuracy defined on the entire class space, *i.e.*, the mixture of base and new classes, denoted by OverallAcc. Compared with HM, this metric can assess detection to some extent by penalizing misaligned OOD/ID scores between domains. However, as shown in Sec. 3.2, we find that OverallAcc **is highly sensitive to the domain distribution, violating (P3)**. To fix this issue, one naive option is AUROC (Fawcett, 2006; Yang & Ying, 2023; Bao et al., 2025), a popular metric for OOD detection (Hendrycks & Gimpel, 2017) due to its insensitivity to label distribution. However, AUROC **ignores the classification performance, overlooking (P2)**. Hence, a natural question arises: *Can we find a single metric that simultaneously satisfies three properties?*

To answer this question, in Sec. 4, we establish a novel metric named OpenworldAUC, which evaluates base-to-new detection, base classification, and new classification in a unified manner. Specifically, this measure has a pairwise formulation, where each pair consists of a base instance and a new instance. For each pair, OpenworldAUC measures the joint probability that 1) the new instance has a higher OOD score than the base instance **(P1)**, and 2) the two instances are correctly classified in their respective domains **(P2)**. This ranking-based approach can naturally evaluate model performance under any domain distribution **(P3)**. Further analysis theoretically shows that OpenworldAUC overcomes the limitations of the aforementioned metrics.

In view of this, we design a learning framework to maximize OpenworldAUC effectively in Sec. 5.1. To achieve this

goal, the key challenge is that a single prompt is insufficient to balance all three components in the OpenworldAUC objective since each component involves different inputs and sub-objectives. To address this issue, we propose a novel Gated Mixture-of-Prompts (GMoP) approach. On the one hand, each prompt targets a specific component. On the other hand, the gating mechanism selects the correctly-classified instances to optimize the detection performance via an AUROC-like ranking loss. In this way, we achieve a divide-and-conquer optimization, which is exactly consistent with the OPT pipeline. Besides, we adopt the pseudo base-to-new partitions of the training set as proposed in (Zhou et al., 2024) to calculate the ranking loss.

Last but not least, in Sec. 5.2, we explore the generalization guarantee for our proposed learning framework, which, to the best of our knowledge, is rarely discussed in the PT community. Our theoretical results suggest that a training set with a moderate volume and a proper partition number can guarantee a satisfactory generalization performance. Finally, in Sec. 6, extensive empirical results on fifteen benchmarks in the open-world scenario demonstrate that our proposed method outperforms the state-of-the-art methods on both OpenworldAUC and other metrics.

Overall, the contributions of this paper are as follows:

- **Novel metric.** A novel metric named OpenworldAUC is proposed for the OPT task, which embraces base-to-new detection, base classification, and new classification in a **unified** way and is also **insensitive** towards varying domain distributions.

- **Learning framework.** A learning framework named GMoP is proposed with theoretical guarantee, where multiple prompts jointly optimize OpenworldAUC via a divide-and-conquer strategy.

- **Empirical studies.** Comprehensive empirical results on fifteen benchmarks in the open-world scenario speak to the efficacy of our proposed method on both OpenworldAUC and other metrics.

## 2. Problem Formulation

This paper focuses on prompt tuning for open-world classification. Let $x \in \mathcal{X}$ be the input sample and $y \in \mathcal{Y} := \mathcal{Y}_b \cup \mathcal{Y}_n$ be the corresponding label, where $\mathcal{X}$ is the input space; $\mathcal{Y}_b$ and $\mathcal{Y}_n$ are the label spaces of base classes and new classes, respectively. In open-world learning, the model is first trained on the base domain $\mathcal{D}_b := \mathcal{X} \times \mathcal{Y}_b$ and further evaluated on the overall domain $\mathcal{D} := \mathcal{D}_b \cup \mathcal{D}_n$, where $\mathcal{D}_n := \mathcal{X} \times \mathcal{Y}_n$ denotes the new domain.

To achieve this goal, Open-world Prompt Tuning (OPT) (Zhou et al., 2024) fine-tunes the prompt of a foundation model, such as CLIP (Radford et al., 2021), on the base dataset $\mathcal{S}_b = \{(x_b^{(i)}, y_b^{(i)})\}_{i=1}^{N_b}$ sampled from $\mathcal{D}_b$. Specifically, let $f_v$ be the visual feature of the input image $x$ obtained by the frozen image encoder. For text representation, class $i$ is first embedded through a prompt template $P([CLASS_i]; \theta)$ parameterized by the leanable token $\theta$. Then, the text feature $f_{t_i}$ is obtained by feeding the $i$-th class prompt into the frozen text encoder. Finally, the posterior probability $\mathbb{P}(y = i | x)$ is assumed be proportional to the cosine similarity between $f_{t_i}$ and $f_v$.

During testing, OPT adopts a divide-and-conquer strategy by introducing an additional evaluation of detection, compared with the common PT task (Zhou et al., 2022b). In a nutshell, this task involves **two stages**: base-to-new detection and domain-specific classification. **In the first stage**, a base-to-new detector $r : \mathcal{X} \to [0, 1]$ identifies whether the input sample belongs to the base or the new domain according to a threshold $t$, where a larger $r(x)$ indicates a higher probability of the sample belonging to the base domain. In other words, this stage actually performs OOD detection with an OOD score defined by $1 - r(x)$. **In the second stage**, $x$ is further sent to its corresponding classifier, denoted by $g$ and $h$ for the base and new domain.

## 3. Existing Metrics and Their Limitations

In this section, we discuss the limitations of the existing metrics, including HM, OverallAcc and AUROC from three key perspectives: comprehensively evaluating first-stage detection **(P1)**, second-stage classification **(P2)**, and being insensitive towards domain distribution **(P3)**.

### 3.1. Harmonic Mean

HM is a popular metric in prompt tuning for the base-to-new task (Xian et al., 2017; Zhou et al., 2022b;a; Yao et al.,

2024). Let BaseAcc and NewAcc represent the classification accuracy defined on the base domain $\mathcal{D}_b$ and the new domain $\mathcal{D}_n$, respectively. Suppose $s_b \in \mathbb{R}^{C_b}$ and $s_n \in \mathbb{R}^{C_n}$ are the logit scores of the base and new domain produced by $g$ and $h$, where $C_b$ and $C_n$ are the numbers of classes in each domain. Then, BaseAcc and NewAcc are defined as follows:

$$\text{BaseAcc} := \frac{1}{N_b} \sum_{(x_b, y_b) \in \mathcal{S}_b} \mathbf{1}[y_b = \arg \max_{y' \in \mathcal{Y}_b} s_b],$$

$$\text{NewAcc} := \frac{1}{N_n} \sum_{(x_n, y_n) \in \mathcal{S}_n} \mathbf{1}[y_n = \arg \max_{y' \in \mathcal{Y}_n} s_n],$$

where $N_b$ and $N_n$ are the number of samples in $\mathcal{S}_b$ and $\mathcal{S}_n$, respectively and $\mathbf{1}[\cdot]$ is the indicator function. Then, HM is defined by the harmonic mean of BaseAcc and NewAcc:

$$\text{HM} := \frac{2 \times \text{BaseAcc} \times \text{NewAcc}}{\text{BaseAcc} + \text{NewAcc}}.$$

From this formulation, one can easily find that HM only evaluates the domain-specific classification performance in OPT. This limitation becomes evident in the following proposition, whose proof can be found in App.A.1.

**Proposition 3.1.** *Given a dataset $\mathcal{S}$ sampled from the overall domain $\mathcal{D}$, for any $(g, h)$, one can always find a worse-performing $(\tilde{g}, \tilde{h})$ in OPT that satisfies* $\text{HM}(g, h) = \text{HM}(\tilde{g}, \tilde{h})$.

**Remark.** The proof follows this intuition: consider one base sample $x_b$ and suppose that $(g, h)$ predicts it correctly while satisfying $\max_{y' \in \mathcal{Y}_b} s_b - \max_{y' \in \mathcal{Y}_n} s_n > 0$. We can construct $(\tilde{g}, \tilde{h})$ such that $\tilde{g}$ remains correct on $x_b$ but satisfies $\max_{y' \in \mathcal{Y}_b} \tilde{s}_b - \max_{y' \in \mathcal{Y}_n} \tilde{s}_n < 0$. In this case, $(g, h)$ and $(\tilde{g}, \tilde{h})$ share the same HM score, yet $(\tilde{g}, \tilde{h})$ misidentifies more base sample as new than $(g, h)$ in OPT, thereby matching the empirical observations in (Zhou et al., 2024).

From Prop.3.1, we can conclude that HM ignores the detection performance **(P1)** and thus is not a proper choice for the OPT task.

### 3.2. Overall Accuracy

To alleviate this issue of ignoring **(P1)**, the recent study DeCoOp (Zhou et al., 2024) turns to evaluate the accuracy performance directly defined on the overall domain, denoted by OverallAcc. This metric is defined as:

$$\text{OverallAcc} := \frac{1}{N} \sum_{(x, y) \in \mathcal{S}} \mathbf{1}[y = \arg \max_{y' \in \mathcal{Y}} s_b || s_n],$$

where $N$ is the number of samples in $\mathcal{S}$ and $||$ denotes the concatenation operation.

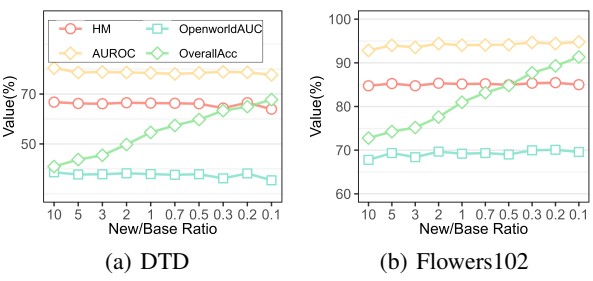

(a) DTD       (b) Flowers102

*Figure 2.* The sensitive analysis of existing metric *w.r.t.* new/base ratio. OverallAcc is sensitive to the domain distribution, while other metrics remains stable with varying ratios of new samples.

Compared with HM, OverallAcc considers detection to some extent. To be concrete, the prediction could be regarded as correct only if the input has been identified in the correct domain. In other words, there exists an implicit detector $r(\boldsymbol{x}) \propto \max_{y' \in \mathcal{Y}_b} \boldsymbol{s}_b - \max_{y' \in \mathcal{Y}_n} \boldsymbol{s}_n$. However, **this metric becomes sensitive to the domain distribution, violating (P3)**. To illustrate this issue, we can rewrite the definition of OverallAcc as follows:

$$\text{OverallAcc} = \mathbb{P}_b \cdot \text{TPR}_b + \mathbb{P}_n \cdot \text{TPR}_n,$$

where $\text{TPR}_b$ and $\text{TPR}_n$ are True Positive Rate of base and new domain, respectively, and $\mathbb{P}_b = N_b/N$ and $\mathbb{P}_n = N_n/N$. From this formulation, one can find that OverallAcc can be dominated by the performance in the domain with more samples. As shown in Fig.2, OverallAcc increases significantly as the number of base domain samples grows, while the other metrics remain stable. Since the domain distribution is generally unknown, a distribution-sensitive metric is improper for OPT.

### 3.3. AUROC

Another potential choice to evaluate the detection performance is AUROC (Fawcett, 2006), which is a popular metric for OOD detection (Hendrycks & Gimpel, 2017). To adapt this metric to the OPT task, we should consider the base-to-new detection as a binary classification problem. Specifically, we first assign all base/new classes to one base/new super-class, respectively. Then, AUROC can be defined as the area under the curve of the True Positive Rate ($\text{TPR}_{ns}$) and the False Positive Rate ($\text{FPR}_{ns}$) of the new super-class, with a pairwise reformulation:

$$\text{AUROC} := \int_{t=0}^{t=1} \text{TPR}_{ns}(t) \cdot d\,\text{FPR}_{ns}(t)$$
$$= \mathbb{E}_{\mathcal{D}} \left[ \mathbf{1}[r(\boldsymbol{x}_b) > r(\boldsymbol{x}_n)] \right].$$

Benefiting from the pairwise ranking formulation, AUROC measures detection performance **under any domain distribution** (Yang & Ying, 2023). As shown in Fig.2, AUROC is **insensitive** towards the base/new ratios. However, it is

clear that AUROC **ignores the classification performance in the second stage, overlooking (P2)**.

## 4. OpenworldAUC: A Novel OPT Metric

In this section, we first explore a naive solution by aggregating multiple metrics. Then, we develop a single metric named OpenworldAUC that can **satisfy the aforementioned three properties simultaneously**.

### 4.1. A Naive Multiple Metrics Aggregation

Since HM and AUROC are complementary and both insensitive to domain distributions, one potential solution is to aggregate them via linear combination. However, the following proposition shows that this naive solution can induce another issue, with proof in App.A.2.

**Proposition 4.1.** *Given a dataset $\mathcal{S}$ sampled from the overall domain $\mathcal{D}$, for any $(r, g, h)$ under mild assumption $\text{AUROC}(r) < 1$, one can always find $(\tilde{r}, \tilde{g}, \tilde{h})$ performs worse in OPT that satisfies:*

$$\text{Lin}(\text{AUROC}(r), \text{BaseAcc}(g), \text{NewAcc}(h))$$
$$= \text{Lin}(\text{AUROC}(\tilde{r}), \text{BaseAcc}(\tilde{g}), \text{NewAcc}(\tilde{h}))$$

*where $\text{Lin}(\cdot)$ denotes the linear combination function.*

**Remark.** The proof comes from the following intuition: given two base samples $\boldsymbol{x}_b^1$ and $\boldsymbol{x}_b^2$, suppose that $r$ and $g$ are both wrong on $\boldsymbol{x}_b^1$ but both correct on $\boldsymbol{x}_b^2$. We can construct $(\tilde{r}, \tilde{g})$ such that $\tilde{r}$ misidentifies $\boldsymbol{x}_b^1$ but $\tilde{g}$ makes a correct prediction on $\boldsymbol{x}_b^1$, and vice versa for $\boldsymbol{x}_b^2$. In this case, the linear combination remains the same, but $(\tilde{r}, \tilde{g})$ performs inferior to $(r, g)$ since more samples are misclassified.

### 4.2. OpenworldAUC

To tailor this problem, we have to **seek a single measure, in which the prediction is regarded to be correct only when the model performs well in both stages**. Following this idea, we first define the Miss Rate of the base class ($\text{MissRate}_b$) and the Hit Rate of the new class ($\text{HitRate}_n$) in this pipeline:

$$\text{MissRate}_b := \mathbb{E}_{\mathcal{D}_b}[\mathbf{1}[r(\boldsymbol{x}_b) \leq t] + \mathbf{1}[r(\boldsymbol{x}_b) > t, y_b \neq g(\boldsymbol{x}_b)]]$$
$$\text{HitRate}_n := \mathbb{E}_{\mathcal{D}_n}[\mathbf{1}[r(\boldsymbol{x}_n) \leq t, y_n = h(\boldsymbol{x}_n)]]$$

However, the two rates rely on the threshold $t$, which is hard to decide without prior knowledge of the new domain. Inspired by AUROC, we calculate the area under the $\text{MissRate}_b$-$\text{HitRate}_n$ curve, denoted as OpenworldAUC:

$$\text{OpenworldAUC} := \int_{t=0}^{t=1} \text{HitRate}_n(t) \cdot d\text{MissRate}_b(t)$$

At first glance, the integral formulation is hard to compute. Fortunately, the following proposition can alleviate this issue, with proof in App.A.3.

**Proposition 4.2.** *Given a pair $(\boldsymbol{x}_b, y_b)$ and $(\boldsymbol{x}_n, y_n)$ sampled from $\mathcal{D}$,* OpenworldAUC *equals to the joint possibility that 1) $r$ ranks $\boldsymbol{x}_b$ higher than $\boldsymbol{x}_n$, and 2) $g$ and $h$ make correct predictions on $\boldsymbol{x}_b$ and $\boldsymbol{x}_n$, respectively.*

$$\mathbb{E}_{\mathcal{D}}\Big[\underbrace{\mathbf{1}[y_b = g(\boldsymbol{x}_b)]}_{\text{Base (P2)}} \cdot \underbrace{\mathbf{1}[r(\boldsymbol{x}_b) > r(\boldsymbol{x}_n)]}_{\text{Base-to-New (P1)}} \cdot \underbrace{\mathbf{1}[y_n = h(\boldsymbol{x}_n)]}_{\text{New (P2)}}\Big]$$

Benefiting from this concise formulation, one can easily find that OpenworldAUC satisfies **(P1)** and **(P2)**. Furthermore, as shown in Fig.2, OpenworldAUC is insensitive to the domain distribution due to the ranking formulation, fulfilling **(P3)**. To further highlight its advantages over existing metrics, we provide the following proposition, with proof in App.A.4.

**Proposition 4.3.** *For any $(r, g, h)$, we have:*

- *Assume that* OpenworldAUC *equals $u$, a lower bound proportional to $u$ is available for* HitRate$_n$ *for **any domain distributions**.*

- *Constructing $(\tilde{r}, \tilde{g}, \tilde{h})$ as outlined in Prop.4.1 yields* OpenworldAUC$(\tilde{r}, \tilde{g}, \tilde{h}) <$ OpenworldAUC$(r, g, h)$.

These properties show that: 1) Compared with OverallAcc, optimizing OpenworldAUC can guarantee the model performance on the new domain **under arbitrary base/new ratio**. 2) OpenworldAUC is more consistent with the model performance than the naive aggregation of multiple metrics. These advantages encourage us to design an effective method to optimize OpenworldAUC in the next section.

# 5. The Optimization of OpenworldAUC

Next, we explore how to optimize OpenworldAUC effectively, whose overall pipeline is shown in Fig.1.

## 5.1. Gated Mixture-of-Prompts for OpenworldAUC Optimization

We start by reformulating OpenworldAUC maximization to an equivalent minimization problem via the following proposition, with proof provided in App.A.5.

**Proposition 5.1.** *Maximizing the* OpenworldAUC *metric can be achieved by solving the following problem:*

$$\min_{r, g, h} \mathbb{E}_{\mathcal{D}}\left[\mathbf{1}[y_b \neq g(\boldsymbol{x}_b)] + \mathbf{1}[y_n \neq h(\boldsymbol{x}_n)]\right.$$
$$\left. + \mathbf{1}[y_b = g(\boldsymbol{x}_b)] \cdot \mathbf{1}[r(\boldsymbol{x}_b) \leq r(\boldsymbol{x}_n)] \cdot \mathbf{1}[y_n = h(\boldsymbol{x}_n)]\right]$$

Since the distribution $\mathcal{D}$ is unknown, we resort to minimizing the empirical risk to tune the prompt $P(\cdot; \boldsymbol{\theta})$, which is

parameterized by learnable tokens $\boldsymbol{\theta}$:

$$\min_{\boldsymbol{\theta}} \hat{\mathbb{E}}_{\mathcal{S}_b}\left[\mathbf{1}[y_b \neq g(\boldsymbol{x}_b; \boldsymbol{\theta})]\right] + \hat{\mathbb{E}}_{\mathcal{S}_n}\left[\mathbf{1}[y_n \neq h(\boldsymbol{x}_n; \boldsymbol{\theta})]\right]$$
$$+ \hat{\mathbb{E}}_{\mathcal{S}}[v_b \cdot \ell_{0,1}(r(\boldsymbol{x}_b; \boldsymbol{\theta}) - r(\boldsymbol{x}_n; \boldsymbol{\theta})) \cdot v_n]$$

where $v_b := \mathbf{1}[y_b = g(\boldsymbol{x}_b; \boldsymbol{\theta})]$ and $v_n := \mathbf{1}[y_n = h(\boldsymbol{x}_n; \boldsymbol{\theta})]$; $\ell_{0,1}$ denotes the 0-1 loss. From this formulation, we identify two key challenges: **(C1)** Training a single prompt to balance the three components is inherently conflicting, as each component requires distinct inputs and targets on different sub-objectives. This can lead to **mutual interference** during optimization. **(C2)** New class samples are unavailable during training. Next, we elaborate on how to address these challenges.

**Gated Mixture-of-Prompts.** To address **(C1)**, we propose a novel Mixture-of-Prompts approach with three specialized prompts $P(\cdot; \boldsymbol{\theta}_r), P(\cdot; \boldsymbol{\theta}_g), P(\cdot; \boldsymbol{\theta}_h)$ targeting for $r, g, h$, respectively. Among these, optimizing $\boldsymbol{\theta}_r$ is more challenging because the third term in the objective involves a non-differentiable sample-selection operation $v_b, v_n$. To tackle this issue, we use smooth and differentiable functions $\varphi_b := \sigma(\boldsymbol{s}_b^y(\boldsymbol{x}_b; \boldsymbol{\theta}_g))$ and $\varphi_n := \sigma(\boldsymbol{s}_n^y(\boldsymbol{x}_n; \boldsymbol{\theta}_h))$ to approximate the selection process, where $\sigma(\cdot)$ denotes the sigmoid function; $\boldsymbol{s}_b^y$ and $\boldsymbol{s}_n^y$ are the ground-truth channels of $\boldsymbol{s}_b$ and $\boldsymbol{s}_n$, respectively. Here, the sigmoid function acts as a **gate** since correctly classified samples tend to have higher confidence scores in the ground-truth channel. This gate adaptively assigns the weights to each base-new pair based on the outputs of $g, h$ to optimize $r$, encouraging correctly classified samples to be ranked correctly. Following this approach, $\boldsymbol{\theta}_g, \boldsymbol{\theta}_h, \boldsymbol{\theta}_r$ are merged through the gate mechanism into a mixture $\boldsymbol{\theta}_{\text{mix}}$:

$$\ell(\boldsymbol{x}_b, \boldsymbol{x}_n; \boldsymbol{\theta}_{\text{mix}}) = \varphi_b \cdot \ell_{sq}(r(\boldsymbol{x}_b; \boldsymbol{\theta}_r) - r(\boldsymbol{x}_n; \boldsymbol{\theta}_r)) \cdot \varphi_n$$

where $\boldsymbol{\theta}_{\text{mix}} = (\boldsymbol{\theta}_r, \boldsymbol{\theta}_g, \boldsymbol{\theta}_h)$ represents all learnable tokens. Besides, following the surrogate loss framework (Gao & Zhou, 2015; Liu et al., 2020; Yang et al., 2022; 2023; Bao et al., 2022), we replace the non-differentiable 0-1 loss with a convex upper bound $\ell(t)$, such as $\ell_{sq}(t) = (1-t)^2$ for scores in $[0, 1]$. This smooth approximation enables gradient-based optimization while preserving ranking semantics. Hence, our goal is then to solve the following problem:

$$(OP_0) \min_{\boldsymbol{\theta}_{\text{mix}}} \hat{\mathcal{R}}(r, g, h) = \hat{\mathbb{E}}_{\mathcal{S}_b}\left[\ell_{ce}(\boldsymbol{s}_b(\boldsymbol{x}_b; \boldsymbol{\theta}_g), y_b)\right]$$
$$+ \hat{\mathbb{E}}_{\mathcal{S}_n}\left[\ell_{ce}(\boldsymbol{s}_n(\boldsymbol{x}_n; \boldsymbol{\theta}_h), y_n)\right] + \hat{\mathbb{E}}_{\mathcal{S}}\left[\ell(\boldsymbol{x}_b, \boldsymbol{x}_n; \boldsymbol{\theta}_{\text{mix}})\right]$$

where we use the CE loss $\ell_{ce}$ for $(g, h)$ and replace the non-differentiable $\ell_{0,1}$ with the squared loss $\ell_{sq}(t) = (1-t)^2$.

**Pseudo Base-to-New Partition.** To handle **(C2)**, we follow the prior arts (Zhou et al., 2024) to perform pseudo base-to-new partition $\hat{\mathcal{Y}} = (\hat{\mathcal{Y}}_b, \hat{\mathcal{Y}}_n)$ over the known class

$\mathcal{Y}_b$. Specifically, $\hat{\mathcal{Y}}_b$ and $\hat{\mathcal{Y}}_n$ are pseudo base and new domain which satisfy $\hat{\mathcal{Y}}_b \cup \hat{\mathcal{Y}}_n = \mathcal{Y}_b$ and $\hat{\mathcal{Y}}_b \cap \hat{\mathcal{Y}}_n = \emptyset$. To reduce the partition bias, we estimate the expectation of the partition distribution by generating $K$ pseudo partitions $\hat{\mathcal{Y}}^{(k)}, k \in [1, K]$, satisfying $\bigcup_{k=1}^{K} \hat{\mathcal{Y}}_b^{(k)} = \mathcal{Y}_b$. Accordingly, we assign each partition with a sub-detector $r^{(k)}(\cdot; \boldsymbol{\theta}_{r,k})$ to avoid conflicts. During testing, the highest score across all sub-detectors will be used as the final score $r(\boldsymbol{x}; \boldsymbol{\theta}_r)$.

**Zero-shot New Domain Classifier.** Last but not least, we observe that training a new domain classifier on the base domain can lead to overfitting, reducing new class performance (Zhou et al., 2022b). To balance accuracy and efficiency, we use a zero-shot CLIP model with a fixed hand-crafted prompt $\boldsymbol{\theta}_h^*$ for the new domain classifier $h$.

Overall, we have the following final objective, where $\hat{\mathcal{S}}^{(k)}$ denotes the dataset partition defined on $\mathcal{X} \times \hat{\mathcal{Y}}^{(k)}$ and $\boldsymbol{\theta}'_{\text{mix}} = (\boldsymbol{\theta}_{r,1}, \cdots, \boldsymbol{\theta}_{r,k}, \boldsymbol{\theta}_g, \boldsymbol{\theta}_h^*)$

$$(OP_{\text{fin}}) \min_{\boldsymbol{\theta}'_{\text{mix}}} \hat{\mathcal{R}}'(r, g, h) = \hat{\mathbb{E}}_{\mathcal{S}_b} \left[ \ell_{ce}(\boldsymbol{s}_b(\boldsymbol{x}_b; \boldsymbol{\theta}_g), y_b) \right]$$
$$+ \frac{1}{K} \cdot \hat{\mathbb{E}}_{\hat{\mathcal{S}}^{(k)}} [\ell(\boldsymbol{x}_b, \boldsymbol{x}_n; \boldsymbol{\theta}'_{\text{mix}})]$$

### 5.2. Generalization Bound

In this part, we explore how well the proposed method can generalize to test data theoretically. Specifically, let $\mathcal{R}(r, g, h)$ be the population risk of $\hat{\mathcal{R}}(r, g, h)$ in $(OP_0)$. Our task is to bound the difference between $\mathcal{R}(r, g, h)$ and $\hat{\mathcal{R}}'(r, g, h)$. To this end, we present the following informal theorem, whose detailed proof is presented in App.B.

**Theorem 5.2.** *Let $\mathcal{E}$ and $\mathcal{E}'$ be the distributions over the expectation of $\mathcal{Y}$ and $\hat{\mathcal{Y}}$, respectively. Given the function space of $\mathcal{H}_r$ and $\mathcal{H}_g$, the following inequality holds for all $r \in \mathcal{H}_r, \boldsymbol{s}_b \in \mathcal{H}_g$ with high probability:*

$$\mathcal{R}(r, g, h) \lesssim \underbrace{\hat{\mathcal{R}}'(r, g, h)}_{(1)} + \underbrace{\mathbb{E}_{\mathcal{D}_n} \left[ \ell_{ce}(\boldsymbol{s}_n(\boldsymbol{x}_n), y_n) \right]}_{(2)}$$
$$+ \underbrace{\mathbb{C} \cdot O(K^{-1/2} + N_b^{-1/2}) + \frac{||\mathcal{E} - \mathcal{E}'||_\infty}{(C_b + C_n)!}}_{(3)},$$

*where $\mathbb{C}$ denotes the complexity of the hypothesis space; $\lesssim$ denotes the asymptotic notation that omits undominated terms.*

**Theoretical insights.** The bound in Thm.5.2 consists of the following terms: **(1) Empirical Error:** The empirical loss on the training set, optimized in $(OP_{\text{fin}})$. **(2) Error for New Classes:** The expected error of zero-shot classifier $h$ in the new domain, which can be reduced with carefully-designed prompts or more powerful foundation models. **(3) Stochastic Error** including three parts: First, the data estimation error from approximating the training distribution $\mathcal{D}$

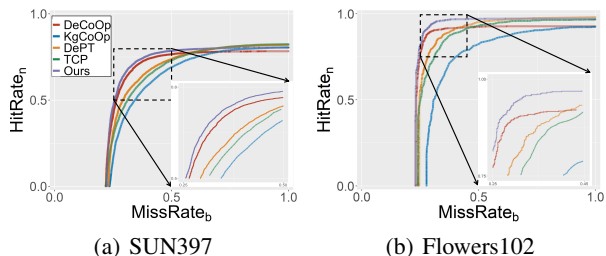

(a) SUN397      (b) Flowers102

*Figure 3.* The $\text{MissRate}_n$-$\text{HitRate}_b$ curve on SUN397 and Flowers102. Our method can outperform other competitors on the meaningful region with lower $\text{MissRate}_b$ and higher $\text{HitRate}_n$.

with the empirical average from $\mathcal{S}$, which can be minimized with a moderate $N_b$ due to the small parameter space of the prompt. Second, the error from finite pseudo base-to-new partitions during detector optimization, which is reduced by increasing partition number $K$. Third, the error from a potential shift from the pseudo new domain to the underlying real new domain.

## 6. Experiments

In this section, we describe details of the experiments and present our results. Due to space limitations, please refer to App.C and App.D for more results about experiments.

### 6.1. Experimental Setup

**Task and Dataset Description.** We evaluate **three** open-world tasks: **open-world recognition**, **open-world domain generalization** and **cross-dataset generalization**. For recognition, models are trained on base classes and tested on the mixture of base and new classes within each dataset, with test sets resampled at varying base/new ratios to study imbalance. To examine domain-level imbalance, we resample the test sets with varying base/new ratios, which is different from class-level imbalance in long-tail settings (Wang et al., 2024; Zhao et al., 2024a;b; Wang et al., 2023). For domain generalization, models are trained on base classes of ImageNet and are tested on the full classes of ImageNet variants with domain shifts. For the practical cross-dataset setting, the prompts are tunned on the base domain in ImageNet dataset and tested on other full datasets. Experiments use 11 image recognition benchmarks, 4 ImageNet variants, and 10 domain-imbalanced datasets (1:5 ratio) built via random sampling. Details are provided in App.C.2.

**Competitors.** To demonstrate the effectiveness of our proposed method, we compare our method with 10 competitive SOTA competitors: $a$) Baseline, Zero-shot CLIP (Radford et al., 2021); $b$) HM-oriented methods, including CoOp (Zhou et al., 2022b), Maple (Khattak et al., 2023a), KgCoOp (Yao et al., 2023), PromptSRC (Khattak et al., 2023b), DePT-Kg (Zhang et al., 2024) and TCP (Yao

*Table 2.* The OpenworldAUC empirical results on eleven benchmark for open world prompt tuning recognition.

| Method | IN | C101 | Pets | Cars | F102 | Food | AC | SUN | DTD | ES | UCF | Avg. |
|--------|-----|------|------|------|------|------|------|------|------|------|------|------|
| CLIP | 47.31 | 82.31 | 76.17 | 43.43 | 48.51 | 75.09 | 7.23 | 42.52 | 25.22 | 28.01 | 50.37 | 47.83 |
| CoOp | 48.93 | 83.29 | 80.71 | 35.38 | 59.65 | 67.27 | 5.60 | 48.03 | 25.48 | 41.96 | 43.95 | 47.59 |
| Maple | 51.89 | 87.15 | 85.59 | 49.99 | 65.44 | 76.83 | 9.58 | 52.84 | 36.22 | **56.55** | 58.72 | 57.35 |
| PromptSRC | 52.44 | 86.74 | 86.10 | 50.90 | 69.36 | 77.33 | 11.40 | 54.19 | 40.30 | 52.56 | 58.94 | 58.21 |
| LoCoOp | 45.12 | 86.59 | 86.62 | 46.52 | 61.17 | 73.09 | 8.67 | 50.55 | 32.26 | 41.35 | 46.90 | 52.62 |
| KgCoOp | 51.45 | 86.63 | 86.80 | 49.40 | 62.96 | 76.96 | 8.18 | 52.09 | 34.87 | 39.16 | 57.29 | 55.07 |
| DePT-Kg | 51.35 | 92.74 | 87.81 | 55.24 | 69.46 | 79.45 | **12.71** | 56.42 | 37.56 | 44.90 | 61.38 | 59.00 |
| Gallop | 49.01 | 87.51 | 86.94 | 50.69 | 65.69 | 73.60 | 11.38 | 50.62 | 40.22 | 51.38 | 58.91 | 56.90 |
| DeCoOp | 51.98 | 92.72 | 88.72 | 53.59 | 70.28 | 80.67 | 8.17 | 57.00 | 37.07 | 46.66 | 59.57 | 58.77 |
| TCP | 51.34 | 88.65 | 85.50 | 53.18 | 69.20 | 77.27 | 10.72 | 54.86 | 37.92 | 55.89 | **63.39** | 58.90 |
| **Ours** | **52.64** | **92.81** | **89.77** | **55.31** | **72.79** | **81.25** | 11.42 | **58.54** | **40.37** | 53.09 | 62.39 | **60.94** |

*Table 3.* The OpenworldAUC empirical results on ten imbalance benchmark with different domain distributions. Fwd means base domain number is $5\times$ larger than new domain, and Bwd means the opposite.

| Method | DTD | | Food101 | | Flowers102 | | OxfordPets | | SUN397 | | Avg. |
|--------|------|------|------|------|------|------|------|------|------|------|------|
| | Fwd | Bwd | Fwd | Bwd | Fwd | Bwd | Fwd | Bwd | Fwd | Bwd | |
| CLIP | 25.71 | 25.74 | 76.11 | 75.76 | 48.97 | 45.37 | 82.82 | 80.30 | 42.39 | 42.68 | 54.59 |
| CoOp | 24.98 | 25.35 | 67.22 | 67.14 | 52.64 | 51.03 | 77.53 | 77.41 | 48.25 | 48.16 | 53.97 |
| MaPLe | 33.86 | 34.15 | 76.85 | 76.85 | 65.98 | 64.95 | 85.67 | 85.88 | 53.36 | 52.94 | 63.05 |
| PromptSRC | 40.51 | 39.40 | 77.51 | 77.25 | 70.08 | 69.44 | 85.47 | 86.10 | 54.43 | 54.16 | 65.44 |
| LoCoOp | 32.34 | 31.62 | 73.23 | 72.96 | 61.25 | 60.67 | 86.29 | 86.32 | 51.09 | 50.68 | 60.65 |
| KgCoOp | 34.75 | 34.95 | 77.03 | 76.80 | 63.28 | 62.05 | 86.74 | 87.01 | 52.47 | 52.05 | 62.71 |
| DePT-Kg | 37.32 | 37.06 | 79.71 | 79.28 | 69.24 | 69.41 | 87.52 | 87.85 | 56.76 | 56.45 | 66.06 |
| Gallop | **41.25** | 40.91 | 73.61 | 73.88 | 66.22 | 65.86 | 86.65 | 86.30 | 51.14 | 50.89 | 63.67 |
| DeCoOp | 36.74 | 36.61 | 80.65 | **81.11** | 70.09 | 69.89 | 89.76 | 89.10 | 57.37 | 57.37 | 66.87 |
| TCP | 38.18 | 37.75 | 77.42 | 77.18 | 70.08 | 69.34 | 85.63 | 85.58 | 55.09 | 55.04 | 65.13 |
| **Ours** | 41.13 | **40.97** | **81.65** | 81.09 | **73.50** | **73.10** | **90.36** | **89.79** | **58.59** | **58.26** | **68.84** |

et al., 2024); $c$) OOD-oriented methods, including Lo-CoOp (Miyai et al., 2023) and Gallop (Lafon et al., 2024); $d$) OverallAcc-oriented algorithms, DeCoOp (Zhou et al., 2024). Detailed descriptions of each competitor are provided in App.C.3

**Implementation details.** We adopt 16-shot prompt tuning setting following previous studies (Zhou et al., 2022b; Yao et al., 2024; Khattak et al., 2023b). To ensure fairness, we reimplement all competitor models on our device using their open-source code. Results are reported as the average over 5 runs with different random seeds $1, 2, 3, 4, 5$. The default backbone used is the publicly available ViT-B/16 of CLIP. For all competitors except DeCoOp, we use the maximum probability among the base domain as the base-to-new detection score $r$. The computation of the OpenworldAUC metric and the implementation of the corresponding AUC loss functions are based on the open-source library open-source library **XCurve**. More implementation details are deferred in the App.C.4 and App.D.3.

### 6.2. Performance Comparison

**Openworld Recognition.** Tab. 2, Tab. 8, Tab. 9 compare the overall performance across eleven benchmarks

*Table 4.* The OpenworldAUC empirical results on ImageNet variants benchmark for open world domain generalization task.

| Method | Source | Target | | | |
|--------|--------|--------|------|------|------|
| | ImageNet | V2 | S | R | A |
| CLIP | 47.31 | 39.49 | 22.57 | 57.04 | 23.83 |
| CoOp | 48.93 | 39.88 | 21.69 | 56.58 | 24.22 |
| Maple | 51.89 | 42.44 | 23.79 | 59.71 | 25.10 |
| PromptSRC | 52.44 | 42.55 | 24.50 | 60.65 | 25.08 |
| LoCoOp | 45.12 | 38.45 | 20.88 | 57.31 | 22.98 |
| KgCoOp | 51.45 | 42.14 | 24.24 | 59.75 | 25.79 |
| DePT-Kg | 51.35 | 42.45 | 23.97 | 60.60 | 25.84 |
| Gallop | 49.01 | 39.86 | 21.57 | 56.78 | 22.28 |
| DeCoOp | 51.98 | 43.01 | 24.65 | 61.01 | 25.31 |
| TCP | 51.34 | 41.66 | 23.15 | 58.46 | 24.62 |
| **Ours** | **52.64** | **43.98** | **25.64** | **62.67** | **26.49** |

for the open-world recognition task. Our method outperforms existing approaches on most datasets, achieving an average 2% improvement in OpenworldAUC with a smaller parameter cost, shown in Fig.5. This highlights the effectiveness of the proposed optimization methods. To visualize the OpenworldAUC metric, we also plot the MissRate$_b$-HitRate$_n$ curve shown in Fig.3, which demon-

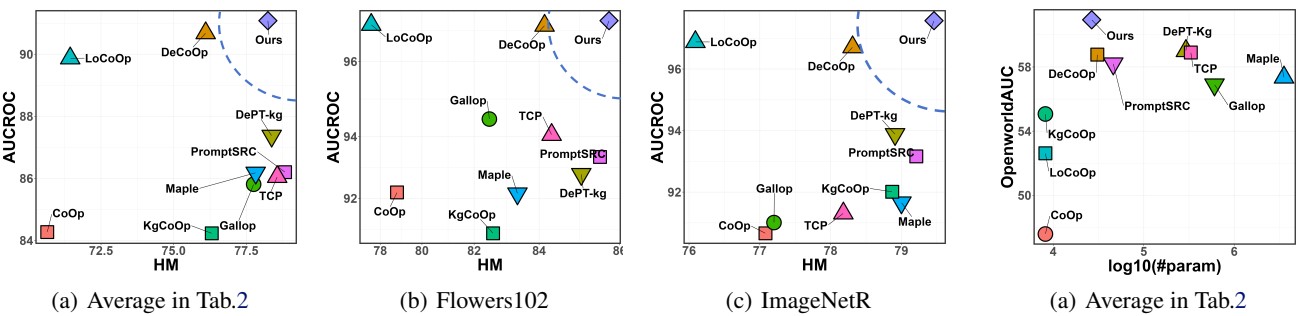

| (a) Average in Tab.2 | (b) Flowers102 | (c) ImageNetR | (a) Average in Tab.2 |

*Figure 4.* Trade-off between the first-stage AUROC and the second-stage HM metrics. Our approach, located in the **upper right corner**, shows a better **trade-off** between first-stage detection and second-stage classification performance.

*Figure 5.* Tradeoff between performance and prompt complexity across different methods.

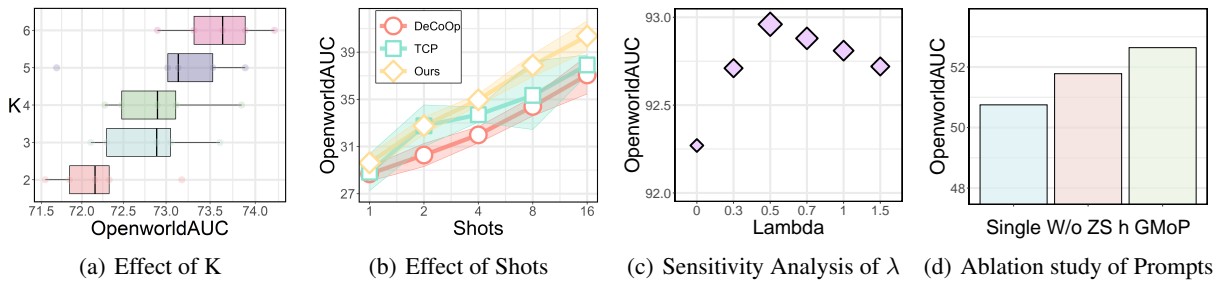

| (a) Effect of K | (b) Effect of Shots | (c) Sensitivity Analysis of $\lambda$ | (d) Ablation study of Prompts |

*Figure 6.* Sensitivity Analysis and Ablation Study. (a), (b), (c) and (d) are performed on Flowers102, DTD, Caltech101 and ImageNet, respectively. More results are provided in App.D.7, App.D.8, App.D.10 and App.D.11.

strates that our optimization method can significantly outperform other competitors on the region with $\mathsf{MissRate}_b \leq \alpha$ and $\mathsf{HitRate}_n \geq \beta$. This area is particularly meaningful, as it represents the task's key concern: a skillful model should balance a high HitRate with a low MissRate, as emphasized in recent AUC literature (Yang et al., 2023).

**Tradeoff between the** AUROC **and the** HM **metrics.** As shown in Fig.4 (a)-(b) and Fig.7, methods targeting the HM metric often lack base-to-new detection abilities **(P1)**, while OOD-oriented methods typically compromise classification performance **(P2)**. These limitations align with the constraints of the metrics, as discussed in Sec.3. Compared with these methods, our OpenworldAUC-oriented optimization method can achieve a better trade-off between the AUROC and HM, which further validates the comprehensiveness of the OpenworldAUC metric and the efficiency of our optimization method. This is consistent with **(P1)** and **(P2)**.

**Imbalance Setting for Recognition.** Tab. 3 compares the overall OpenworldAUC performance across five datasets with different domain imbalance ratios. OpenworldAUC as a comprehensive and distribution-insensitive metric shown in Sec.D.2, can effectively capture the performance of the model in the imbalance setting. Our method consistently outperforms existing approaches across all datasets, achieving an average 2% improvement. This demonstrates that our OpenworldAUC-oriented optimization method is also

robust towards the varying domain distribution, consistent with **(P3)**.

**Openworld Domain Generalization** Tab. 4 compares the robustness performance of methods trained on ImageNet to various domain-shift datasets. We observe that our method consistently improves against all competitors in terms of OpenworldAUC. Fig.4 (c), Tab. 10 and Fig.8 further demonstrate that our method can also achieve better performance trade-offs for stage-wise metrics on this more generalizable task. Our fine-tuning strategy can further enhance the transferability of the model to the domain-shift scenarios, which is crucial for real-world applications.

**Cross-Dataset Generalization** Tab. 13 compares the performance of our method with existing competitors on the cross-dataset generalization task. The comprehensive results of this cross-domain evaluation test the model's ability to handle both base and new categories across diverse visual domains, validating the robustness of our method.

### 6.3. Ablation Study

**Sensitivity Analysis of** $K$**.** As shown in Fig.6 (a) and Fig.10, the performance monotonically increases with more base-to-new partitions ($K$) on Flowers102 and SUN397, consistent with Thm. 5.2. In practice, we set $K = 3$ for efficiency-performance balance.

**Effect of Different Shots.** As shown in Fig.6 (b) and Fig.11, our method surpasses representative competitors across all shots (1/2/4/8/16) on DTD and OxfordPets, further demonstrating its effectiveness. Additionally, OpenworldAUC improves as $N$ increases, aligning with Thm. 5.2.

**Sensitivity Analysis of $\lambda$.** In the practical objective ($OP_{fin}$), the second term serves as an AUC-style ranking loss. However, recent studies (Yuan et al., 2022; Yang et al., 2023; Han et al., 2024) show that optimizing AUC loss from scratch can harm feature representations. The cross-entropy regularization term $\lambda \cdot \ell_{ce}$ improves ranking performance while mitigating this issue, as illustrated in Fig.6(c) and Fig.12. Optimal results are usually achieved when $\lambda \in [1/2, 1]$.

**Effect of Mixture-of-Prompts** Fig.6 (d) and Fig.13 show the effectiveness of the mixture-of-prompt strategy. This validates the challenge of optimizing OpenworldAUC within a single prompt as discussed in Sec.5.1.

**Ablation Study of the Gating Mechanism.** We evaluate the gating mechanism in Tab. 17. Using a fixed 0-1 mask ("Ours 0-1 Gate") slightly outperforms removing the gate ("Ours w/o Gate") but underperforms the adaptive sigmoid gate, confirming the effectiveness of both sparse sample selection and gate approximation.

# 7. Related Work

In this section, we briefly review the closely related work on few-shot prompt tuning and Open-world Learning.

## 7.1. Few-shot Prompt Tuning for VLM

CoOp (Zhou et al., 2022b) first introduced prompt tuning to adapt VLMs for downstream tasks, significantly improving classification on base classes but showing poor generalization to new classes. To address this, early methods proposed more flexible prompt structures. For example, CoCoOp (Zhou et al., 2022a) generates input-conditional tokens using lightweight networks, enabling dynamic prompts per instance. MaPLe (Khattak et al., 2023a) adopts a multimodal joint learning strategy (Jiang et al., 2025; Xu et al., 2025). Another line of work (Yao et al., 2023; Bulat & Tzimiropoulos, 2023; Zhu et al., 2023; Khattak et al., 2023b; Roy & Etemad, 2024; Zhang et al., 2024; Hua et al., 2024; Yao et al., 2024) enhances generalization by introducing task-agnostic objectives and preserving the zero-shot knowledge of VLMs. KgCoOp (Yao et al., 2023) and TCP (Yao et al., 2024) reduce the gap between learned and hand-crafted prompt features. Recent advances (Li et al., 2024; Mistretta et al., 2024; Wu et al., 2024; Zhang et al., 2023; Khattak et al., 2024) further boost performance by leveraging external knowledge. For instance,(Li et al., 2024; Mistretta et al., 2024; Wu et al., 2024) apply knowledge

distillation from larger models, while (Zhang et al., 2023; Khattak et al., 2024) use LLMs to generate prompts with richer semantics.

While most methods focus on improving in-domain classification, they overlook inter-domain misclassification risks—such as confusing base and new domain samples—which is critical for open-world scenarios. This has led to growing interest in prompt tuning for OOD detection. LoCoOp (Miyai et al., 2023) treats non-semantic regions (e.g., background) as new-domain signals and adds scoring constraints to the CoOp objective. Follow-up methods introduce hierarchical (Lafon et al., 2024) and negative prompts (Zeng et al., 2024; Nie et al., 2024) to better balance accuracy and robustness.

Recently, DeCoOp (Zhou et al., 2024) proposed the first open-world prompt tuning framework that jointly addresses OOD detection and balanced classification across domains. It combines an OOD detector for base-to-new separation with domain-specific classifiers to improve within-domain discriminability.

## 7.2. Opens-Set Recognition

A closely related topic to this paper is Open-Set Recognition (OSR). OSR is a challenging and practical setting, where the model is required to detect open-set samples which do not come from the training and also correctly classify the close-set samples (Yue et al., 2021; Feng et al., 2022). Compared to the open-world learning discussed in this paper, OSR does not require precise classification of open-set samples. In this direction, a variety of novel metrics (Yue et al., 2021; Kong & Ramanan, 2025; Scherreik & Rigling, 2016; Wang et al., 2022) have been proposed to evaluate OSR models, focusing on close-set classification and open-set OOD detection. However, these metrics have limitations when applied to our open-world setting because they cannot assess the classification performance of open-set samples.

# 8. Conclusion

This paper explores Open-world Prompt Tuning for Vision-Language Models, involving base-to-new detection and domain-specific classification. We argue that ideal metrics should consistently evaluate both stages and remain robust to domain distributions, yet existing metrics fall short. To bridge this gap, we propose OpenworldAUC, a unified metric that simultaneously assesses detection and classification through pairwise sample comparison, thus being insensitive towards the varying domain distributions. In pursuit of this, our Gated Mixture-of-Prompts is proposed with a theoretical guarantee where each prompt fulfills its specific responsibility to jointly maximize OpenworldAUC. Extensive experiments speak to the effectiveness of our method.

## Impact Statement

We propose a general evaluation metric for openworld recognition to deal with the potential bias toward new domain samples. For fairness-sensitive real scenarios, it might be helpful to improve fairness for groups with fewer occurrences.

## Acknowledgements

This work was supported in part by the National Key R&D Program of China under Grant 2018AAA0102000, in part by National Natural Science Foundation of China: 62236008, 62441232, U21B2038, U23B2051 62122075, 62206264 and 92370102, in part by Youth Innovation Promotion Association CAS, in part by the Strategic Priority Research Program of the Chinese Academy of Sciences, Grant No.XDB0680201, in part by the Fundamental Research Funds for the Central Universities, in part by the China National Postdoctoral Program for Innovative Talents under Grant BX20240384, and in part by the Postdoctoral Fellowship Program of CPSF under Grant GZB20240729.

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

# Contents

# A. Proof for the Proposition

## A.1. Proof for Proposition 3.1.

**Restate of Proposition 3.1.** Given a dataset $\mathcal{S}$ sampled from the overall domain $\mathcal{D}$, for any $(g, h)$, one can always find a worse-performing $(\tilde{g}, \tilde{h})$ in OPT that satisfies $\mathsf{HM}(g, h) = \mathsf{HM}(\tilde{g}, \tilde{h})$.

*Proof.* Let $\boldsymbol{s}_b \in \mathbb{R}^{C_b}$ and $\boldsymbol{s}_n \in \mathbb{R}^{C_n}$ be the logits scores of the base and new domain produced by $g$ and $h$. Here, we only focus on differences in predictions between $(g, h)$ and $(\tilde{g}, \tilde{h})$. Given one base sample $(\boldsymbol{x}_b, y_b)$, $(g, h)$ and $(\tilde{g}, \tilde{h})$ make the following predictions:

- $y_b = g(\boldsymbol{x}_b)$, $\max_{y' \in \mathcal{Y}_b} \boldsymbol{s}_b(\boldsymbol{x}_b) - \max_{y' \in \mathcal{Y}_n} \boldsymbol{s}_n(\boldsymbol{x}_b) > 0$

- $y_b = \tilde{g}(\boldsymbol{x}_b)$, $\max_{y' \in \mathcal{Y}_b} \tilde{\boldsymbol{s}}_b(\boldsymbol{x}_b) - \max_{y' \in \mathcal{Y}_n} \tilde{\boldsymbol{s}}_n(\boldsymbol{x}_b) < 0$

In this case, although $\tilde{g}$ correctly classifies $\boldsymbol{x}_b$, the joint model in OPT $(\tilde{g}, \tilde{h})$ nevertheless categorizes $\boldsymbol{x}_b$ into the new domain. This occurs because the class associated with the highest prediction probability for $\boldsymbol{x}_b$ resides in the new domain. Therefore, $\mathsf{HM}(g, h) = \mathsf{HM}(\tilde{g}, \tilde{h})$ holds but $(\tilde{g}, \tilde{h})$ performs inferior to $(g, h)$ in OPT setting.

In OPT task, the prediction could be regarded as correct only if the input has been identified into the correct domain. In other words, there exists an implicit detector defined as:

$$r(\boldsymbol{x}) \propto \max_{y' \in \mathcal{Y}_b} \boldsymbol{s}_b(\boldsymbol{x}) - \max_{y' \in \mathcal{Y}_n} \boldsymbol{s}_n(\boldsymbol{x})$$

The OPT requires $r(\boldsymbol{x}_b) > 0$ for base samples and $r(\boldsymbol{x}_n) < 0$ for new samples. From the definition of HM, one can easily find that HM lacks evaluations of this detector.

$\square$

## A.2. Proof for Proposition 4.1.

**Restate of Proposition 4.1.** Given a dataset $\mathcal{S}$ sampled from the overall domain $\mathcal{D}$, for any $(r, g, h)$ under mild assumption $\mathsf{AUROC}(r) < 1$, one can always find $(\tilde{r}, \tilde{g}, \tilde{h})$ performs worse in OPT that satisfies:

$$\mathsf{Lin}(\mathsf{AUROC}(r), \mathsf{BaseAcc}(g), \mathsf{NewAcc}(h)) = \mathsf{Lin}(\mathsf{AUROC}(\tilde{r}), \mathsf{BaseAcc}(\tilde{g}), \mathsf{NewAcc}(\tilde{h}))$$

where $\mathsf{Lin}(\cdot)$ denotes the linear combination function.

*Proof.* Similarly, we only focus only on differences in predictions between $(r, g, h)$ and $(\tilde{r}, \tilde{g}, \tilde{h})$. Given two base-domain samples $(x_b^1, y_b^1)$, $(x_b^2, y_b^2)$ and two new-domain samples $(x_n^1, y_n^1)$, $(x_n^2, y_n^2)$, suppose that $(\tilde{r}, \tilde{g}, \tilde{h})$ and $(r, g, h)$ only make same predictions except the following cases:

- $g(x_b^1), \tilde{g}(x_b^1) = y_b^1$ and $g(x_b^2), \tilde{g}(x_b^2) \neq y_b^2$

- $h(x_n^1), \tilde{h}(x_n^1) \neq y_n^1$ and $h(x_n^2), \tilde{h}(x_n^2) = y_n^2$

- $r(x_b^1) > r(x_n^1) > r(x_b^2) > r(x_n^2)$

- $\tilde{r}(x_b^2) > \tilde{r}(x_n^2) > \tilde{r}(x_b^1) > \tilde{r}(x_n^1)$

- $r(x_b^1) = \tilde{r}(x_b^2), r(x_n^1) = \tilde{r}(x_n^2), r(x_b^2) = \tilde{r}(x_b^1), r(x_n^2) = \tilde{r}(x_n^1)$

From this, we observe that:

- $\mathsf{BaseAcc}(g) = \mathsf{BaseAcc}(\tilde{g})$ and $\mathsf{NewAcc}(h) = \mathsf{NewAcc}(\tilde{h})$ since $(\tilde{g}, \tilde{h})$ and $(g, h)$ share the same predictions.

- $\mathsf{AUROC}(r) = \mathsf{AUROC}(\tilde{r})$ since the ordering between base and new samples is also not changed.

Consequently, we have $\mathsf{Lin}(\mathsf{AUROC}(r), \mathsf{BaseAcc}(g), \mathsf{NewAcc}(h)) = \mathsf{Lin}(\mathsf{AUROC}(\tilde{r}), \mathsf{BaseAcc}(\tilde{g}), \mathsf{NewAcc}(\tilde{h}))$.

However, $(\tilde{r}, \tilde{g}, \tilde{h})$ performs worse than $(r, g, h)$ in the open-world prompt tuning task. Specifically, if we select a threshold $t$ such that $r(x_n^1) > t > r(x_b^2)$, then $(r, g, h)$ correctly predicts $x_b^1$ and $x_n^2$, while $(\tilde{r}, \tilde{g}, \tilde{h})$ misclassify all four samples. This shows that the linear aggregation of BaseAcc, NewAcc, and AUROC does not align with actual model performance. $\square$

### A.3. Proof for Proposition 4.2.

**Restate of Proposition 4.2.** Given a pair $(\boldsymbol{x}_b, y_b)$ and $(\boldsymbol{x}_n, y_n)$ sampled from $\mathcal{D}$, OpenworldAUC equals to the joint possibility that 1) $r$ ranks $\boldsymbol{x}_b$ higher than $\boldsymbol{x}_n$, and 2) $g$ and $h$ make correct predictions on $\boldsymbol{x}_b$ and $\boldsymbol{x}_n$, respectively.

$$\mathbb{E}_{\mathcal{D}}\Big[ \underbrace{\mathbf{1}[y_b = g(\boldsymbol{x}_b)]}_{\textbf{Base (P2)}} \cdot \underbrace{\mathbf{1}[r(\boldsymbol{x}_b) > r(\boldsymbol{x}_n)]}_{\textbf{Base-to-New (P1)}} \cdot \underbrace{\mathbf{1}[y_n = h(\boldsymbol{x}_n)]}_{\textbf{New (P2)}} \Big]$$

*Proof.* We first review the definition of $\mathsf{MissRate}_b$ and $\mathsf{HitRate}_n$ as:

$$\mathsf{MissRate}_b := \mathbb{E}_{(x_b, y_b) \sim \mathcal{D}_b} [\mathbf{1}[r(x_b) \le t] + \mathbf{1}[r(x_b) > t, y_b \ne g(x_b)]]$$

$$\mathsf{HitRate}_n := \mathbb{E}_{(x_n, y_n) \sim \mathcal{D}_n} [\mathbf{1}[r(x_n) \le t, y_n = h(x_n)]]$$

Specifically, $\mathsf{MissRate}_b$ measures the probability that $(h, r)$ misclassifies the base samples and $\mathsf{HitRate}_n$ is the probability that $(g, r)$ makes correct predictions on the new samples.

$$\mathsf{MissRate}_b = \mathbb{P}[r(x_b) \le t] + \mathbb{P}[r(x_b) > t, y_b \ne g(x_b)]$$

$$\mathsf{HitRate}_n = \mathbb{P}[r(x_n) \le t, y_n = h(x_n)]$$

Thus, we can rewrite the OpenworldAUC as:

$$\mathsf{OpenworldAUC} := \int_{t=0}^{t=1} \mathsf{HitRate}_n(t) \cdot d\mathsf{MissRate}_b(t)$$

$$= \int_{t=0}^{t=1} \mathbb{P}[r(x_n) \le t, y_n = h(x_n)] \cdot d\{\mathbb{P}[r(x_b) \le t] + \mathbb{P}[r(x_b) > t, y_b \ne g(x_b)]\}$$

$$= \underbrace{\int_{t=0}^{t=1} \mathbb{P}[r(x_n) \le t, y_n = h(x_n)] \cdot d\{\mathbb{P}[r(x_b) \le t]}_{(a)} + \underbrace{\int_{t=0}^{t=1} \mathbb{P}[r(x_n) \le t, y_n = h(x_n)] \cdot d\mathbb{P}[r(x_b) > t, y_b \ne g(x_b)]}_{(b)}$$

For $(a)$,

$$(a) = \int_{t=0}^{t=1} \mathbb{P}[r(x_n) \le t, y_n = h(x_n)] \cdot d\mathbb{P}[r(x_b) \le t]$$

$$= \mathbb{P}[y_n = h(x_n)] \cdot \int_{t=0}^{t=1} \mathbb{P}[r(x_n) \le t | y_n = h(x_n)] \cdot d\mathbb{P}[r(x_b) \le t]$$

$$= \mathbb{P}[y_n = h(x_n)] \cdot \mathbb{P}[r(x_b) > r(x_n) | y_n = h(x_n)]$$

$$= \mathbb{P}[r(x_b) > r(x_n), y_n = h(x_n)]$$

For $(b)$,

$$(b) = -\int_{t=1}^{t=0} \mathbb{P}[r(x_n) \le t, y_n = h(x_n)] \cdot d\mathbb{P}[r(x_b) > t, y_b \ne g(x_b)]$$

$$= -\mathbb{P}[y_n = h(x_n)] \cdot \mathbb{P}[y_b \ne g(x_b)] \cdot \int_{t=1}^{t=0} \mathbb{P}[r(x_n) \le t | y_n = h(x_n)] \cdot d\mathbb{P}[r(x_b) > t | y_b \ne g(x_b)]$$

$$= -\mathbb{P}[y_n = h(x_n)] \cdot \mathbb{P}[y_b \ne g(x_b)] \cdot \int_{t=0}^{t=1} \mathbb{P}[r(x_n) \le t | y_n = h(x_n)] \cdot d\mathbb{P}[r(x_b) \le t | y_b \ne g(x_b)]$$

$$= -\mathbb{P}[y_n = h(x_n)] \cdot \mathbb{P}[y_b \ne g(x_b)] \cdot \mathbb{P}[r(x_b) > r(x_n) | y_b \ne g(x_b), y_n = h(x_n)]$$

$$= -\mathbb{P}[r(x_b) > r(x_n), y_b \ne g(x_b), y_n = h(x_n)]$$

Thus,

$$
\begin{aligned}
\text{OpenworldAUC} &= (a) + (b) \\
&= \mathbb{P}[r(x_b) > r(x_n), y_n = h(x_n)] - \mathbb{P}[r(x_b) > r(x_n), y_b \neq g(x_b), y_n = h(x_n)] \\
&= \mathbb{P}[r(x_b) > r(x_n), y_b = g(x_b), y_n = h(x_n)] \\
&= \mathop{\mathbb{E}}_{\substack{(x_b, y_b) \sim \mathcal{D}_b \\ (x_n, y_n) \sim \mathcal{D}_n}} \Big[ \underbrace{\mathbf{1}[y_b = g(\boldsymbol{x}_b)]}_{\textbf{Base (P2)}} \cdot \underbrace{\mathbf{1}[r(\boldsymbol{x}_b) > r(\boldsymbol{x}_n)]}_{\textbf{Base-to-New (P1)}} \cdot \underbrace{\mathbf{1}[y_n = h(\boldsymbol{x}_n)]}_{\textbf{New (P2)}} \Big]
\end{aligned}
$$

$\square$

### A.4. Proof for Proposition 4.3.

**Restate of Proposition 4.3.** For any $(r, g, h)$, we have:

- Assume that OpenworldAUC equals $u$, a lower bound proportional to $u$ is available for $\text{HitRate}_n$ for **any domain distributions**.

- Constructing $(\tilde{r}, \tilde{g}, \tilde{h})$ as outlined in Prop.4.1 yields $\text{OpenworldAUC}(\tilde{r}, \tilde{g}, \tilde{h}) < \text{OpenworldAUC}(r, g, h)$.

*Proof.* We first demonstarte the first property that OpenworldAUC **avoids domination by domains with larger sample sizes**. To this end, we review that the OverallAcc is defined as:

$$
\text{OverallAcc} = \frac{\sum_{i \in \mathcal{Y}_b \cup \mathcal{Y}_n} \text{TP}_i}{\sum_{i \in \mathcal{Y}_b \cup \mathcal{Y}_n} (\text{TP}_i + \text{FN}_i)} = \mathbb{P}_b \cdot \text{TPR}_b + \mathbb{P}_n \cdot \text{TPR}_n
$$

$$
\text{TPR}_b = \frac{\sum_{i \in \mathcal{Y}_b} \text{TP}_i}{\sum_{i \in \mathcal{Y}_b} (\text{TP}_i + \text{FN}_i)}, \text{TPR}_n = \frac{\sum_{i \in \mathcal{Y}_n} \text{TP}_i}{\sum_{i \in \mathcal{Y}_n} (\text{TP}_i + \text{FN}_i)}
$$

$$
\mathbb{P}_b = \frac{N_b}{N}, \mathbb{P}_n = \frac{N_n}{N}
$$

From this formulation, OverallAcc can be **dominated** by performance in the domain with more samples. To be specific, a very low $\text{TPR}_n$ with fewer new samples in one prediction can still yield similar OverallAcc scores if $\text{TPR}_b$ compensates. In view of this, the OverallAcc metric is inconsistent with the actual performance of model in OPT setting. In contrast, our OpenworldAUC metric is a ranking-based metric that is not affected by the sample size of each domain. Thus, this metric can effectively avoid the issue of domination by domains with larger sample sizes. To illustrate this, we show that the OpenworldAUC metric guarantees a lower bound for the HitRate of new domain.

According to the definition of OpenworldAUC, we have:

$$
1 - \text{OpenworldAUC} \geq \frac{\left( \sum_{i \in \mathcal{Y}_b} \text{FN}_i \right) \cdot \left( \sum_{j \in \mathcal{Y}_n} \text{FN}_j \right)}{N_b \cdot N_n}
$$

where $N_b$ and $N_n$ are the numbers of base samples and novel samples respectively. Then, given the definition of $\text{HitRate}_n$ and $\text{MissRate}_b$, we have:

$$
\text{MissRate}_b = \frac{\left( \sum_{i \in \mathcal{Y}_b} \text{FN}_i \right)}{N_b}, \text{HitRate}_n = \frac{N_n - \sum_{j \in \mathcal{Y}_n} \text{FN}_j}{N_n}
$$

Thus,

$$
1 - \text{OpenworldAUC} \geq \text{MissRate}_b \cdot (1 - \text{HitRate}_n)
$$

We have:

$$
\text{HitRate}_n \geq 1 - \frac{1 - \text{OpenworldAUC}}{\text{MissRate}_b}
$$

Therefore, when OpenworldAUC equals $u$ and $\mathsf{MissRate}_b = a$, we have $\mathsf{HitRate}_n \geq 1 - \frac{1-u}{a}$ under arbitrary base/new ratio. We conclude that our proposed metric avoids the limitations highlighted in Prop. 4.1.

**Escape from the limitations of Prop.4.1**. This inconsistency arises because $\mathsf{Lin}(\mathsf{AUROC}(r), \mathsf{BaseAcc}(g), \mathsf{NewAcc}(h))$ evaluates performance in a decoupling manner. In contrast, a sample contributes to the OpenworldAUC score only when the detector correctly identifies it, and the domain classifier simultaneously classifies it correctly.

To further examine this issue, we analyze the case where the $\mathsf{Lin}$ metric becomes inconsistent. We consider the following two cases. The difference between the two cases is marked in **red**.

**Case 1:** $(\tilde{r}, \tilde{g}, \tilde{h})$ and $(r, g, h)$ make the predictions as follows:

- $g(x_b^1), \tilde{g}(x_b^1) = y_b^1$ and $g(x_b^2), \tilde{g}(x_b^2) \neq y_b^2$

- $h(x_n^1), \tilde{h}(x_n^1) \neq y_n^1$ and $h(x_n^2), \tilde{h}(x_n^2) \neq y_n^2$

- $r(x_b^1) > r(x_n^1) > r(x_b^2) > r(x_n^2)$

- $\tilde{r}(x_b^2) > \tilde{r}(x_n^2) > \tilde{r}(x_b^1) > \tilde{r}(x_n^1)$

- $r(x_b^1) = \tilde{r}(x_b^2), r(x_n^1) = \tilde{r}(x_n^2), r(x_b^2) = \tilde{r}(x_b^1), r(x_n^2) = \tilde{r}(x_n^1)$

In this case, one can naturally conclude:

$\mathsf{Lin}(\mathsf{AUROC}(r), \mathsf{BaseAcc}(g), \mathsf{NewAcc}(h)) = \mathsf{Lin}(\mathsf{BaseAcc}(\tilde{g}), \mathsf{NewAcc}(\tilde{h}), \mathsf{AUROC}(\tilde{r}))$.

Next, we analyze this case in our metric as following:

$$
\mathsf{OpenworldAUC}(r, g, h) - \mathsf{OpenworldAUC}(\tilde{r}, \tilde{g}, \tilde{h})
$$

$$
= \frac{1}{N_b \cdot N_n} \sum_{j=1}^{N_n} \left[ \mathbb{1}[g(x_b^1) = y_b^1] \cdot \mathbb{1}[r(x_b^1) > r(x_n^j)] \cdot \mathbb{1}[h(x_n^j) = y_n^j] - \mathbb{1}[\tilde{g}(x_b^1) = y_b^1] \cdot \mathbb{1}[\tilde{r}(x_b^1) > \tilde{r}(x_n^j)] \cdot \mathbb{1}[\tilde{h}(x_n^j) = y_n^j] \right]
$$

$$
= \frac{1}{N_b \cdot N_n} \sum_{j=1}^{N_n} \left[ \mathbb{1}[r(x_b^1) > r(x_n^j)] \cdot \mathbb{1}[h(x_n^j) = y_n^j] - \mathbb{1}[r(x_b^2) > r(x_n^j)] \cdot \mathbb{1}[h(x_n^j) = y_n^j] \right]
$$

$$
= \frac{1}{N_b \cdot N_n} \sum_{j=1}^{N_n} \left[ \mathbb{1}[r(x_b^1) > r(x_n^j) > r(x_b^2)] \right] \cdot [h(x_n^j) = y_n^j] \geq 0
$$

where $N_b$ and $N_n$ denote the base samples and novel samples respectively.

The last equality holds only when there are no correctly classified novel samples between $r(x_b^1)$ and $r(x_b^2)$. **This condition is no mild.** Thus, we can conclude that $\mathsf{OpenworldAUC}(r, g, h) > \mathsf{OpenworldAUC}(\tilde{r}, \tilde{g}, \tilde{h})$ in this scenario.

**Case 2:** If $(\tilde{r}, \tilde{g}, \tilde{h})$ and $(r, g, h)$ make the predictions as follows:

- $g(x_b^1), \tilde{g}(x_b^1) = y_b^1$ and $g(x_b^2), \tilde{g}(x_b^2) \neq y_b^2$

- $h(x_n^1), \tilde{h}(x_n^1) \neq y_n^1$ and $h(x_n^2), \tilde{h}(x_n^2) = y_n^2$

- $r(x_b^1) > r(x_n^1) > r(x_b^2) > r(x_n^2)$

- $\tilde{r}(x_b^2) > \tilde{r}(x_n^2) > \tilde{r}(x_b^1) > \tilde{r}(x_n^1)$

- $r(x_b^1) = \tilde{r}(x_b^2), r(x_n^1) = \tilde{r}(x_n^2), r(x_b^2) = \tilde{r}(x_b^1), r(x_n^2) = \tilde{r}(x_n^1)$

The equality $\mathsf{Lin}(\mathsf{AUROC}(r), \mathsf{BaseAcc}(g), \mathsf{NewAcc}(h)) = \mathsf{Lin}(\mathsf{BaseAcc}(\tilde{g}), \mathsf{NewAcc}(\tilde{h}), \mathsf{AUROC}(\tilde{r}))$ still holds.

Similarly, we analyze this case in our metric,

$\mathsf{OpenworldAUC}(r, g, h) - \mathsf{OpenworldAUC}(\tilde{r}, \tilde{g}, \tilde{h})$

$$= \frac{1}{N_b \cdot N_n} \sum_{j=1}^{N_n} \Big[ \mathbf{1}[g(x_b^1) = y_b^1] \cdot \mathbf{1}[r(x_b^1) > r(x_n^j)] \cdot \mathbf{1}[h(x_n^j) = y_n^j] - \mathbf{1}[\tilde{g}(x_b^1) = y_b^1] \cdot \mathbf{1}[\tilde{r}(x_b^1) > \tilde{r}(x_n^j)] \cdot \mathbf{1}[\tilde{h}(x_n^j) = y_n^j] \Big]$$

$$+ \frac{1}{N_b \cdot N_n} \sum_{i=1}^{N_b} \Big[ \mathbf{1}[g(x_b^i) = y_b^i] \cdot \mathbf{1}[r(x_b^i) > r(x_n^2)] \cdot \mathbf{1}[h(x_n^2) = y_n^2] - \mathbf{1}[\tilde{g}(x_b^i) = y_b^i] \cdot \mathbf{1}[\tilde{r}(x_b^i) > \tilde{r}(x_n^2)] \cdot \mathbf{1}[\tilde{h}(x_n^2) = y_n^2] \Big]$$

$$- \frac{1}{N_b \cdot N_n} \mathbf{1}[g(x_b^1) = y_b^1] \cdot \mathbf{1}[r(x_b^1) > r(x_n^2)] \cdot \mathbf{1}[h(x_n^2) = y_n^2]$$

$$= \frac{1}{N_b \cdot N_n} \sum_{j=1}^{N_n} \Big[ \mathbf{1}[r(x_b^1) > r(x_n^j)] \cdot \mathbf{1}[h(x_n^j) = y_n^j] - \mathbf{1}[\tilde{r}(x_b^1) > \tilde{r}(x_n^j)] \cdot \mathbf{1}[\tilde{h}(x_n^j) = y_n^j] \Big]$$

$$+ \frac{1}{N_b \cdot N_n} \sum_{i=1}^{N_b} \Big[ \mathbf{1}[g(x_b^i) = y_b^i] \cdot \mathbf{1}[r(x_b^i) > r(x_n^2)] - \mathbf{1}[\tilde{g}(x_b^i) = y_b^i] \cdot \mathbf{1}[\tilde{r}(x_b^i) > \tilde{r}(x_n^2)] \Big] - \frac{1}{N_b \cdot N_n}$$

$$= \frac{1}{N_b \cdot N_n} \sum_{j=1}^{N_n} \Big[ \mathbf{1}[r(x_b^1) > r(x_n^j), \tilde{r}(x_b^1) < \tilde{r}(x_n^j)] \cdot \mathbf{1}[h(x_n^j) = y_n^j] \Big]$$

$$+ \frac{1}{N_b \cdot N_n} \sum_{i=1}^{N_b} \Big[ \mathbf{1}[g(x_b^i) = y_b^i] \cdot \mathbf{1}[r(x_b^i) > r(x_n^2), \tilde{r}(x_b^i) < \tilde{r}(x_n^2)] \Big] - \frac{1}{N_b \cdot N_n}$$

$$\geq \frac{1}{N_b \cdot N_n} \Big[ \mathbf{1}[r(x_b^1) > r(x_n^2), \tilde{r}(x_b^1) < \tilde{r}(x_n^2)] \cdot \mathbf{1}[h(x_n^2) = y_n^2] \Big]$$

$$+ \frac{1}{N_b \cdot N_n} \Big[ \mathbf{1}[g(x_b^1) = y_b^1] \cdot \mathbf{1}[r(x_b^1) > r(x_n^2), \tilde{r}(x_b^1) < \tilde{r}(x_n^2)] \Big] - \frac{1}{N_b \cdot N_n}$$

$$= \frac{1}{N_b \cdot N_n}$$

Finally, we have:

$$\mathsf{OpenworldAUC}(r, g, h) - \mathsf{OpenworldAUC}(\tilde{r}, \tilde{g}, \tilde{h}) > 0$$

This further demonstrates that OpenworldAUC with joint probability formulation is more **consistent** with the model performance than the naive aggregation of multiple metrics. $\qquad\square$

### A.5. Proof for Proposition 5.1

**Restate of Proposition 5.1.** Maximizing the OpenworldAUC metric can be realized by solving the following minimization problem:

$$\min_{r,g,h} \mathbb{E}_{\mathcal{D}} \left[ \mathbf{1}[y_b \neq g(x_b)] + \mathbf{1}[y_n \neq h(x_n)] + \mathbf{1}[y_b = g(x_b)] \cdot \mathbf{1}[r(x_b) \leq r(x_n)] \cdot \mathbf{1}[y_n = h(x_n)] \right]$$

*Proof.* We first review the definition of OpenworldAUC as:

$$\mathbb{E}_{\mathcal{D}} \Big[ \underbrace{\mathbf{1}[y_b = g(\boldsymbol{x}_b)]}_{I_g} \cdot \underbrace{\mathbf{1}[r(\boldsymbol{x}_b) > r(\boldsymbol{x}_n)]}_{I_r} \cdot \underbrace{\mathbf{1}[y_n = h(\boldsymbol{x}_n)]}_{I_h} \Big]$$

According to the Truth Table in Tab.5, we have:

$$1 - I_g \cdot I_r \cdot I_h = \neg I_g + I_g \cdot \neg I_r \cdot I_h + \neg I_h$$

Thus, maximizing $I_g \cdot I_r \cdot I_h$ is equivalent to minimize $\neg I_g + I_g \cdot \neg I_r \cdot I_h + \neg I_h$. Therefore, the maximization of OpenworldAUC is equivalent to the following minimization problem:

$$\min_{r,g,h} \mathbb{E}_{\mathcal{D}} \left[ \mathbf{1}[y_b \neq g(x_b)] + \mathbf{1}[y_n \neq h(x_n)] + \mathbf{1}[y_b = g(x_b)] \cdot \mathbf{1}[r(x_b) \leq r(x_n)] \cdot \mathbf{1}[y_n = h(x_n)] \right]$$

*Table 5.* Truth Table for the Objective Function.

| $I_g$ | $I_r$ | $I_h$ | $1 - I_g \cdot I_r \cdot I_h$ | $\neg I_g + I_g \cdot \neg I_r \cdot I_h + \neg I_h$ |
|---|---|---|---|---|
| 1 | 1 | 1 | 0 | 0 |
| 1 | 1 | 0 | 1 | 1 |
| 1 | 0 | 1 | 1 | 1 |
| 1 | 0 | 0 | 1 | 1 |
| 0 | 1 | 1 | 1 | 1 |
| 0 | 1 | 0 | 1 | 1 |
| 0 | 0 | 1 | 1 | 1 |
| 0 | 0 | 0 | 1 | 1 |

This completes the proof.

$\square$

# B. Proof for the Generalization Bound

In this section, we present the complete proof of our theoretical results. Tab.6 summarizes the key notations used in the proof. We begin with a proof sketch to give an intuitive overview. Next, we provide the key lemma and then present the detailed proof of the main results. Our primary theoretical results show how well the proposed method can generalize to test data.

## B.1. Proof Outline

We first define the hypothesis space of functions $r$ and $s_b$.

**The Hypothesis Class** In this paper, we consider the generalization ability base on the prompt architecture. Our functions are chosen from the following hypothesis class $\mathcal{H}_r$ and $\mathcal{H}_g$.

$$\mathcal{H}_r = \left\{ r^{(k)}(\cdot; \boldsymbol{\theta}_{r,k}) : \mathcal{X} \to [0,1], k \in [1, K], \boldsymbol{\theta}_{r,k} \text{ represents learnable tokens in the prompt } P(\cdot; \boldsymbol{\theta}_{r,k}) \right\}$$

where $r^{(k)}(x; \theta_{r,k})$ is the possibility that $x$ belongs to the base domain. For the sake of simplicity, we denote the ensemble $\{r^{(k)}(x; \boldsymbol{\theta}_{r,k})\}_{k \in [K]}$ as $r(x)$ and use $r \in \mathcal{H}_r$ to express choosing one such collections out of $\mathcal{H}_r$.

$$\mathcal{H}_g = \left\{ \boldsymbol{s}_b(\cdot; \boldsymbol{\theta}_g) : \mathcal{X} \to \mathbb{R}^{C_b}, \boldsymbol{\theta}_g \text{ represents learnable tokens in the prompt } P(\cdot; \boldsymbol{\theta}_g) \right\}$$

Here $\boldsymbol{s}_b(x)$ abbreviates $\boldsymbol{s}_b(x; \boldsymbol{\theta}_g)$ and use $\boldsymbol{s}_b \in \mathcal{H}_g$ to express choosing one out of the hypothesis class $\mathcal{H}_g$.

**The Norm of Hypothesis** To measure the complexity of $\mathcal{H}_r$ and $\mathcal{H}_g$, we define a norm on each hypothesis respectively. Here, we adopt the overall infinity norm (all classes and all input features $x \in \mathcal{X}$):

$$\|r\|_\infty := \sup_{x \in \mathcal{X}} |r(x)|$$

$$\|\boldsymbol{s}_b\|_\infty := \max_{j \in \{1, 2, \cdots, C_b\}} \sup_{x \in \mathcal{X}} \left| \boldsymbol{s}_b^{(j)}(x) \right|$$

Here $\boldsymbol{s}_b^{(j)}(x)$ is the logit score of the $j$-th channel for raw feature $x$.

In practical training, we aim to solve the following $OP_{\text{fin}}$ problem:

$$(OP_{\text{fin}}) \min_{r,g} \hat{\mathcal{R}}'(r, g, h) = \hat{\mathbb{E}}_{\mathcal{S}_b} \left[ \ell_{ce}(\boldsymbol{s}_b, y_b) \right] + \frac{1}{K} \cdot \hat{\mathbb{E}}_{\hat{\mathcal{S}}^{(k)}} [\varphi_b \cdot \ell_{sq}(r(\boldsymbol{x}_b) - r(\boldsymbol{x}_n)) \cdot \varphi_n]$$

where $\hat{\mathcal{S}}^{(k)}$ is sampled from the data distribution $\hat{\mathcal{D}}^{(k)} := \mathcal{X} \times \hat{\mathcal{Y}}^{(k)}$ under the $k$-th base-to-new pseudo partition.

Theoretically, the generalization gap $\Delta$ should be measured by the expected risk on the joint distribution of overall label space $\mathcal{Y}$ and the training data $\mathcal{D}$, expressed as:

$$\mathcal{R}(r, g, h) = \mathbb{E}_{\mathcal{D}_b} \left[ \ell_{ce}(\boldsymbol{s}_b, y_b) \right] + \mathbb{E}_{\mathcal{Y}} \mathbb{E}_{\mathcal{D}} [\varphi_b \cdot \ell_{sq}(r(\boldsymbol{x}_b) - r(\boldsymbol{x}_n)) \cdot \varphi_n] + \mathbb{E}_{\mathcal{D}_n} \left[ \ell_{ce}(\boldsymbol{s}_n, y_n) \right]$$

Assume that the detection function $r$, base domain score function $s_b$ are chosen from the hypothesis space $\mathcal{H}_r$, $\mathcal{H}_g$, respectively. The generalization ability of the entire hypothesis set is often measured by the worst-case generalization gap $\Delta$:

$$\Delta := \sup_{\boldsymbol{s}_b \in \mathcal{H}_g, r \in \mathcal{H}_r} \left[ \mathcal{R}(r, g, h) - \hat{\mathcal{R}}'(r, g, h) \right]$$

However, this gap is complex. Intuitively, it mainly comes from three parts:

$$\Delta = \Delta_r + \Delta_g + \Delta_h$$

where $\Delta_r$, $\Delta_g$, and $\Delta_h$ are the generalization gaps for the detector $r$, base-domain classifier $g$, and new-domain classifier $h$, respectively.

*Table 6.* Some Important Notations Used in the Proof.

| Notation | Description |
|---|---|
| **Sigmoid score** | |
| $\varphi_b^i = \sigma(\boldsymbol{s}_b^{(y)}(x_b^i))$ | The gate score of $i$-th base-domain sample. |
| $\varphi_n^j = \sigma(\boldsymbol{s}_n^{(y)}(x_n^j))$ | The gate score of $j$-th new-domain sample. |
| $v_\infty$ | $\sup_x \left\|\sigma(\boldsymbol{s}^{(y)}(x))\right\|$, $\boldsymbol{s} \in \{\boldsymbol{s}_b, \boldsymbol{s}_n\}$. The overall infinity norm on all input features $\mathcal{X}$. |
| **AUC-type Risks** | |
| $\ell_{sq}(r, x_b^i, x_n^j)$ | $\ell_{sq}(r(x_b^i) - r(x_n^j))$. |
| $\ell_\infty$ | $\sup_{(x_b, x_n)} \|\ell_{sq}(r, x_b, x_n)\|$ The overall infinity norm on all input features $\mathcal{X}$. |
| $\hat{R}_{\hat{S}}(r, g, h)$ | $\hat{\mathbb{E}}_{\substack{(x_b, y_b) \sim \hat{S}_b \\ (x_n, y_n) \sim \hat{S}_n}} \varphi_b \cdot \ell_{sq}(r, x_n, x_b) \cdot \varphi_n = \frac{1}{\tilde{N}_b \cdot \tilde{N}_n} \sum_{i=1}^{\tilde{N}_b} \sum_{j=1}^{\tilde{N}_n} \varphi_b^i \cdot \ell_{sq}(r(x_b^i) - r(x_n^j)) \cdot \varphi_n^j$ .  The empirical loss on training data under a fixed pseudo partition $\hat{\mathcal{Y}}$ |
| $R_{\hat{D}}(r, g, h)$ | $\mathbb{E}_{\substack{(x_b, y_b) \sim \hat{D}_b \\ (x_n, y_n) \sim \hat{D}_n}} \varphi_b \cdot \ell_{sq}(r, x_n, x_b) \cdot \varphi_n$. The expected loss under a fixed pseudo partition $\hat{\mathcal{Y}}$. |
| $\hat{R}_{\hat{S}^{(k)}}(r, g, h)$ | $\hat{\mathbb{E}}_{\substack{(x_b, y_b) \sim \hat{S}_b^{(k)} \\ (x_n, y_n) \sim \hat{S}_n^{(k)}}} \varphi_b \cdot \ell_{sq}(r, x_n, x_b) \cdot \varphi_n$. The empirical risk on training data under a $k$-th pseudo partition $\hat{\mathcal{Y}}^{(k)}$. |
| $R_{\hat{D}^{(k)}}(r, g, h)$ | $\mathbb{E}_{\substack{(x_b, y_b) \sim \hat{D}_b^{(k)} \\ (x_n, y_n) \sim \hat{D}_n^{(k)}}} \varphi_b \cdot \ell_{sq}(r, x_n, x_b) \cdot \varphi_n$. The expected loss on training data under a $k$-th pseudo partition $\hat{\mathcal{Y}}^{(k)}$. |
| $\hat{\mathbb{E}}_{\hat{\mathcal{Y}}}\left[\hat{R}_{\hat{S}}(r, g, h)\right]$ | $\frac{1}{K}\sum_{k=1}^K \hat{R}_{\hat{S}^{(k)}}$. The empirical average of $\hat{R}_{\hat{S}^{(k)}}$ over mutiple partitions. |
| $\hat{\mathbb{E}}_{\hat{\mathcal{Y}}}\left[R_{\hat{D}}(r, g, h)\right]$ | $\frac{1}{K}\sum_{k=1}^K R_{\hat{D}^{(k)}}(r, g, h)$. The empirical average of $R_{\hat{D}^{(k)}}(r, g, h)$ over mutiple partitions. |
| $\mathbb{E}_{\hat{\mathcal{Y}}}\left[R_{\hat{D}}(r, g, h)\right]$ | The expected $R_{\hat{D}}(r, g, h)$ over the meta distribution of pseudo partition distribution. |
| $\mathbb{E}_{\mathcal{Y}}\left[R_{\mathcal{D}}(r, g, h)\right]$ | The expected $R_{\mathcal{D}}(r, g, h)$ over the meta distribution of real partition distribution. |
| **Distribution** | **Generic definition** |
| $\mathcal{X}$ | Input space. |
| $\mathcal{Y} := \mathcal{Y}_b \cup \mathcal{Y}_n$ | Overall label space. |
| $\mathcal{Y}_b$ | Base label space. |
| $\mathcal{Y}_n$ | New label space. |
| $\mathcal{D}_b := \mathcal{X} \times \mathcal{Y}_b$ | Base domain data distribution. |
| $\mathcal{S}_b := \{(\boldsymbol{x}_b^i, y_b^i)\}_{i=1}^{N_b}$ | The base dataset sampled from $\mathcal{D}_b$ |
| $\mathcal{D}_n := \mathcal{X} \times \mathcal{Y}_n$ | New domain data distribution. |
| $\mathcal{S}_n$ | The new dataset sampled from $\mathcal{D}_n$ |
| $\hat{\mathcal{Y}} = \left(\hat{\mathcal{Y}}_b, \hat{\mathcal{Y}}_n\right)$ | $\hat{\mathcal{Y}}$ is the pseudo base-to-new partition distribution. |
| $\hat{\mathcal{Y}}^{(k)} = \left(\hat{\mathcal{Y}}_b^{(k)}, \hat{\mathcal{Y}}_n^{(k)}\right)$ | The $k$-th base-to-new pseudo partition during the training stage. $k \in [1, K]$. |
| $K$ | The number of base-to-new pseudo partition during the training stage. |
| $\hat{\mathcal{Y}}_b^{(k)}$ | The **pseudo** base domain in $k$-**th** base-to-new pseudo partition. |
| $\hat{\mathcal{Y}}_n^{(k)}$ | The **pseudo** new domain in $k$-**th** base-to-new pseudo partition. |
| $\hat{\mathcal{D}} = \left(\hat{\mathcal{D}}_b, \hat{\mathcal{D}}_n\right)$ | $\left(\mathcal{X} \times \hat{\mathcal{Y}}_b, \mathcal{X} \times \hat{\mathcal{Y}}_n\right)$. The data distribution under a fixed base-to-new pseudo partition $\hat{\mathcal{Y}}$. |
| $\hat{\mathcal{S}} = \left(\hat{\mathcal{S}}_b, \hat{\mathcal{S}}_n\right)$ | $\hat{\mathcal{S}}_b, \hat{\mathcal{S}}_n$ are sampled from $\hat{\mathcal{D}}_b$ and $\hat{\mathcal{D}}_n$, respectively. |
| $\hat{\mathcal{D}}^{(k)} = \left(\hat{\mathcal{D}}_b^{(k)}, \hat{\mathcal{D}}_n^{(k)}\right)$ | $\left(\mathcal{X} \times \hat{\mathcal{Y}}_b^{(k)}, \mathcal{X} \times \hat{\mathcal{Y}}_n^{(k)}\right)$ The data distribution under the $k$-th base-to-new pseudo partition. |
| $\hat{\mathcal{D}}_b^{(k)}, \hat{\mathcal{D}}_n^{(k)}$ | The **pseudo** base and new data distribution under the $k$-th base-to-new pseudo partition. |
| $\hat{\mathcal{S}}^{(k)} = \left(\hat{\mathcal{S}}_b^{(k)}, \hat{\mathcal{S}}_n^{(k)}\right)$ | The dataset sampled from $\hat{\mathcal{D}}^{(k)}$. |
| $\hat{\mathcal{S}}_b^{(k)}, \hat{\mathcal{S}}_n^{(k)}$ | The **pseudo** base and new dataset sampled from $\mathcal{D}_b^{(k)}, \mathcal{D}_n^{(k)}$. |

To be specific, $\Delta_g$ is defined as:

$$\Delta_g = \sup_{\boldsymbol{s}_b \in \mathcal{H}_g} \left[ \mathbb{E}_{\mathcal{D}_b} \left[ \ell_{ce}(\boldsymbol{s}_b, y_b) \right] - \hat{\mathbb{E}}_{\mathcal{S}_b} \left[ \ell_{ce}(\boldsymbol{s}_b, y_b) \right] \right]$$

This error arises from approximating the expectation over the training distribution $\mathcal{D}$ using the empirical average from the training data $\mathcal{S}$. Meanwhile, this term meets the basic assumptions of standard generalization analysis techniques, where the loss function is represented as a sum of independent terms. Consequently, bounding this term is relatively straightforward.

$\Delta_r$ is defined as:

$$\Delta_r = \sup_{\boldsymbol{s}_b \in \mathcal{H}_g, r \in \mathcal{H}_r} \left[ \mathbb{E}_{\mathcal{Y}} \left[ R_{\mathcal{D}} \right] - \hat{\mathbb{E}}_{\hat{y}} \left[ \hat{R}_{\hat{\mathcal{S}}} \right] \right]$$

This formulation reveals two key challenges in bounding $\Delta_r$:

- The empirical risk is defined on the $\hat{y}$ and $\hat{\mathcal{S}}$ spaces, while the expected risk is defined over the $\mathcal{Y}$ and $\mathcal{D}$ spaces. This coupled discrepancy makes it difficult to directly bound the generalization gap for $r$. To address this issue, we decompose $\Delta_r$ into three types of errors: partition-distribution approximation error, empirical partitions estimation error, and data estimation error.

- The standard generalization analysis techniques require the loss function to be expressed as a sum of independent terms. However, **the AUC-type risk violates this assumption due to pairwise sample dependency**. For instance, the optimization functions for the detector, $\ell_{sq}(r, x_n^j, x_b^i)$ and $\ell_{sq}(r, \tilde{x}_n^j, \tilde{x}_b^i)$, are interdependent if any term is shared (e.g., $\tilde{x}_n^j = x_n^j$ or $\tilde{x}_b^i = x_b^i$). To address this issue, we use **covering numbers** and $\epsilon$-**net arguments** in the subsequent proof to derive the generalization bound.

We decompose $\Delta_r$ into three types of errors:

- **Partition-distribution Approximation Error:** This error occurs because, the **pseudo** partition distribution $\hat{y}$ is used to approximate the open-world **real** partition distribution $\mathcal{Y}$.

$$\sup_{\boldsymbol{s}_b \in \mathcal{H}_g, r \in \mathcal{H}_r} \left[ \mathbb{E}_{\mathcal{Y}} \left[ R_{\mathcal{D}} \right] - \mathbb{E}_{\hat{y}} \left[ R_{\hat{\mathcal{D}}} \right] \right]$$

- **Empirical Partitions Estimation Error:** This error refers to the discrepancy between the expectation of pseudo base-to-new partition distributions $\hat{y}$ and the empirical averages over $\hat{y}^{(k)}, k \in [1, K]$.

$$\sup_{\boldsymbol{s}_b \in \mathcal{H}_g, r \in \mathcal{H}_r} \left[ \mathbb{E}_{\hat{y}} \left[ R_{\hat{\mathcal{D}}} \right] - \hat{\mathbb{E}}_{\hat{y}} \left[ R_{\hat{\mathcal{D}}} \right] \right]$$

- **Data Estimation Error:** This error stems from approximating the expectation over the training distribution $\hat{\mathcal{D}}$ with the empirical average from the training data $\hat{\mathcal{S}}$

$$\sup_{\boldsymbol{s}_b \in \mathcal{H}_g, r \in \mathcal{H}_r} \left[ \hat{\mathbb{E}}_{\hat{y}} \left[ R_{\hat{\mathcal{D}}} \right] - \hat{\mathbb{E}}_{\hat{y}} \left[ \hat{R}_{\hat{\mathcal{S}}} \right] \right] = \sup_{\boldsymbol{s}_b \in \mathcal{H}_g, r \in \mathcal{H}_r} \left[ \frac{1}{K} \sum_{k=1}^{K} R_{\hat{\mathcal{D}}^{(k)}} - \frac{1}{K} \sum_{k=1}^{K} \hat{R}_{\hat{\mathcal{S}}^{(k)}} \right]$$

$\Delta_h$ is defined as:

$$\Delta_h = \mathbb{E}_{\mathcal{D}_n} \left[ \ell_{ce}(\boldsymbol{s}_n, y_n) \right]$$

Without prior knowledge of the new domain, we cannot bound the expected error. We can reduce this error by carefully designing prompts or using powerful foundation models.

## B.2. Preliminary Lemma

**Definition 1** (Bounded Difference Property). Given a group of independent random variables $X_1, X_2, \cdots, X_n$ where $X_t \in \mathbb{X}, \forall t$, $f(X_1, X_2, \cdots, X_n)$ is satisfied with the bounded difference property, if there exists some non-negative constants $c_1, c_2, \cdots, c_n$, such that:

$$\sup_{x_1, x_2, \cdots, x_n, x_t'} |f(x_1, \cdots, x_n) - f(x_1, \cdots, x_{t-1}, x_t', \cdots, x_n)| \leq c_t, \ \forall t, 1 \leq t \leq n. \tag{1}$$

Hereafter, if any function $f$ holds the Bounded Difference Property, the following Mcdiarmid's inequality is always satisfied.

**Lemma B.1** (Mcdiarmid's Inequality). *Assume we have $n$ independent random variables $X_1, X_2, \ldots, X_n$ that all of them are chosen from the set $\mathcal{X}$. For a function $f : \mathcal{X} \to \mathbb{R}, \forall t, 1 \leq t \leq n$, if the following inequality holds:*

$$\sup_{x_1, x_2, \cdots, x_n, x_t'} |f(x_1, \cdots, x_n) - f(x_1, \cdots, x_{t-1}, x_t', \cdots, x_n)| \leq c_t, \ \forall t, 1 \leq t \leq n.$$

*with $\boldsymbol{x} \neq \boldsymbol{x}'$, then for all $\epsilon > 0$, we have*

$$\mathbb{P}[|f - \mathbb{E}(f)| \geq \epsilon] \leq 2 \cdot \exp\left(\frac{-2\epsilon^2}{\sum_{t=1}^{n} c_t^2}\right).$$

**Lemma B.2** (Hoeffding's Inequality). *If $Z_1, \cdots, Z_n$ are independent random variables such that $Z_i \in [a, b]$ almost surely, then for any $t \geq 0$,*

$$P\left(\left|\frac{1}{n}\sum_{i=1}^{n} Z_i - \frac{1}{n}\sum_{i=1}^{n} \mathbb{E}[Z_i]\right| \geq t\right) \leq 2\exp\left(\frac{-2nt^2}{(a-b)^2}\right).$$

**Definition 2** ($\epsilon$-Covering). Let $(\mathcal{F}, \rho)$ be a (pseudo) metric space, and $\mathcal{G} \subseteq \mathcal{F}$. $\{f_1, \ldots, f_K\}$ is said to be an $\epsilon$-covering of $\mathcal{G}$ if $\mathcal{G} \subseteq \bigcup_{i=1}^{K} \mathcal{B}(f_i, \epsilon)$, i.e., $\forall g \in \mathcal{G}, \exists i$ such that $\rho(g, f_i) \leq \epsilon$.

**Definition 3** (Covering Number). According to the notations in Def.2, the covering number of $\mathcal{G}$ with radius $\epsilon$ is defined as:

$$\mathcal{N}(\epsilon; \mathcal{G}, \rho) = \min\{n : \exists \epsilon - covering \ over \ \mathcal{G} \ with \ size \ n\}$$

**Lemma B.3.** *The covering number of the hypothesis class $\mathcal{H}_R$ has the following upper bound:*

$$\log \mathcal{N}(\epsilon; \mathcal{H}_R, \rho) \leq d \log\left(\frac{3r}{\epsilon}\right), \tag{2}$$

*where $d$ is the dimension of embedding space.*

**Lemma B.4** (Union bound/Boole's inequality). *Given the countable or finite set of events $E_i$, the probability that at least one event happens is less than or equal to the sum of all probabilities of the events happened individually, i.e.,*

$$\mathbb{P}\left[\bigcup_i E_i\right] \leq \sum_i \mathbb{P}[E_i] \tag{3}$$

**Lemma B.5** ($\phi$-Lipschitz Continuous). *Given a set $\mathcal{X}$ and a function $f : \mathcal{X} \to \mathbb{R}$, if $f$ is continuously differentiable on $\mathcal{X}$ such that, $\forall x, y \in \mathcal{X}$, the following condition holds with a real constant $\phi$:*

$$\|f(x) - f(y)\| \leq \phi \|x - y\|.$$

*Thereafter, $f$ is said to be a $\phi$-Lipschitz continuous function.*

**Assumption 1.** The $f_g$, $f_h$, and $r$ are Lipschitz continuous *w.r.t.* the input variable.

**Corollary B.6.** *The* Sigmoid *function is $\frac{1}{4}$-Lipschitz continuous.*

**Corollary B.7.** *The Squared loss function is $2$-Lipschitz continuous.*

**Lemma B.8.** *Let $x$ and $y$ be postive and satisfy $x + y = C$. The maximum value of the function $f(x, y) = \frac{1}{x} + \frac{1}{y}$ is $\frac{4}{C}$.*

**Lemma B.9.** *(Yang et al., 2024) The volume of an $c$-dimensional probability simplex is $c!$. In other words, we have:*

$$V_c = \int_{\sum_{i=1}^c x_i = 1} 1 \cdot dx_1 \cdot dx_2 \cdot \ldots \cdot dx_c = \frac{1}{c!}$$

*Proof.* We proof it by induction.

**Base Case, c=1**. Obviously, we have $V_1 = \int_0^1 dx = 1$.

**Induction**. Supposes that $V_{i-1} = (i-1)!$, we have:

$$
\begin{aligned}
V_i &= \int_{\sum_{j=1}^i x_j = 1} 1 \cdot dx_1 \cdot dx_2 \cdot \ldots \cdot dx_i \\
&= \int_0^1 \left( \int_{\sum_{j=1}^{i-1} x_j = 1 - x_i} 1 \cdot dx_1 \cdot dx_2 \cdot \ldots \cdot dx_{i-1} \right) dx_i \\
&= \int_0^1 (1 - x_i)^{i-1} \cdot \left( \int_{\sum_{j=1}^{i-1} u_j = 1} 1 \cdot du_1 \cdot du_2 \cdot \ldots \cdot du_{i-1} \right) dx_i \\
&= V_{i-1} \cdot \int_0^1 (1 - x_i)^{i-1} dx_i \\
&= (1/i) \cdot V_{i-1} \\
&= 1/i!
\end{aligned}
$$

The proof is then completed by expanding the induction recursively. $\square$

### B.3. Key Lemmas

**Lemma B.10.** *When $g(\cdot)$ is Lipschitz continuous, the following holds:*

$$\|g(x) - g(\tilde{x})\|_\infty \le \sup \|\nabla_x g\|_p \cdot \|x - \tilde{x}\|_q,$$

*where $\frac{1}{p} + \frac{1}{q} = 1$.*

*Proof.*

$$
\begin{aligned}
|g(x) - g(\tilde{x})| &= \left| \int_0^1 \langle \nabla g(\tau x + (1 - \tau)\tilde{x}), x - \tilde{x} \rangle \, d\tau \right| \\
&\le \sup_{x \in \mathcal{X}} \left[ \|\nabla g\|_p \right] \cdot \left\| x - \tilde{x} \right\|_q
\end{aligned}
$$

$\square$

**Corollary B.11.** *Specifically, when $p = 1$ and $q = \infty$, we have*

$$\|g(x) - g(\tilde{x})\|_\infty \le \sup \|\nabla_x g\|_1 \cdot \|x - \tilde{x}\|_\infty.$$

**Lemma B.12.** *The cross-entropy loss function $\ell_{ce}(f(x), y)$, where $(f(x) \in \mathbb{R}^C)$ and $C$ is the number of classes, is 2-Lip continuous w.r.t. the defined infinity norm $\|f\|_\infty$.*

*Proof.* According to Corollary.B.11, if:

$$\sup_{(x,y),f\in\mathcal{F}} \|\nabla_f \ell_{ce}(f(x),y)\|_1 \leq 2 \tag{4}$$

Then we have:

$$\left|\ell_{ce}(f(x),y) - \ell_{ce}(\tilde{f}(x),y)\right| \leq 2 \cdot \|\tilde{f} - f\|_\infty$$

Hence, we only need to proof 4. To see this:

$$\left|\frac{\partial \ell_{ce}(f(x),y)}{\partial f^j(x)}\right| = \left|\sigma^j - \mathbf{1}[j=y]\right|$$

where

$$\sigma^j = \frac{f^j(x)}{\sum_{i=1}^C \exp(f^i(x))}$$

Since we have:

$$\sup_{(x,y),f\in\mathcal{F}} \|\nabla_f \ell_{ce}(f(x),y)\|_1 = \sum_j \left|\sigma^j - \mathbf{1}[j\neq y]\right| \leq 2$$

This proof is completed since $x$, $y$, $f$ are arbitrarily chosen. $\qquad\square$

**Lemma B.13.** *Let $\{x_i, y_i\}_{i=1}^N$ be i.i.d samples from a data distribution, the loss function of a prediction is given by $\ell(, x_i, y_i) = \ell(f(x_i), y_i) \in [0, B]$ for a scoring function $f \in \mathcal{F}$. If the loss function $L_c$-lipschitz continuous w.r.t. the defined infinity norm $\|f\|_\infty$. With high probability, the following inequality holds for all $f \in \mathcal{F}$:*

$$|\mathbb{E}[\ell] - \hat{\mathbb{E}}[\ell]| \leq B \cdot \sqrt{\frac{2d}{N} \cdot \log(3 \cdot r \cdot N)}$$

*Proof.* Based on the basic property of the covering number, we can construct a covering of $\mathcal{F}$ using a set of open balls $\mathcal{B}_1, \cdots, \mathcal{B}_\mathcal{M}$. Each open ball $\mathcal{B}_j$ is centered at $f_j$, where $\mathcal{B}_j = \{f \in \mathcal{F} : \|f - f_j\|_\infty \leq \epsilon/4L_c\}$.

Accoring to the lemma.B.4, the following inequality holds:

$$\mathbb{P}\left[\sup_{f\in\mathcal{F}} \left[\left|\mathbb{E}[\ell] - \hat{\mathbb{E}}[\ell]\right|\right] \geq \epsilon\right] \leq \sum_{j=1}^{\mathcal{M}'} \mathbb{P}\left[\sup_{f\in\mathcal{B}_j} \left[\left|\mathbb{E}[\ell] - \hat{\mathbb{E}}[\ell]\right|\right] \geq \epsilon\right]$$

For one $j$, we have:

$$\mathbb{P}\left[\sup_{f\in\mathcal{B}_j} \left[\left|\mathbb{E}[\ell] - \hat{\mathbb{E}}[\ell]\right|\right] \geq \epsilon\right] \leq \mathbb{P}\left[\sup_{f\in\mathcal{B}_j} \left[|\mathbb{E}[\ell] - \mathbb{E}[\ell_j]|\right] \geq \frac{\epsilon}{4}\right] + \mathbb{P}\left[\left|\mathbb{E}[\ell_j] - \hat{\mathbb{E}}[\ell_j]\right| \geq \frac{\epsilon}{2}\right] + \mathbb{P}\left[\sup_{f\in\mathcal{B}_j} \left[\left|\mathbb{E}[\ell] - \hat{\mathbb{E}}[\ell_j]\right|\right] \geq \frac{\epsilon}{4}\right]$$

Since $\ell$ is $L$-lipschitz continuous *w.r.t.* the defined infinity norm, we have:

$$|\ell - \ell_j| \leq L_c \cdot \|f - f_j\|_\infty \leq L_c \cdot \frac{\epsilon}{4L_c} \leq \frac{\epsilon}{4}.$$

Naturally, we have:

$$\mathbb{P}\left[\sup_{f\in\mathcal{B}_j} \left[|\mathbb{E}[\ell] - \mathbb{E}[\ell_j]|\right] \geq \frac{\epsilon}{4}\right] = 0, \quad \mathbb{P}\left[\sup_{f\in\mathcal{B}_j} \left[\left|\mathbb{E}[\ell] - \hat{\mathbb{E}}[\ell_j]\right|\right] \geq \frac{\epsilon}{4}\right] = 0$$

According to the lemma.B.2, we have:

$$\mathbb{P}\left[\sup_{f\in\mathcal{B}_j} \left[\left|\mathbb{E}[\ell] - \hat{\mathbb{E}}[\ell]\right|\right] \geq \epsilon\right] \leq \mathbb{P}\left[\left|\mathbb{E}[\ell_j] - \hat{\mathbb{E}}[\ell_j]\right| \geq \frac{\epsilon}{2}\right] \leq 2 \cdot \exp\left(-\frac{\epsilon^2 \cdot N}{2 \cdot B^2}\right)$$

Thus,

$$\mathbb{P}\left[\sup_{f\in\mathcal{F}}\left[\left|\mathbb{E}[\ell]-\hat{\mathbb{E}}[\ell]\right|\right]\geq\epsilon\right]\leq 2\cdot\mathcal{N}\left(\mathcal{F},\frac{\epsilon}{4L_c},\|\cdot\|_\infty\right)\cdot\exp\left(-\frac{\epsilon^2\cdot N}{2\cdot B^2}\right)$$

By futher choosing,

$$\epsilon=B\cdot\sqrt{\frac{2d}{N}\cdot\log\left(3\cdot r\cdot N\right)}$$

we have:

$$\mathbb{P}\left[\sup_{f\in\mathcal{F}}\left[\left|\mathbb{E}[\ell]-\hat{\mathbb{E}}[\ell]\right|\right]\geq B\cdot\sqrt{\frac{2d}{N}\cdot\log\left(3\cdot r\cdot N\right)}\right]\leq 2\cdot\left(\frac{4\cdot L_c}{B\cdot\sqrt{2\cdot d\cdot N\cdot\log\left(3\cdot r\cdot N\right)}}\right)^d$$

Finally, we can conclude that, with high probability,

$$\sup_{f\in\mathcal{F}}\left[\left|\mathbb{E}[\ell]-\hat{\mathbb{E}}[\ell]\right|\right]\leq B\cdot\sqrt{\frac{2d}{N}\cdot\log\left(3\cdot r\cdot N\right)}$$

This completed the proof. $\qquad\square$

**Lemma B.14.** *Assume that the gated weighting function $\varphi(\cdot)$ is $L_v$-lipschitz continuous and the loss function $\ell$ is $L_\ell$-Lipschitz continuous. Let $\epsilon$ be the generalization error between the empirical risk $\hat{\mathbb{E}}_{\hat{y}}\left[R_{\hat{\mathcal{D}}}(r,g,h)\right]$ and the expected risk $\mathbb{E}_{\hat{y}}\left[R_{\hat{\mathcal{D}}}(r,g,h)\right]$. Then by constructing $\sigma$-covering $\{r_1,r_2,\cdots,r_{\mathcal{N}_r}\}$ of $\mathcal{H}_r$ with $\epsilon_r$ and $\sigma$-covering $\left\{s_{b,1},s_{b,2},\cdots,s_{b,\mathcal{N}_g}\right\}$ of $\mathcal{H}_g$ with $\epsilon_g$, the following inequality holds:*

$$\mathbb{P}\left[\sup_{r\in\mathcal{B}(r_l,\epsilon_r),s_b\in\mathcal{B}(s_{b,u},\epsilon_g)}\left|\mathbb{E}_{\hat{y}}\left[R_{\hat{\mathcal{D}}}(r,g,h)\right]-\hat{\mathbb{E}}_{\hat{y}}\left[R_{\hat{\mathcal{D}}}(r,g,h)\right]\right|\leq\epsilon\right]\geq\mathbb{P}\left[\left|\mathbb{E}_{\hat{y}}\left[R_{\hat{\mathcal{D}}}(r_l,g_u,h)\right]-\hat{\mathbb{E}}_{\hat{y}}\left[R_{\hat{\mathcal{D}}}(r_l,g_u,h)\right]\right|\leq\frac{\epsilon}{2}\right]$$

*Proof.* Some simplified notations applicable only to this proof are given as follows:

$$\sigma(s_b(x_b))=\varphi_b,\quad\sigma(s_{b,u}(x_b))=\tilde{\varphi}_b,\quad\sigma(s_n(x_n))=\varphi_n$$
$$\ell_{sq}(r(x_b)-r(x_n))=\ell,\quad\ell_{sq}(r_l(x_b)-r_l(x_n))=\tilde{\ell}$$

We first turn to prove the following inequality,

$$\left|\left|\mathbb{E}_{\hat{y}}\left[R_{\hat{\mathcal{D}}}(r,g,h)\right]-\hat{\mathbb{E}}_{\hat{y}}\left[R_{\hat{\mathcal{D}}}(r,g,h)\right]\right|-\left|\mathbb{E}_{\hat{y}}\left[R_{\hat{\mathcal{D}}}(r_l,g_u,h)\right]-\hat{\mathbb{E}}_{\hat{y}}\left[R_{\hat{\mathcal{D}}}(r_l,g_u,h)\right]\right|\right|$$
$$\overset{(*)}{\leq}\left|\hat{\mathbb{E}}_{\hat{y}}\left[R_{\hat{\mathcal{D}}}(r,g,h)\right]-\hat{\mathbb{E}}_{\hat{y}}\left[R_{\hat{\mathcal{D}}}(r_l,g_u,h)\right]\right|+\left|\mathbb{E}_{\hat{y}}\left[R_{\hat{\mathcal{D}}}(r,g,h)\right]-\mathbb{E}_{\hat{y}}\left[R_{\hat{\mathcal{D}}}(r_l,g_u,h)\right]\right|\leq\frac{\epsilon}{2}$$

$(*)$ follows the rule that $|x+y|\leq|x|+|y|$ and $||x|-|y||\leq|x-y|$. Thus, we only need to prove the following inequality,

$$\left|\hat{\mathbb{E}}_{\hat{y}}\left[R_{\hat{\mathcal{D}}}(r,g,h)\right]-\hat{\mathbb{E}}_{\hat{y}}\left[R_{\hat{\mathcal{D}}}(r_l,g_u,h)\right]\right|\leq\frac{\epsilon}{4}$$

We have,

$$\left| \hat{\mathbb{E}}_{\hat{y}} \left[ R_{\hat{\mathcal{D}}} \left( r, g, h \right) \right] - \hat{\mathbb{E}}_{\hat{y}} \left[ R_{\hat{\mathcal{D}}} \left( r_l, g_u, h \right) \right] \right|$$

$$\overset{(*)}{\leq} \left| \frac{1}{K} \sum_{k=1}^{K} \left[ R_{\hat{\mathcal{D}}^{(k)}}(r, g, h) - R_{\hat{\mathcal{D}}^{(k)}}(r_l, g_u, h) \right] \right|$$

$$\leq \max_{\hat{\mathcal{Y}}^{(k)}} \left| R_{\hat{\mathcal{D}}^{(k)}}(r, g, h) - R_{\hat{\mathcal{D}}^{(k)}}(r_l, g_u, h) \right|$$

$$\leq \max_{\hat{\mathcal{Y}}^{(k)}} \mathop{\mathbb{E}}_{\substack{(x_b, y_b) \sim \hat{\mathcal{D}}_b^{(k)} \\ (x_n, y_n) \sim \hat{\mathcal{D}}_n^{(k)}}} \left| \varphi_b \cdot \ell \cdot \varphi_n - \tilde{\varphi}_b \cdot \tilde{\ell} \cdot \varphi_n \right|$$

$$\overset{(**)}{\leq} \max_{\hat{\mathcal{Y}}^{(k)}} \mathop{\mathbb{E}}_{\substack{(x_b, y_b) \sim \hat{\mathcal{D}}_b^{(k)} \\ (x_n, y_n) \sim \hat{\mathcal{D}}_n^{(k)}}} \left\{ \left| \varphi_b \cdot \ell \cdot \varphi_n - \varphi_b \cdot \tilde{\ell} \cdot \varphi_n \right| + \left| \varphi_b \cdot \tilde{\ell} \cdot \varphi_n - \tilde{\varphi}_b \cdot \tilde{\ell} \cdot \varphi_n \right| \right\}$$

$$\overset{(***)}{\leq} \max_{\hat{\mathcal{Y}}^{(k)}} \mathop{\mathbb{E}}_{\substack{(x_b, y_b) \sim \hat{\mathcal{D}}_b^{(k)} \\ (x_n, y_n) \sim \hat{\mathcal{D}}_n^{(k)}}} \left\{ 2 \cdot L_\ell \cdot v_\infty^2 \cdot \|r - r_\ell\|_\infty + \ell_\infty \cdot L_v \cdot \|s_b - s_{b,u}\|_\infty \right\}$$

$$\leq 2 \cdot L_\ell \cdot v_\infty^2 \cdot \epsilon_r + \ell_\infty \cdot L_v \cdot \epsilon_g \leq \frac{\epsilon}{4}$$

$(*)$ follows the fact that we perform $K$ base-to-new pseudo partitions. $(**)$ follows the rule that $|x + y| \leq |x| + |y|$. $(***)$ follows the Lem.B.5. By further choosing $\epsilon_r = \frac{\epsilon}{8 \cdot L_\ell \cdot v_\infty^2 + 4 \cdot \ell_\infty \cdot L_v}$ and $\epsilon_g = \frac{\epsilon}{8 \cdot L_\ell \cdot v_\infty^2 + 4 \cdot \ell_\infty \cdot L_v}$, we could construct the covering number $\mathcal{N}_r$ and $\mathcal{N}_g$:

$$\mathcal{N}(\epsilon_r = \frac{\epsilon}{8 \cdot L_\ell \cdot v_\infty^2 + 4 \cdot \ell_\infty \cdot L_v}, \mathcal{H}_r, \|\cdot\|_\infty)$$

$$\mathcal{N}(\epsilon_g = \frac{\epsilon}{8 \cdot L_\ell \cdot v_\infty^2 + 4 \cdot \ell_\infty \cdot L_v}, \mathcal{H}_g, \|\cdot\|_\infty)$$

such that the following inequality holds:

$$\left| \hat{\mathbb{E}}_{\hat{y}} \left[ R_{\hat{\mathcal{D}}} \left( r, g, h \right) \right] - \hat{\mathbb{E}}_{\hat{y}} \left[ R_{\hat{\mathcal{D}}} \left( r_l, g_u, h \right) \right] \right| \leq \frac{\epsilon}{4}$$

This completes the proof. $\qquad\qquad\square$

**Lemma B.15.** *Let $(r, g, h)$ be the detector-classifier triplet and $K$ be the number of pseudo base-to-new partitions. The following inequality holds given the definition of $v_\infty$ and $\ell_\infty$:*

$$\mathbb{P}\left[ \left| \mathbb{E}_{\hat{y}} \left[ R_{\hat{\mathcal{D}}} \left( r, g, h \right) \right] - \hat{\mathbb{E}}_{\hat{y}} \left[ R_{\hat{\mathcal{D}}} \left( r, g, h \right) \right] \right| \geq \frac{\epsilon}{2} \right] \leq 2 \cdot \exp\left( -\frac{\epsilon^2 \cdot K}{2 \cdot v_\infty^4 \cdot \ell_\infty^2} \right)$$

*Proof.* **Step 1:** We first need to demonstrate that the $\hat{\mathbb{E}}_{\hat{y}} \left[ R_{\hat{\mathcal{D}}} \left( r, g, h \right) \right]$ satisfies the bounded difference property defined in Def.1. Let $\hat{\mathcal{Y}}$ and $\hat{\mathcal{Y}}'$ be two independent base-to-new pseudo partition distribution where exactly one partition is different. The difference between $\hat{\mathcal{Y}}$ and $\hat{\mathcal{Y}}'$ could be caused by $i$-th pseudo partition process, denoted as $\hat{\mathcal{Y}}^{(i)}$ and $\hat{\mathcal{Y}}'^{(i)}$, respectively. Thus, we need to prove the following bound:

$$\sup_{s_b \in \mathcal{H}_g, r \in \mathcal{H}_r} \left| \hat{\mathbb{E}}_{\hat{y}} \left[ R_{\hat{\mathcal{D}}} \left( r, g, h \right) \right] - \hat{\mathbb{E}}_{\hat{y}'} \left[ R_{\hat{\mathcal{D}}} \left( r, g, h \right) \right] \right|$$

Taking a further step, we have:

$$\sup_{s_b \in \mathcal{H}_g, r \in \mathcal{H}_r} \left| \hat{\mathbb{E}}_{\hat{y}} \left[ R_{\hat{\mathcal{D}}} \left( r, g, h \right) \right] - \hat{\mathbb{E}}_{\hat{y}'} \left[ R_{\hat{\mathcal{D}}} \left( r, g, h \right) \right] \right|$$

$$\leq \sup_{s_b \in \mathcal{H}_g, r \in \mathcal{H}_r} \frac{1}{K} \sum_{k=1}^{K} \left| R_{\hat{\mathcal{D}}^{(k)}} \left( r, g, h \right) - R_{\hat{\mathcal{D}}'^{(k)}} \left( r, g, h \right) \right|$$

$$= \sup_{s_b \in \mathcal{H}_g, r \in \mathcal{H}_r} \frac{1}{K} \left| R_{\hat{\mathcal{D}}^{(i)}} \left( r, g, h \right) - R_{\hat{\mathcal{D}}'^{(i)}} \left( r, g, h \right) \right| \overset{(*)}{\leq} \frac{v_\infty^2 \cdot \ell_\infty}{K}$$

$(*)$ holds because $0 \leq \varphi_b \cdot \ell_{sq}(r, x_n, x_b) \cdot \varphi_n \leq v_\infty^2 \cdot \ell_\infty$. As a result, the expected loss also satisfies $R_{\hat{\mathcal{D}}^{(i)}} \leq v_\infty^2 \cdot \ell_\infty$. This leads us to conclude that $\hat{\mathbb{E}}_{\hat{y}}\left[R_{\hat{\mathcal{D}}}(r, g, h)\right]$ satisfies the bounded difference property.

**Step 2:** Then, according to the Lem.B.1, we have:

$$\mathbb{P}\left[\left|\mathbb{E}_{\hat{y}}\left[R_{\hat{\mathcal{D}}}(r, g, h)\right] - \hat{\mathbb{E}}_{\hat{y}}\left[R_{\hat{\mathcal{D}}}(r, g, h)\right]\right| \geq \frac{\epsilon}{2}\right] \leq 2 \cdot \exp\left(-\frac{\epsilon^2 \cdot K}{2 \cdot v_\infty^4 \cdot \ell_\infty^2}\right)$$

This completes the proof. $\qquad\square$

**Lemma B.16.** *Equipped with Lem.B.14 and Lem.B.15, for $\forall r \in \mathcal{H}_r, s_b \in \mathcal{H}_g$, the following inequation holds with high probability:*

$$\left|\mathbb{E}_{\hat{y}}\left[R_{\hat{\mathcal{D}}}(r, g, h)\right] - \hat{\mathbb{E}}_{\hat{y}}\left[R_{\hat{\mathcal{D}}}(r, g, h)\right]\right| \leq v_\infty^2 \cdot \ell_\infty \cdot \sqrt{\frac{2d}{K} \cdot \log\left(3 \cdot r \cdot K\right)}$$

*Proof.* First, we need to figure out the following probability:

$$\mathbb{P}\left[\sup_{s_b \in \mathcal{H}_g, r \in \mathcal{H}_r}\left|\mathbb{E}_{\hat{y}}\left[R_{\hat{\mathcal{D}}}(r, g, h)\right] - \hat{\mathbb{E}}_{\hat{y}}\left[R_{\hat{\mathcal{D}}}(r, g, h)\right]\right| \geq \epsilon\right]$$

$$\leq \mathbb{P}\left[\sup_{r \in \bigcup_{l=1}^{\mathcal{N}_r} \mathcal{B}_r(r_l, \epsilon_r), s_b \in \bigcup_{u=1}^{\mathcal{N}_g} \mathcal{B}_g(s_{b,u}, \epsilon_g)}\left|\mathbb{E}_{\hat{y}}\left[R_{\hat{\mathcal{D}}}(r, g, h)\right] - \hat{\mathbb{E}}_{\hat{y}}\left[R_{\hat{\mathcal{D}}}(r, g, h)\right]\right| \geq \epsilon\right]$$

$$\leq \sum_{u=1}^{\mathcal{N}_g}\sum_{l=1}^{\mathcal{N}_r} \mathbb{P}\left[\sup_{r \in \mathcal{B}(r_l, \epsilon_r), s_b \in \mathcal{B}(s_{b,u}, \epsilon_g)}\left|\mathbb{E}_{\hat{y}}\left[R_{\hat{\mathcal{D}}}(r, g, h)\right] - \hat{\mathbb{E}}_{\hat{y}}\left[R_{\hat{\mathcal{D}}}(r, g, h)\right]\right| \geq \epsilon\right]$$

$$\overset{Lem.B.14}{\leq} \sum_{u=1}^{\mathcal{N}_g}\sum_{l=1}^{\mathcal{N}_r} \mathbb{P}\left[\left|\mathbb{E}_{\hat{y}}\left[R_{\hat{\mathcal{D}}}(r_l, g_u, h)\right] - \hat{\mathbb{E}}_{\hat{y}}\left[R_{\hat{\mathcal{D}}}(r_l, g_u, h)\right]\right| \geq \frac{\epsilon}{2}\right]$$

$$\leq \mathcal{N}_g \mathcal{N}_r \mathbb{P}\left[\left|\mathbb{E}_{\hat{y}}\left[R_{\hat{\mathcal{D}}}(r_l, g_u, h)\right] - \hat{\mathbb{E}}_{\hat{y}}\left[R_{\hat{\mathcal{D}}}(r_l, g_u, h)\right]\right| \geq \frac{\epsilon}{2}\right]$$

$$\overset{Lem.B.15}{\leq} \mathcal{N}_g \mathcal{N}_r 2 \exp\left(-\frac{\epsilon^2 \cdot K}{2 \cdot v_\infty^4 \cdot \ell_\infty^2}\right)$$

According to Lem.B.14, we have

$$\mathcal{N}_g = \mathcal{N}(\epsilon_r = \frac{\epsilon}{8 \cdot L_\ell \cdot v_\infty^2 + 4 \cdot \ell_\infty \cdot L_v}, \mathcal{H}_r, \|\cdot\|_\infty)$$

$$\mathcal{N}_r = \mathcal{N}(\epsilon_g = \frac{\epsilon}{8 \cdot L_\ell \cdot v_\infty^2 + 4 \cdot \ell_\infty \cdot L_v}, \mathcal{H}_g, \|\cdot\|_\infty)$$

By further choosing

$$\epsilon = v_\infty^2 \cdot \ell_\infty \cdot \sqrt{\frac{2d}{K} \cdot \log\left(3 \cdot r \cdot K\right)}$$

we further have:

$$\mathbb{P}\left[\sup_{s_b \in \mathcal{H}_g, r \in \mathcal{H}_r}\left|\mathbb{E}_{\hat{y}}\left[R_{\hat{\mathcal{D}}}(r, g, h)\right] - \hat{\mathbb{E}}_{\hat{y}}\left[R_{\hat{\mathcal{D}}}(r, g, h)\right]\right| \geq v_\infty^2 \cdot \ell_\infty \cdot \sqrt{\frac{2d}{K} \cdot \log\left(3 \cdot r \cdot K\right)}\right] \leq 2 \cdot \left(\frac{\Gamma}{2d \log 3 \cdot r \cdot K}\right)^d$$

where $\Gamma$ is universal constant depending on $v_\infty, \ell_\infty, L_\ell, L_v, r$.

Therefore, we arrive at the conclusion that for $\forall s_b \in \mathcal{H}_g, r \in \mathcal{H}_r$, the following inequation holds with high probability

$$\left|\mathbb{E}_{\hat{y}}\left[R_{\hat{\mathcal{D}}}(r, g, h)\right] - \hat{\mathbb{E}}_{\hat{y}}\left[R_{\hat{\mathcal{D}}}(r, g, h)\right]\right| \leq v_\infty^2 \cdot \ell_\infty \cdot \sqrt{\frac{2d}{K} \cdot \log\left(3 \cdot r \cdot K\right)}$$

$\qquad\square$

**Lemma B.17.** *Assume that the gated weighting function $\varphi(\cdot)$ is $L_v$-lipschitz continuous and the loss function $\ell$ is $L_\ell$-Lipschitz continuous. Let $\epsilon$ be the generalization error between $\hat{R}_{\hat{\mathcal{S}}}(r, g, h)$ and $\mathbb{E}[\hat{R}_{\hat{\mathcal{S}}}(r, g, h)]$. Then by constructing $\sigma$-covering $\{r_1, r_2, \cdots, r_{\mathcal{N}_r}\}$ of $\mathcal{H}_r$ with $\epsilon_r$ and $\sigma$-covering $\{s_{b,1}, s_{b,2}, \cdots, s_{b,\mathcal{N}_g}\}$ of $\mathcal{H}_g$ with $\epsilon_g$, the following inequality holds:*

$$\mathbb{P}\left[\sup_{r \in \mathcal{B}(r_l, \epsilon_r), s_b \in \mathcal{B}(s_{b,u}, \epsilon_g)} \left|\hat{R}_{\hat{\mathcal{S}}}(r, g, h) - \mathbb{E}[\hat{R}_{\hat{\mathcal{S}}}(r, g, h)]\right| \leq \epsilon\right] \geq \mathbb{P}\left[\left|\hat{R}_{\hat{\mathcal{S}}}(r_\ell, g_u, h) - \mathbb{E}[\hat{R}_{\hat{\mathcal{S}}}(r_\ell, g_u, h)]\right| \leq \frac{\epsilon}{2}\right]$$

*Proof.* Firstly, some simplified notations applicable only to this proof are given as follows:

$$\sigma(f_g(x_b^i)) = \varphi_b^i, \quad \sigma(f_{g_u}(x_b^i)) = \tilde{\varphi}_b^i, \quad \sigma(f_h(x_n^j)) = \varphi_n^j$$
$$\ell_{sq}(r(x_b^i) - r(x_n^j)) = \ell_{i,j}, \quad \ell_{sq}(r_l(x_b^i) - r_l(x_n^j)) = \tilde{\ell}_{i,j}$$

Given $\epsilon_r$-covering $\{r_1, r_2, \cdots, r_n\}$ of $\mathcal{H}_r$ and $\epsilon_g$-covering $\{s_{b,1}, s_{b,2}, \cdots, s_{b,\mathcal{N}_g}\}$ of $\mathcal{H}_g$, to prove the above lemma, we turn to prove the following inequality holds:

$$\left|\left|\hat{R}_{\hat{\mathcal{S}}}(r, g, h) - \mathbb{E}[\hat{R}_{\hat{\mathcal{S}}}(r, g, h)]\right| - \left|\hat{R}_{\hat{\mathcal{S}}}(r_\ell, g_u, h) - \mathbb{E}[\hat{R}_{\hat{\mathcal{S}}}(r_\ell, g_u, h)]\right|\right|$$
$$\leq \left|\hat{R}_{\hat{\mathcal{S}}}(r, g, h) - \hat{R}_{\hat{\mathcal{S}}}(r_\ell, g_u, h)\right| + \left|\mathbb{E}[\hat{R}_{\hat{\mathcal{S}}}(r, g, h)] - \mathbb{E}[\hat{R}_{\hat{\mathcal{S}}}(r_\ell, g_u, h)]\right| \leq \frac{\epsilon}{2}$$

Similarly, we have:

$$\left|\hat{R}_{\hat{\mathcal{S}}}(r, g, h) - \hat{R}_{\hat{\mathcal{S}}}(r_\ell, g_u, h)\right| \leq \left|\frac{1}{\tilde{N}_b \cdot \tilde{N}_n} \sum_{i=1}^{\tilde{N}_b} \sum_{j=1}^{\tilde{N}_n} \left|\varphi_g^i \cdot \ell_{i,j} \cdot \varphi_n^j - \tilde{\varphi}_b^i \cdot \tilde{\ell}_{i,j} \cdot \varphi_n^j\right|\right|$$
$$\leq \max_{(x_b^i, x_n^j)} \left|\varphi_g^i \cdot \ell_{i,j} \cdot \varphi_n^j - \tilde{\varphi}_b^i \cdot \tilde{\ell}_{i,j} \cdot \varphi_n^j\right|$$
$$\overset{(*)}{\leq} \max_{(x_b^i, x_n^j)} \left\{\left|\varphi_g^i \cdot \ell_{i,j} \cdot \varphi_n^j - \varphi_g^i \cdot \tilde{\ell}_{i,j} \cdot \varphi_n^j\right| + \left|\varphi_g^i \cdot \tilde{\ell}_{i,j} \cdot \varphi_n^j - \tilde{\varphi}_b^i \cdot \tilde{\ell}_{i,j} \cdot \varphi_n^j\right|\right\}$$
$$\overset{(**)}{\leq} \max_{(x_b^i, x_n^j)} \left\{2 \cdot v_\infty^2 \cdot L_\ell \cdot \|r - r_\ell\|_\infty + \ell_\infty \cdot v_\infty \cdot L_v \cdot \|s_b - s_{b,u}\|_\infty\right\}$$
$$\leq 2 \cdot L_\ell \cdot v_\infty^2 \cdot \epsilon_r + \ell_\infty \cdot L_v \cdot \epsilon_g \leq \frac{\epsilon}{4}$$

where $\tilde{N}_b$ and $\tilde{N}_n$ are the number of pseudo base and new samples in $\hat{\mathcal{S}}$. $(*)$ follows the fact that $|x + y| \leq |x| + |y|$. $(**)$ follows the Lem.B.5.

By further choosing $\epsilon_r = \frac{\epsilon}{8 \cdot L_\ell \cdot v_\infty^2 + 4 \cdot \ell_\infty \cdot L_v}$ and $\epsilon_g = \frac{\epsilon}{8 \cdot L_\ell \cdot v_\infty^2 + 4 \cdot \ell_\infty \cdot L_v}$, we could construct the covering number $\mathcal{N}_r$ and $\mathcal{N}_g$ as follows:

$$\mathcal{N}_r = \mathcal{N}(\epsilon_r = \frac{\epsilon}{8 \cdot L_\ell \cdot v_\infty^2 + 4 \cdot \ell_\infty \cdot L_v}, \mathcal{H}_r, \rho_r), \rho_r = \|r_l - r\|_\infty$$
$$\mathcal{N}_g = \mathcal{N}(\epsilon_g = \frac{\epsilon}{8 \cdot L_\ell \cdot v_\infty^2 + 4 \cdot \ell_\infty \cdot L_v}, \mathcal{H}_g, \rho_g), \rho_g = \|s_{b,u} - s_b\|_\infty$$

such that the following inequality holds:

$$\left|\hat{R}_{\hat{\mathcal{S}}}(r, g, h) - \hat{R}_{\hat{\mathcal{S}}}(r_\ell, g_u, h)\right| \leq \frac{\epsilon}{4}$$

Thus, we have:

$$\left|\hat{R}_{\hat{\mathcal{S}}}(r, g, h) - \hat{R}_{\hat{\mathcal{S}}}(r_\ell, g_u, h)\right| + \left|\mathbb{E}[\hat{R}_{\hat{\mathcal{S}}}(r, g, h)] - \mathbb{E}[\hat{R}_{\hat{\mathcal{S}}}(r_\ell, g_u, h)]\right| \leq \frac{\epsilon}{2}$$

This completes the proof. $\square$

**Lemma B.18.** *Let $\hat{\mathcal{S}}$ and $\hat{\mathcal{S}}'$ be two independent datasets where exactly one instance is different. We conclude that $\hat{R}_{\hat{\mathcal{S}}}(r, g, h)$ satisfies the bounded difference property.*

*Proof.* The following simplified notations are used in this proof:

$$\sigma(f_g(x_b^i)) = \varphi_b^i, \quad \sigma(s_b(x_b)) = \varphi_b, \quad \sigma(s_b(\tilde{x}_b)) = \tilde{\varphi}_b$$
$$\sigma(f_h(x_n^j)) = \varphi_n^j, \quad \sigma(s_n(x_n)) = \varphi_n, \quad \sigma(s_n(\tilde{x}_n)) = \tilde{\varphi}_n$$
$$\ell_{sq}(r(x_b) - r(x_n^j)) = \ell(r, x_b, x_n^j), \quad \ell_{sq}(r(\tilde{x}_b) - r(x_n^j)) = \ell(r, \tilde{x}_b, x_n^j)$$
$$\ell_{sq}(r(x_b^i) - r(x_n)) = \ell(r, x_b^i, x_n), \quad \ell_{sq}(r(x_b^i) - r(\tilde{x}_n)) = \ell(r, x_b^i, \tilde{x}_n)$$

To prove this lemma, we need to seek the upper bound of

$$\sup_{s_b \in \mathcal{H}_g, r \in \mathcal{H}_r} \left| \hat{R}_{\hat{\mathcal{S}}'}(r, g, h) - \hat{R}_{\hat{\mathcal{S}}}(r, g, h) \right|$$

We first recall that

$$\hat{R}_{\hat{\mathcal{S}}}(r, g, h) = \mathop{\hat{\mathbb{E}}}_{\substack{(x_b, y_b) \sim \hat{\mathcal{S}}_b \\ (x_n, y_n) \sim \hat{\mathcal{S}}_n}} \varphi_b \cdot \ell_{sq}(r, x_n, x_b) \cdot \varphi_n = \frac{1}{\tilde{N}_b \cdot \tilde{N}_n} \sum_{i=1}^{\tilde{N}_b} \sum_{j=1}^{\tilde{N}_n} \varphi_b^i \cdot \ell_{sq}(r(x_b^i) - r(x_n^j)) \cdot \varphi_n^j$$

Thus, the difference between $\hat{\mathcal{S}}$ and $\hat{\mathcal{S}}'$ could be caused by either the **pseudo base domain sample** (i.e. $x_b$ and $\tilde{x}_b$) or the **pseudo new domain sample** (i.e. $x_n$ and $\tilde{x}_n$). Hence, we have the following two possible cases:

**Case 1:** Only a **base domain** sample is different. Since $x_b$ and $\tilde{x}_b$ are different in this case, we have:

$$\sup_{s_b \in \mathcal{H}_g, r \in \mathcal{H}_r} \left| \hat{R}_{\hat{\mathcal{S}}'}(r, g, h) - \hat{R}_{\hat{\mathcal{S}}}(r, g, h) \right|$$
$$= \sup_{s_b \in \mathcal{H}_g, r \in \mathcal{H}_r} \left| \frac{1}{\tilde{N}_b \tilde{N}_n} \sum_{j=1}^{\tilde{N}_n} \varphi_n^j \cdot \ell_{sq}(r(x_b) - r(x_n^j)) \cdot \varphi_b - \frac{1}{\tilde{N}_b \tilde{N}_n} \sum_{j=1}^{\tilde{N}_n} \varphi_n^j \cdot \ell_{sq}(r(\tilde{x}_b) - r(x_n^j)) \cdot \tilde{\varphi}_b \right|$$
$$\leq \frac{1}{\tilde{N}_b \tilde{N}_n} \sup_{s_b \in \mathcal{H}_g, r \in \mathcal{H}_r} \sum_{j=1}^{\tilde{N}_n} \left| \varphi_n^j \cdot \ell(r, x_b, x_n^j) \cdot \varphi_b - \varphi_n^j \cdot \ell(r, \tilde{x}_b, x_n^j) \cdot \tilde{\varphi}_b \right| \overset{(*)}{\leq} \frac{v_\infty^2 \cdot \ell_\infty}{\tilde{N}_b}$$

$(*)$ holds because $\varphi_n^j \cdot \ell(r, x_b, x_n^j) \cdot \varphi_b \in [0, v_\infty^2 \cdot \ell_\infty]$.

**Case 2:** Only a **new domain** sample is different. Since $x_n$ and $\tilde{x}_n$ are different in this case, similarly we have:

$$\sup_{s_b \in \mathcal{H}_g, r \in \mathcal{H}_r, h \in \mathcal{H}_h} \left| \hat{R}_{\hat{\mathcal{S}}'}(r, g, h) - \hat{R}_{\hat{\mathcal{S}}}(r, g, h) \right| \leq \frac{v_\infty^2 \cdot \ell_\infty}{\tilde{N}_n}$$

Finally, taking two cases into account, we can conclude that $\hat{R}_S(r, g, h)$ is satisfied with the bounded difference property. □

**Lemma B.19.** *Let $\tilde{N}_b$ and $\tilde{N}_n$ be the number of pseudo base and new samples in dataset $\hat{\mathcal{S}}$. Equipped with the above Lem.B.1 and Lem.B.18, the following inequality holds given the definition of $v_\infty$ and $\ell_\infty$:*

$$\mathbb{P}\left[ \left| \hat{R}_{\hat{\mathcal{S}}}(r_l, g_u, h) - \mathbb{E}[\hat{R}_{\hat{\mathcal{S}}}(r_l, g_u, h)] \right| \geq \frac{\epsilon}{2} \right] \leq 2 \exp\left( -\frac{\epsilon^2 \cdot \tilde{N}}{2} \right)$$

*where*

$$\tilde{N} = v_\infty^{-4} \cdot \ell_\infty^{-2} \cdot \left( \frac{1}{\tilde{N}_b} + \frac{1}{\tilde{N}_n} \right)^{-1}$$

*Proof.* The proof could be easily achieved by applying Lem.B.1 on top of Lem.B.18 as:

$$\mathbb{P}\left[\left|\hat{R}_{\hat{\mathcal{S}}}(r_l, g_u, h) - \mathbb{E}[\hat{R}_{\hat{\mathcal{S}}}(r_l, g_u, h)]\right| \geq \frac{\epsilon}{2}\right] \leq 2 \cdot \exp\left(-\frac{\epsilon^2}{2 \cdot \sum_{t=1}^{n} c_t^2}\right)$$

$$\leq 2 \cdot \exp\left(-\frac{\epsilon^2}{2 \cdot \left(\frac{v_{\infty}^4 \cdot \ell_{\infty}^2}{\tilde{N}_b} + \frac{v_{\infty}^4 \cdot \ell_{\infty}^2}{\tilde{N}_n}\right)}\right)$$

$\square$

**Lemma B.20.** *Let $\mathbb{E}[\hat{R}_{\hat{\mathcal{S}}}(r, g, h)]$ be the population risk of $\hat{R}_{\hat{\mathcal{S}}}(r, g, h)$. Then, equipped with Lem.B.17 and Lem.B.18, for $\forall r \in \mathcal{H}_r, \boldsymbol{s}_b \in \mathcal{H}_g$, the following inequation holds with high probability:*

$$\left|\hat{R}_{\hat{\mathcal{S}}}(r, g, h) - \mathbb{E}[\hat{R}_{\hat{\mathcal{S}}}(r, g, h)]\right| \leq \sqrt{\frac{2d \log 3 \cdot r \cdot \tilde{N}}{\tilde{N}}}$$

*Proof.* First, we need to figure out the following probability:

$$\mathbb{P}\left[\sup_{\boldsymbol{s}_b \in \mathcal{H}_g, r \in \mathcal{H}_r} \left|\hat{R}_{\hat{\mathcal{S}}}(r, g, h) - \mathbb{E}[\hat{R}_{\hat{\mathcal{S}}}(r, g, h)]\right| \geq \epsilon\right]$$

$$\leq \mathbb{P}\left[\sup_{r \in \bigcup_{l=1}^{\mathcal{N}_r} \mathcal{B}_r(r_l, \epsilon_r), \boldsymbol{s}_b \in \bigcup_{u=1}^{\mathcal{N}_g} \mathcal{B}_g(\boldsymbol{s}_{b,u}, \epsilon_g)} \left|\hat{R}_{\hat{\mathcal{S}}}(r, g, h) - \mathbb{E}[\hat{R}_{\hat{\mathcal{S}}}(r, g, h)]\right| \geq \epsilon\right]$$

$$\leq \sum_{u=1}^{\mathcal{N}_g} \sum_{l=1}^{\mathcal{N}_r} \mathbb{P}\left[\sup_{r \in \mathcal{B}(r_l, \epsilon_r), \boldsymbol{s}_b \in \mathcal{B}(\boldsymbol{s}_{b,u}, \epsilon_g)} \left|\hat{R}_{\hat{\mathcal{S}}}(r, g, h) - \mathbb{E}[\hat{R}_{\hat{\mathcal{S}}}(r, g, h)]\right| \geq \epsilon\right]$$

$$\overset{Lem.B.17}{\leq} \sum_{u=1}^{\mathcal{N}_g} \sum_{l=1}^{\mathcal{N}_r} \mathbb{P}\left[\left|\hat{R}_{\hat{\mathcal{S}}}(r_\ell, g_u, h) - \mathbb{E}[\hat{R}_{\hat{\mathcal{S}}}(r_\ell, g_u, h)]\right| \geq \frac{\epsilon}{2}\right]$$

$$\leq \mathcal{N}_g \mathcal{N}_r \mathbb{P}\left[\left|\hat{R}_{\hat{\mathcal{S}}}(r_\ell, g_u, h) - \mathbb{E}[\hat{R}_{\hat{\mathcal{S}}}(r_\ell, g_u, h)]\right| \geq \frac{\epsilon}{2}\right]$$

$$\overset{Lem.B.18}{\leq} \mathcal{N}_g \mathcal{N}_r 2 \exp\left(-\frac{\epsilon^2 \cdot \tilde{N}}{2}\right)$$

where $\mathcal{N}_r, \mathcal{N}_g$ are the covering numbers of the hypothesis space $\mathcal{H}_r, \mathcal{H}_g$, respectively.

$$\mathcal{N}_r = \mathcal{N}(\epsilon_r = \frac{\epsilon}{8 \cdot L_\ell \cdot v_\infty^2 + 4 \cdot \ell_\infty \cdot L_v}, \mathcal{H}_r, \rho_r), \rho_r = \|r_l - r\|_\infty$$

$$\mathcal{N}_g = \mathcal{N}(\epsilon_g = \frac{\epsilon}{8 \cdot L_\ell \cdot v_\infty^2 + 4 \cdot \ell_\infty \cdot L_v}, \mathcal{H}_g, \rho_g), \rho_g = \|\boldsymbol{s}_{b_u} - \boldsymbol{s}_b\|_\infty$$

By futher choosing,

$$\epsilon = \sqrt{\frac{2 \cdot d \cdot \log\left(3 \cdot r \cdot \tilde{N}\right)}{\tilde{N}}}$$

we have:

$$\mathbb{P}\left[\left|\hat{R}_{\hat{\mathcal{S}}}(r, g, h) - \mathbb{E}[\hat{R}_{\hat{\mathcal{S}}}(r, g, h)]\right| \geq \sqrt{\frac{2d \log 3 \cdot r \cdot \tilde{N}}{\tilde{N}}}\right] \leq 2 \cdot \left(\frac{\Gamma}{2d \log 3 \cdot r \cdot \tilde{N}}\right)^d$$

Similarly, we conclude that the following inequality holds with high probability,

$$\left| \hat{R}_{\hat{S}}(r,g,h) - \mathbb{E}[\hat{R}_{\hat{S}}(r,g,h)] \right| \leq \sqrt{\frac{2d\log 3 \cdot r \cdot \tilde{N}}{\tilde{N}}}, \quad \forall r \in \mathcal{H}_r, \boldsymbol{s}_b \in \mathcal{H}_g$$

$\square$

### B.4. Formal Proof

We present an upper bound of the expected risk $\mathcal{R}(r,g,h)$ in $(OP_0)$. Note that our subsequent analysis is based on a standard assumption that the base domain logit score function $\boldsymbol{s}_b$ and the detection function $r$ are chosen from hypotheses class $\mathcal{H}_g$ and $\mathcal{H}_r$ respectively. These hypothesis classes are the specific types of frozen CLIP with learnable prompts.

**Restate of Theorem 5.2.** Let $\mathcal{Y}_b$ be the base label space and $\mathcal{Y}$ be the overall label space, where the total number of classes is $C$. Suppose $\hat{\mathcal{Y}}$ represents the pseudo base-to-new partition distribution, and let $\mathcal{E}$ and $\mathcal{E}'$ denote the expectation distribution over $\mathcal{Y}$ and $\hat{\mathcal{Y}}$. Furthermore, consider a training dataset $\mathcal{S}_b = \{(\boldsymbol{x}_b^i, y_b^i)\}_{i=1}^{N_b}$ comprising *i.i.d.* samples drawn from the data distribution $\hat{\mathcal{D}}$. Define $\mathcal{R}(r,g,h)$ as the population risk of $\hat{\mathcal{R}}(r,g,h)$ in $OP_0$. For the loss function, let $\ell_\infty = \sup_{(x_b, x_n)} |\ell_{sq}(r, x_b, x_n)|$ bounds the squared loss, $\ell_{c,\infty} = \sup_{(x,y)} \ell_{ce}(\boldsymbol{s}_b(x), y)$ bounds the cross-entropy loss and $v_\infty = \sup_x \left| \sigma(\boldsymbol{s}_b^{(y)}(x)) \right|$ bounds the sigmoid function. Assume that $\mathcal{N}(\epsilon; \mathcal{H}, \rho) \leq \left(\frac{3r}{\epsilon}\right)^d$. Then, for all $r \in \mathcal{H}_r, \boldsymbol{s}_b \in \mathcal{H}_g$, with high probability, the following inequality holds:

$$\mathcal{R}(h,g,r) \leq \hat{\mathcal{R}}'(h,g,r) + \mathbb{E}_{\mathcal{D}_n}\left[\ell_{ce}(\boldsymbol{s}_n(x_n), y_n)\right] + \frac{v_\infty^2 \cdot \ell_\infty \cdot \|\mathcal{E}(\mathbb{P}) - \mathcal{E}'(\mathbb{P})\|_\infty}{C!}$$
$$+ v_\infty^2 \cdot \ell_\infty \cdot \sqrt{\frac{2d}{K} \cdot \log\left(3 \cdot r \cdot K\right)} + 2 \cdot v_\infty^2 \cdot \ell_\infty \cdot \sqrt{\frac{2d\log 3 \cdot r \cdot N_b}{N_b}}$$

*Proof.* We first decompose the overall excess risk into the following three parts:

$$\Delta = \sup_{\boldsymbol{s}_b \in \mathcal{H}_g, r \in \mathcal{H}_r} \left[ \mathcal{R}(r,g,h) - \hat{\mathcal{R}}'(r,g,h) \right]$$
$$\leq \underbrace{\sup_{\boldsymbol{s}_b \in \mathcal{H}_g, r \in \mathcal{H}_r} \left[ \mathbb{E}_{\mathcal{E}}\left[R_{\mathcal{D}}(r,g,h)\right] - \hat{\mathbb{E}}_{\hat{\mathcal{E}}}\left[\hat{R}_{\hat{S}}(r,g,h)\right] \right]}_{\Delta_r} + \underbrace{\sup_{\boldsymbol{s}_b \in \mathcal{H}_g} \left[ \mathbb{E}_{\mathcal{D}_b}\left[\ell_{ce}(\boldsymbol{s}_b(x_b), y_b)\right] - \hat{\mathbb{E}}_{\mathcal{S}_b}\left[\ell_{ce}(\boldsymbol{s}_b(x_b), y_b)\right] \right]}_{\Delta_g}$$
$$+ \underbrace{\mathbb{E}_{\mathcal{D}_n}\left[\ell_{ce}(\boldsymbol{s}_n(x_n), y_n)\right]}_{\Delta_h}$$

According to our analysis in the proof outline, analyzing $\Delta_h$ and bounding $\Delta_g$ are relatively straightforward. We will first analyze $\Delta_h$ and $\Delta_g$ and then provide a detailed analysis of $\Delta_r$.

**As for $\Delta_h$.** Without prior knowledge of the new domain, bounding the expected error $\Delta_h$ remains challenging. In practice, we can reduce this term by carefully designing hand-crafted prompts or using more powerful foundation models.

**As for $\Delta_g$.** Accoring to the Lem.B.12, the loss function $\ell_{ce}$ is lipschitz continuous. By applying Lem.B.13, the following inequality holds with high probability:

$$\sup_{\boldsymbol{s}_b \in \mathcal{H}_g} \left[ \mathbb{E}_{\mathcal{D}_b}\left[\ell_{ce}(\boldsymbol{s}_b(x_b), y_b)\right] - \hat{\mathbb{E}}_{\mathcal{S}_b}\left[\ell_{ce}(\boldsymbol{s}_b(x_b), y_b)\right] \right] \leq \ell_{c,\infty} \cdot \sqrt{\frac{2d}{N_b} \cdot \log\left(3 \cdot r \cdot N_b\right)}$$

**As for $\Delta_r$.** Directly bounding $\Delta_r$ is difficult, so we decompose the $\Delta_r$ as follows:

$$
\begin{aligned}
\Delta_r &= \sup_{\boldsymbol{s}_b \in \mathcal{H}_g, r \in \mathcal{H}_r} \left[ \mathbb{E}_{\mathcal{Y}}[R_{\mathcal{D}}] - \hat{\mathbb{E}}_{\hat{y}}\left[\hat{R}_{\hat{\mathcal{S}}}\right] \right] \\
&= \sup_{\boldsymbol{s}_b \in \mathcal{H}_g, r \in \mathcal{H}_r} \left[ \mathbb{E}_{\mathcal{Y}}[R_{\mathcal{D}}] - \mathbb{E}_{\hat{y}}\left[R_{\hat{\mathcal{D}}}\right] + \mathbb{E}_{\hat{y}}\left[R_{\hat{\mathcal{D}}}\right] - \hat{\mathbb{E}}_{\hat{y}}\left[R_{\hat{\mathcal{D}}}\right] + \hat{\mathbb{E}}_{\hat{y}}\left[R_{\hat{\mathcal{D}}}\right] - \hat{\mathbb{E}}_{\hat{y}}\left[\hat{R}_{\hat{\mathcal{S}}}\right] \right] \\
&\leq \underbrace{\sup_{\boldsymbol{s}_b \in \mathcal{H}_g, r \in \mathcal{H}_r} \left[ \mathbb{E}_{\mathcal{Y}}[R_{\mathcal{D}}] - \mathbb{E}_{\hat{y}}\left[R_{\hat{\mathcal{D}}}\right] \right]}_{(a)} + \underbrace{\sup_{\boldsymbol{s}_b \in \mathcal{H}_g, r \in \mathcal{H}_r} \left[ \mathbb{E}_{\hat{y}}\left[R_{\hat{\mathcal{D}}}\right] - \hat{\mathbb{E}}_{\hat{y}}\left[R_{\hat{\mathcal{D}}}\right] \right]}_{(b)} + \underbrace{\sup_{\boldsymbol{s}_b \in \mathcal{H}_g, r \in \mathcal{H}_r} \left[ \hat{\mathbb{E}}_{\hat{y}}\left[R_{\hat{\mathcal{D}}}\right] - \hat{\mathbb{E}}_{\hat{y}}\left[\hat{R}_{\hat{\mathcal{S}}}\right] \right]}_{(c)}
\end{aligned}
$$

**As for (a)** Since $\varphi_b \cdot \ell_{sq}(r, x_n, x_b) \cdot \varphi_n$ is bounded by $[0, v_\infty^2 \cdot \ell_\infty]$, we have

$$
\sup_{\boldsymbol{s}_b \in \mathcal{H}_g, r \in \mathcal{H}_r} \left[ \mathbb{E}_{\mathcal{Y}}\left[R_{\mathcal{D}|\mathcal{Y}}\right] - \mathbb{E}_{\hat{y}}\left[R_{\hat{\mathcal{D}}}\right] \right] \leq v_\infty^2 \cdot \ell_\infty \int_{\mathbb{S}_{c-1}} |\mathcal{E}(\mathbb{P}) - \mathcal{E}'(\mathbb{P})| \, d\mathbb{P}
$$

$$
\overset{Lem.B.9}{\leq} \frac{v_\infty^2 \cdot \ell_\infty \cdot ||\mathcal{E}(\mathbb{P}) - \mathcal{E}'(\mathbb{P})||_\infty}{C!}
$$

where $\mathbb{S}_{c-1}$ is the probabilistic simplex on $\mathbb{R}^C$, $\mathbb{P}$ is a $C$-dimensional probability allocation sampled either from $\mathcal{E}$ or $\mathcal{E}'$.

**As for (b)**, by applying Lem.B.16, the following inequality holds with high probability:

$$
\sup_{\boldsymbol{s}_b \in \mathcal{H}_g, r \in \mathcal{H}_r} \left| \mathbb{E}_{\hat{y}}\left[R_{\hat{\mathcal{D}}}(r, g, h)\right] - \hat{\mathbb{E}}_{\hat{y}}\left[R_{\hat{\mathcal{D}}}(r, g, h)\right] \right| \leq v_\infty^2 \cdot \ell_\infty \cdot \sqrt{\frac{2d}{K} \cdot \log(3 \cdot r \cdot K)}
$$

**As for (c)**, we have:

$$
\sup_{\boldsymbol{s}_b \in \mathcal{H}_g, r \in \mathcal{H}_r} \left[ \hat{\mathbb{E}}_{\hat{y}}\left[R_{\hat{\mathcal{D}}}\right] - \hat{\mathbb{E}}_{\hat{y}}\left[\hat{R}_{\hat{\mathcal{S}}}\right] \right] = \sup_{\boldsymbol{s}_b \in \mathcal{H}_g, r \in \mathcal{H}_r} \left[ \frac{1}{K} \sum_{k=1}^{K} R_{\hat{\mathcal{D}}^{(k)}} - \frac{1}{K} \sum_{k=1}^{K} \hat{R}_{\hat{\mathcal{S}}^{(k)}} \right]
$$

For on fixed $\hat{\mathcal{Y}}^{(k)}$, according to the Lem.B.20, the following inequality holds with high probability:

$$
\sup_{\boldsymbol{s}_b \in \mathcal{H}_g, r \in \mathcal{H}_r} \left| \hat{R}_{\hat{\mathcal{S}}^{(k)}}(r, g, h) - \mathbb{E}[\hat{R}_{\hat{\mathcal{S}}^{(k)}}(r, g, h)] \right| \leq \sqrt{\frac{2d \log 3 \cdot r \cdot \tilde{N}}{\tilde{N}}}
$$

where

$$
\tilde{N} = v_\infty^{-4} \cdot \ell_\infty^{-2} \cdot \left( \frac{1}{\tilde{N}_b^{(k)}} + \frac{1}{\tilde{N}_n^{(k)}} \right)^{-1}
$$

Due to $\tilde{N}_n^{(k)} + \tilde{N}_b^{(k)} = N_b$, we have:

$$
\tilde{N} \geq \tilde{N}_{min} = \frac{N_b}{4 \cdot v_\infty^4 \cdot \ell_\infty^2}
$$

Thus, for all $\hat{\mathcal{Y}}^{(k)}, k \in [1, K]$, with high probability, we further have:

$$
\sup_{\boldsymbol{s}_b \in \mathcal{H}_g, r \in \mathcal{H}_r} \left| \hat{R}_{\hat{\mathcal{S}}^{(k)}}(r, g, h) - \mathbb{E}[\hat{R}_{\hat{\mathcal{S}}^{(k)}}(r, g, h)] \right| \leq \sqrt{\frac{2d \log 3 \cdot r \cdot \tilde{N}_{min}}{\tilde{N}_{min}}} \leq 2 \cdot v_\infty^2 \cdot \ell_\infty \cdot \sqrt{\frac{2d \log 3 \cdot r \cdot N_b}{N_b}}
$$

Finally, by combining the bound for $\Delta_r$, $\Delta_g$ and $\Delta_h$, we reach the conclusion of this theorem. $\qquad \square$

# C. Additional Experimental Setup

## C.1. Task Description

We perform empirical evaluations on **Open-world recognition**, **Open-world domain generalization** and **Open-world cross-dataset generalization** tasks:

- **Open-world recognition task.** This setting involves dividing the class space of each dataset equally, with 50% of the classes designated as base classes and the remaining 50% as new classes, following (Zhou et al., 2022b). In the imbalance setting, we resample the test sets constructed with different base/new sample ratios. The model is trained on the base classes and evaluated on the **mixture** of base and new classes.

- **Open-world domain generalization task.** The model is trained only on the base classes of ImageNet dataset and evaluated on the **all** classes of ImageNet variants datasets, each with additional different types of domain change.

- **Open-world cross-dataset generalization task.** We train our prompt exclusively on the base domain of ImageNet and subsequently evaluate model performance on a combined test set containing both the original ImageNet-500 classes and new categories from seven external datasets: FGVC-Aircraft, Caltech-101, Stanford-Cars, DTD, EuroSAT, SUN397, and UCF101. To ensure a fair evaluation of open-world generalization, we meticulously remove any categories from these external datasets that overlapped with the ImageNet-500 class space before evaluation.

## C.2. Datasets

For open-world recognition task, we follow prior work (Zhou et al., 2022b;a; 2024) and conduct experiments on 11 benchmark datasets that span a wide range of image recognition tasks. These datasets can be categorized into five main types: 1) Generic object classification, which includes **ImageNet** (Deng et al., 2009) and **Caltech101** (Fei-Fei et al., 2004); 2) Fine-grained classification, encompassing **OxfordPets** (Parkhi et al., 2012), **StanfordCars** (Krause et al., 2013), **Flowers102** (Nilsback & Zisserman, 2008), **Food101** (Bossard et al., 2014), and **FGVCAircraft** (Maji et al., 2013); 3) Scene recognition, represented by **SUN397** (Xiao et al., 2010); 4) Action recognition, with **UCF101** (Soomro et al., 2012); and 5) Specialized domains, such as texture classification with **DTD** (Cimpoi et al., 2014) and satellite image classification with **EuroSAT** (Helber et al., 2019). For the open-world domain generalization task, we use ImageNet (Deng et al., 2009) as the source domain and evaluate on its four variants: **ImageNetV2** (Recht et al., 2019), **ImageNet-Sketch** (Wang et al., 2019), **ImageNet-A** (Hendrycks et al., 2021b), and **ImageNet-R** (Hendrycks et al., 2021a). The details of each dataset are shown in the following:

- **ImageNet**. ImageNet, a widely recognized generic object classification dataset, comprises approximately 1.28 million training images and 50,000 test images across 1,000 object classes. Sourced from the web and organized using the WordNet hierarchy, it has become a benchmark for evaluating object recognition models.

- **Caltech101**. The Caltech101 dataset, designed for general object classification, includes 101 object categories and a background class, featuring around 7,650 training and 3,300 test images. The images, collected at Caltech, present significant variation in scale, orientation, and lighting conditions.

- **OxfordPets**. Focusing on fine-grained pet classification, OxfordPets contains 37 pet breed categories with nearly equal numbers of training (3,680) and test (3,669) images. The dataset provides not only breed annotations but also pixel-level segmentation masks, making it versatile for classification and segmentation tasks.

- **StanfordCars**. StanfordCars targets fine-grained car model recognition with 196 classes distinguished by make, model, and year. It offers 8,144 training images and 8,041 test images, sourced from various car-related platforms, capturing diverse vehicle angles and environments.

- **Flowers102**. Flowers102 is a dataset of 102 flower species, designed for fine-grained classification tasks. With 6,149 training and 1,020 test images, it presents visually intricate patterns, and challenging models to distinguish among visually similar flower categories.

- **Food101**. Food101 is tailored for fine-grained food classification, comprising 101 categories of dishes. The dataset includes 75,750 training images and 25,250 test images, offering challenges in recognizing overlapping ingredients and presentation styles. Images were curated from a range of food-related sources.

- **FGVCAircraft**. FGVCAircraft specializes in fine-grained classification of aircraft, offering 100 categories that distinguish between models and manufacturers. It contains 6,667 training images and 3,333 test images, sourced from aviation databases, with a focus on subtle visual differences in design.

- **SUN397**. SUN397 is a comprehensive scene recognition dataset that encompasses 397 categories, covering diverse environments such as natural landscapes, indoor spaces, and urban areas. It includes approximately 50,000 training and 50,000 test images, offering a rich resource for scene understanding.

- **UCF101**. UCF101 is a video dataset designed for action recognition tasks, featuring 101 action classes ranging from sports to daily activities. The dataset contains about 9,500 training clips and 3,700 test clips, collected from YouTube, with a variety of dynamic scenarios.

- **DTD(Describable Textures Dataset)** The DTD dataset focuses on texture classification, featuring 47 texture categories described using human-interpretable attributes. It offers 3,760 training and 1,880 test images, providing a unique challenge in recognizing visually distinctive patterns from natural and artificial sources.

- **EuroSAT**. EuroSAT is a satellite image dataset for land-use and land-cover classification, containing 10 classes such as agricultural areas, forests, and urban regions. Derived from Sentinel-2 satellite imagery, it includes 21,600 training and 5,400 test images, offering rich spatial and spectral diversity.

- **ImageNet-A** ImageNet-A is a curated subset of ImageNet consisting of images that exhibit challenging, adversarial characteristics. These images are specifically selected to challenge existing models, often including unusual or hard-to-classify variations of the objects from ImageNet's standard classes.

- **ImageNet-R** ImageNet-R is a variant of ImageNet that features images with artistic and stylized renditions of the original objects. These images have been altered to reflect different artistic interpretations, such as paintings, sketches, or computer-generated images. This dataset evaluates how well models generalize across diverse artistic representations of familiar objects.

- **ImageNetV2** ImageNetV2 is a revised version of the original ImageNet dataset, designed to better represent real-world data distribution. It has slight variations in image selection and distribution. It serves as an alternative benchmark to assess the robustness of object recognition models under different data variations.

- **ImageNet-Sketch** ImageNet-Sketch is a variant of ImageNet, created by converting the original images into sketch-like representations. It consists of $50,000$ images, spanning the same $1,000$ classes as the original ImageNet, but each image has been hand-drawn to emphasize the object's outline and basic features. This dataset challenges models to recognize objects in a more abstract form, testing their generalization ability across different visual representations.

We also construct 10 domain imbalance of $1 : 5$ datasets by randomly sampling DTD, Food101, Flowers102, OxfordPets, and SUN397. Forward (Fwd) means base domain number is 5 times the new domain number, while Backward (Bwd) means the opposite.

### C.3. Competitors

To demonstrate the effectiveness of our proposed method, we compare our method with 10 competitive competitors: $a$) Baseline, Zero-shot CLIP (Radford et al., 2021); $b$) HM-oriented methods, including CoOp (Zhou et al., 2022b), Maple (Khattak et al., 2023a), KgCoOp (Yao et al., 2023), PromptSRC (Khattak et al., 2023b), DePT-Kg (Zhang et al., 2024) and TCP (Yao et al., 2024); $c$) OOD-oriented methods, including LoCoOp (Miyai et al., 2023) and Gallop (Lafon et al., 2024); $d$) OverallAcc-oriented algorithms, DeCoOp (Zhou et al., 2024). The details of each competitor are provided in the following:

- **CoOp** proposes learnable continuous prompts to replace manual text templates in CLIP-like vision-language models. Through gradient-based optimization of task-specific textual context, it achieves strong few-shot performance but tends to overfit seen classes, compromising generalization on unseen categories.

- **Maple** couples vision-language prompts hierarchically, thus it mitigates the limitations of single-branch adaptation and improves generalization to unseen classes.

- **KgCoOp** addresses catastrophic forgetting of general knowledge in prompt tuning by minimizing the discrepancy between learned prompts and fixed hand-crafted prompts. It enforces consistency via a regularization loss, balancing base domain adaptation with CLIP's original zero-shot capabilities for improved classification performance on new domain.

- **PromptSRC** introduces a three-pronged regularization framework: (a) mutual agreement with frozen CLIP features, (b) self-ensemble of prompts across training phases, and (c) textual diversity augmentation. This approach reduces overfitting while preserving the model's generalization on both base and new classification tasks.

- **DePT** decouples prompt optimization into distinct components, potentially separating base and new domain classification. We incorporate this method on top of KgCoOp to further improve the generalization performance, denoted as DePT-Kg.

- **TCP** emphasizes class-aware prompt design using textual semantics, aligning prompts with fine-grained class contexts.

- **LoCoOp** leverages CLIP's local features (e.g., background regions) as pseudo-OOD samples during training, and it optimizes text embeddings to separate in-distribution (ID) classes from OOD data, reducing nuisance signals in ID embeddings. However, this method sacrifices classification performance on base and new domain to some extent.

- **Gallop** leverages both global and local visual representations. The key features of GalLoP are the strong discriminability of its local representations and its capacity to produce diverse predictions from both local and global prompts. This method achieves a better trade-off between domain classification and OOD detection.

- **DeCoOp** firstly focuses on open-world prompt tuning, where models must classify mixed base/new classes without prior knowledge. It integrates OOD detection into prompt tuning via the DePT framework, using new-class detectors and sub-classifiers to enhance base/new class discriminability.

To ensure a fair comparison, we use the same backbone architecture (ViT-B/16) for all competitors. We also adopt the same training strategy and hyperparameters according to their open-source code. The number of parameters for each method is shown in Tab.7.

*Table 7.* The number of parameters for each method.

| Method | CLIP | CoOp | Maple | PromptSRC | LoCoOp | KgCoOp | DePT-Kg | Gallop | DeCoOp | TCP | **Ours** |
|--------|------|------|-------|-----------|--------|--------|---------|--------|--------|------|----------|
| param | 0 | 8192 | 3555072 | 46080 | 8192 | 8192 | 292914 | 606208 | 30720 | 331904 | 26624 |

For all competitors, the detection score $r$ is derived from the maximum probability over the base domain. In this task, the new domain class names are known during testing. Compared to out-of-distribution (OOD) detection—which operates without prior knowledge of new classes—the base-to-new detection task is less challenging.

---

**Algorithm 1:** Base-to-new detection score $r$ calculation

**Input:** image_features, text_features, Number of base classes $C_b$
**Output:** Base-to-new detection score $r \in [0, 1]$
logits $\leftarrow$ image_features $\cdot$ text_features$^T$
prob $\leftarrow$ Softmax(logits, dim = 1)
$r \leftarrow \max\limits_{1 \leq j \leq C_b} \text{prob}[:, j]$
**return** $r$

---

### C.4. Implementation Details

All models are implemented using PyTorch (Paszke et al., 2017). The length of the classifier prompt is set to 4. To ensure the parameter size of our method is comparable to recent state-of-the-art methods that incorporate additional structures or deep prompts, the detector prompt's length is set to 16, consistent with DeCoOp (Zhou et al., 2024). Results are reported as the average over 5 runs with different random seeds $1, 2, 3, 4, 5$. We mention that we adopt a fixed hand-crafted prompt for the new domain classifier. To be specific, we use the prompt ensemble:

```
"itap_of_a_{}.  a_bad_photo_of_the_{}.  a_origami_{}.  a_photo_of_the_large_{}.
a_{}_in_a_video_game.  art_of_the_{}.  a_photo_of_the_small_{}"
```

in all datasets except for some fine-grained datasets. For the challenging fine-grained dataset, we use the prompt:

- FGVCAircraft: `"a photo of [CLASS], a type of aircraft."`

- Food101: `"a photo of [CLASS], a type of food"`

- Flowers102: `"a photo of [CLASS], a type of flower"`

to ensure the prompt is consistent with the dataset's characteristics.

### C.5. Efficient Calculation of OpenworldAUC

During the test phase, all samples containing base and new samples are available. To calculate the OpenworldAUC efficiently, we first mask each sample that has been misclassified and then calculate the AUROC on the correctly-classified samples through pairwise instance comparisons. To be specific, there are the following two steps.

**Step 1: Mask misclassified samples.**

$$\tilde{r}(x_n) = \begin{cases} \epsilon + \max_{x_b \in \mathcal{S}'_b} r(x_b) & \text{if } h(x_n) \neq y_n \\ r(x_n) & \text{if } h(x_n) = y_n \end{cases}$$

$$\tilde{r}(x_b) = \begin{cases} \min_{x_n \in \mathcal{S}'_n} r(x_n) - \epsilon & \text{if } g(x_b) \neq y_b \\ r(x_b) & \text{if } g(x_b) = y_b \end{cases}$$

where $\mathcal{S}'_b$ and $\mathcal{S}'_n$ are the base and new domain test datasets.

**Step 2: AUROC Calculation.**

$$\text{OpenworldAUC}(r, g, h) = \frac{1}{N'_b \cdot N'_n} \sum_{\substack{(x_b, y_b) \in \mathcal{S}'_b \\ (x_n, y_n) \in \mathcal{S}'_n}} \left[ \mathbf{1}[y_b = g(\boldsymbol{x}_b)] \cdot \mathbf{1}[r(\boldsymbol{x}_b) > r(\boldsymbol{x}_n)] \cdot \mathbf{1}[y_n = h(\boldsymbol{x}_n)] \right]$$

$$= \frac{1}{N'_b \cdot N'_n} \sum_{\substack{(x_b, y_b) \in \mathcal{S}'_b \\ (x_n, y_n) \in \mathcal{S}'_n}} \left[ \mathbf{1}[\tilde{r}(\boldsymbol{x}_b) > \tilde{r}(\boldsymbol{x}_n)] \right] = \text{AUROC}(\tilde{r})$$

## D. Additional Experimental Results

### D.1. Additional Results for Openworld Recognition Task

In this section, we present full results of open-world recognition task with standard deviation and also show stage-wise metrics (say AUROC and HM) in the Tab.8, Tab.9 and Tab.10. Fig.7 provides a clearer visualization, showing that HM-oriented methods are located on the right side of the image, implying a higher HM metric values. However, these methods still have limitations in base-to-new detection performance, which is consistent with the shortcoming of HM metric **(P1)**. And the OOD-oriented method such as LoCoOp may is located on the left upper of the image, meaning a higher AUROC metric but sacrifices the classification performance to some extent. This limitation is consistent with the AUROC metric **(P2)**. Compared with these methods, our OpenworldAUC-oriented optimization method can achieve a better trade-off between the AUROC and HM, which further demonstrate the comprehensive of OpenworldAUC metric and the efficiency of our optimization method. This is consistent with **(P1)** and **(P2)**.

*Table 8.* Quantitative comparisons on ImageNet, Caltech101, OxfordPets, StanfordCars, Flowers102 and the average of eleven datasets. The best and the runner-up method on each dataset are marked with blue and red, respectively.

| Method | Average | | | ImageNet | | |
|---|---|---|---|---|---|---|
| | HM | AUROC | OpenworldAUC | HM | AUROC | OpenworldAUC |
| CLIP | 71.55 | 79.62 | 47.83 | 70.18±0.00 | 87.91±0.00 | 47.31±0.00 |
| CoOp | 70.65 | 84.27 | 47.59 | 71.19±0.51 | 95.15±0.22 | 48.93±0.71 |
| Maple | 77.82 | 86.18 | 57.35 | 73.31±0.23 | 95.47±0.52 | 51.89±0.47 |
| PromptSRC | 78.83 | 86.20 | 58.21 | 73.59±0.36 | 95.83±0.08 | 52.44±0.53 |
| LoCoOp | 71.44 | 89.87 | 52.62 | 68.67±0.45 | 92.96±0.31 | 45.12±0.60 |
| KgCoOp | 76.31 | 84.23 | 55.07 | 73.00±0.08 | 95.33±0.16 | 51.45±0.13 |
| DePT-kg | 78.37 | 87.37 | 59.00 | 73.13±0.14 | 93.38±0.62 | 51.35±0.21 |
| Gallop | 77.75 | 85.81 | 56.90 | 70.83±0.14 | 95.93±0.15 | 49.01±0.18 |
| DeCoOp | 76.10 | 90.68 | 58.77 | 73.15±0.64 | 96.22±0.09 | 51.98±0.11 |
| TCP | 78.56 | 86.05 | 58.90 | 72.90±0.06 | 95.31±0.11 | 51.34±0.14 |
| **Ours** | 78.24 | 91.08 | 60.94 | 73.19±0.15 | 96.94±0.06 | 52.64±0.16 |

| Method | Caltech101 | | | OxfordPets | | |
|---|---|---|---|---|---|---|
| | HM | AUROC | OpenworldAUC | HM | AUROC | OpenworldAUC |
| CLIP | 95.67±0.00 | 89.44±0.00 | 82.31±0.00 | 93.36±0.00 | 85.09±0.00 | 76.17±0.00 |
| CoOp | 94.10±0.51 | 93.48±0.65 | 83.29±0.65 | 93.87±1.30 | 90.56±0.48 | 80.71±2.31 |
| Maple | 96.38±0.47 | 93.66±0.36 | 87.15±0.99 | 96.02±0.49 | 92.23±0.94 | 85.59±1.62 |
| PromptSRC | 96.71±0.28 | 92.60±0.16 | 86.74±0.35 | 96.43±0.17 | 91.83±0.45 | 86.10±0.42 |
| LoCoOp | 93.28±1.44 | 98.88±0.39 | 86.59±2.65 | 93.86±2.71 | 97.73±0.90 | 86.62±5.52 |
| KgCoOp | 96.37±0.19 | 92.98±0.29 | 86.63±0.34 | 96.42±0.26 | 92.47±0.46 | 86.80±0.43 |
| DePT-kg | 96.59±0.08 | 99.39±0.07 | 92.74±0.17 | 95.90±0.24 | 95.23±0.35 | 87.81±0.35 |
| Gallop | 95.76±0.38 | 95.25±0.36 | 87.51±0.73 | 96.38±0.36 | 93.00±0.22 | 86.94±0.60 |
| DeCoOp | 96.45±0.07 | 99.48±0.07 | 92.72±0.10 | 94.83±0.65 | 98.21±0.42 | 88.72±1.21 |
| TCP | 96.50±0.12 | 92.34±2.85 | 88.65±2.65 | 95.74±0.44 | 92.27±0.27 | 85.50±0.83 |
| **Ours** | 96.51±0.06 | 99.49±0.04 | 92.81±0.12 | 95.52±0.13 | 98.03±0.92 | 89.77±0.97 |

| Method | StanfordCars | | | Flowers102 | | |
|---|---|---|---|---|---|---|
| | HM | AUROC | OpenworldAUC | HM | AUROC | OpenworldAUC |
| CLIP | 69.02±0.00 | 87.91±0.00 | 43.43±0.00 | 73.75±0.00 | 85.56±0.00 | 48.51±0.00 |
| CoOp | 62.54±2.56 | 84.97±1.37 | 35.38±2.85 | 78.86±3.50 | 92.21±0.97 | 59.65±4.40 |
| Maple | 73.69±0.34 | 89.84±0.55 | 49.99±0.50 | 83.38±0.45 | 92.17±0.52 | 65.44±0.33 |
| PromptSRC | 74.52±0.65 | 89.15±0.18 | 50.90±0.85 | 85.55±0.58 | 93.38±0.20 | 69.36±0.92 |
| LoCoOp | 68.66±1.52 | 94.66±0.99 | 46.52±1.86 | 77.50±1.45 | 96.86±0.47 | 61.17±2.07 |
| KgCoOp | 73.82±0.05 | 88.36±0.31 | 49.40±0.09 | 82.63±0.93 | 90.67±1.41 | 62.96±2.28 |
| DePT-kg | 76.88±0.46 | 93.57±0.85 | 55.24±0.44 | 85.10±0.40 | 92.83±0.75 | 69.46±1.02 |
| Gallop | 73.91±0.69 | 89.57±0.56 | 50.69±0.73 | 82.51±0.33 | 94.50±0.65 | 65.69±0.30 |
| DeCoOp | 74.03±0.29 | 95.42±0.53 | 53.59±0.47 | 84.15±0.45 | 96.84±0.36 | 70.28±0.61 |
| TCP | 76.25±0.34 | 88.85±0.20 | 53.18±0.31 | 84.34±1.97 | 94.05±0.26 | 69.20±0.35 |
| **Ours** | 75.32±0.17 | 95.58±0.27 | 55.31±0.32 | 85.76±0.12 | 96.95±0.89 | 72.79±0.60 |

*Table 9.* Quantitative comparisons on Food101, Fgvc-aircraft, SUN397, DTD, EuroSAT and UCF101. The best and the runner-up method on each dataset are marked with blue and red, respectively.

| Method | Food101 | | | FGVC_Aircraft | | |
|---|---|---|---|---|---|---|
| | HM | AUROC | OpenworldAUC | HM | AUROC | OpenworldAUC |
| CLIP | 90.3±0.00 | 89.57±0.00 | 75.09±0.00 | 30.94±0.00 | 60.89±0.00 | 7.23±0.00 |
| CoOp | 85.26±1.29 | 89.12±0.43 | 67.27±1.81 | 26.20±2.83 | 54.89±1.35 | 5.60±0.70 |
| Maple | 90.72±0.25 | 90.77±0.42 | 76.83±0.58 | 35.17±1.70 | 66.82±3.18 | 9.58±0.46 |
| PromptSRC | 90.89±0.38 | 91.13±0.17 | 77.33±0.57 | 39.77±0.75 | 69.53±2.09 | 11.40±0.61 |
| LoCoOp | 86.41±1.73 | 95.93±0.86 | 73.09±3.09 | 31.02±2.06 | 66.64±1.87 | 8.67±0.91 |
| KgCoOp | 90.84±0.18 | 90.67±0.37 | 76.96±0.24 | 34.08±0.51 | 56.01±1.90 | 8.18±0.22 |
| DePT-kg | 91.06±0.15 | 94.56±0.17 | 79.45±0.32 | 38.54±1.04 | 74.47±0.77 | 12.71±0.57 |
| Gallop | 88.96±0.48 | 90.48±0.24 | 73.60±0.92 | 40.32±1.16 | 58.83±1.36 | 11.38±0.42 |
| DeCoOp | 90.56±0.06 | 97.36±0.24 | 80.67±0.27 | 31.41±0.28 | 69.91±3.14 | 8.17±0.32 |
| TCP | 90.87±0.21 | 91.02±0.27 | 77.27±0.40 | 37.77±1.38 | 59.91±1.17 | 10.72±0.73 |
| **Ours** | 90.79±0.06 | 97.63±0.08 | 81.25±0.12 | 37.78±0.61 | 71.23±2.27 | 11.42±0.42 |

| Method | SUN397 | | | DTD | | |
|---|---|---|---|---|---|---|
| | HM | AUROC | OpenworldAUC | HM | AUROC | OpenworldAUC |
| CLIP | 72.37±0.00 | 77.39±0.00 | 42.52±0.00 | 56.62±0.00 | 64.69±0.00 | 25.22±0.00 |
| CoOp | 75.54±0.62 | 81.63±0.46 | 48.03±0.56 | 53.30±1.91 | 76.03±1.06 | 25.48±1.40 |
| Maple | 79.35±0.28 | 81.53±0.50 | 52.84±0.61 | 67.16±0.76 | 73.53±2.03 | 36.22±1.52 |
| PromptSRC | 79.84±0.31 | 82.60±0.56 | 54.19±0.67 | 69.60±0.82 | 77.19±1.23 | 40.30±0.94 |
| LoCoOp | 73.24±0.76 | 89.35±0.28 | 50.55±0.98 | 57.73±3.43 | 81.14±2.01 | 32.26±3.12 |
| KgCoOp | 78.62±0.39 | 81.68±0.41 | 52.09±0.72 | 65.53±3.26 | 74.42±0.64 | 34.87±3.07 |
| DePT-kg | 79.79±0.42 | 85.72±0.26 | 56.42±0.65 | 67.51±0.89 | 72.19±0.99 | 37.56±1.05 |
| Gallop | 77.26±0.49 | 82.56±0.11 | 50.62±0.70 | 70.14±1.57 | 76.85±1.71 | 40.22±0.79 |
| DeCoOp | 78.08±0.11 | 89.97±0.45 | 57.00±0.14 | 63.94±1.14 | 79.64±2.19 | 37.07±1.61 |
| TCP | 80.13±0.18 | 83.03±0.16 | 54.86±0.36 | 66.36±0.65 | 78.47±0.55 | 37.92±0.85 |
| **Ours** | 79.04±0.10 | 90.70±0.48 | 58.54±0.21 | 68.32±0.87 | 78.58±1.67 | 40.37±1.28 |

| Method | EuroSAT | | | UCF101 | | |
|---|---|---|---|---|---|---|
| | HM | AUROC | OpenworldAUC | HM | AUROC | OpenworldAUC |
| CLIP | 60.21±0.00 | 64.08±0.00 | 28.01±0.00 | 74.66±0.00 | 83.29±0.00 | 50.37±0.00 |
| CoOp | 68.07±3.93 | 82.87±3.88 | 41.96±4.94 | 68.18±2.65 | 86.02±0.41 | 43.95±2.94 |
| Maple | 80.15±5.84 | 86.04±5.21 | 56.55±10.18 | 80.71±1.18 | 85.89±0.75 | 58.72±1.80 |
| PromptSRC | 79.10±5.18 | 79.46±4.31 | 52.56±9.26 | 81.16±0.85 | 85.46±0.60 | 58.94±1.36 |
| LoCoOp | 66.59±7.49 | 82.60±4.34 | 41.35±5.27 | 68.88±1.41 | 91.83±0.77 | 46.90±1.29 |
| KgCoOp | 68.85±6.59 | 77.22±2.90 | 39.16±7.31 | 79.21±1.23 | 86.67±0.84 | 57.29±1.48 |
| DePT-kg | 76.84±3.54 | 68.33±2.69 | 44.90±1.96 | 80.74±1.02 | 91.41±0.46 | 61.38±1.37 |
| Gallop | 80.06±5.90 | 78.28±3.13 | 51.38±8.33 | 79.13±1.38 | 88.64±0.52 | 58.91±1.81 |
| DeCoOp | 72.25±2.76 | 80.70±3.44 | 46.66±1.81 | 78.25±0.40 | 93.73±0.78 | 59.57±0.72 |
| TCP | 80.13±4.17 | 83.29±2.60 | 55.89±5.14 | 83.14±0.37 | 87.99±0.36 | 63.39±0.51 |
| **Ours** | 77.66±0.22 | 83.72±2.68 | 53.09±1.36 | 80.76±0.32 | 93.07±1.15 | 62.39±0.56 |

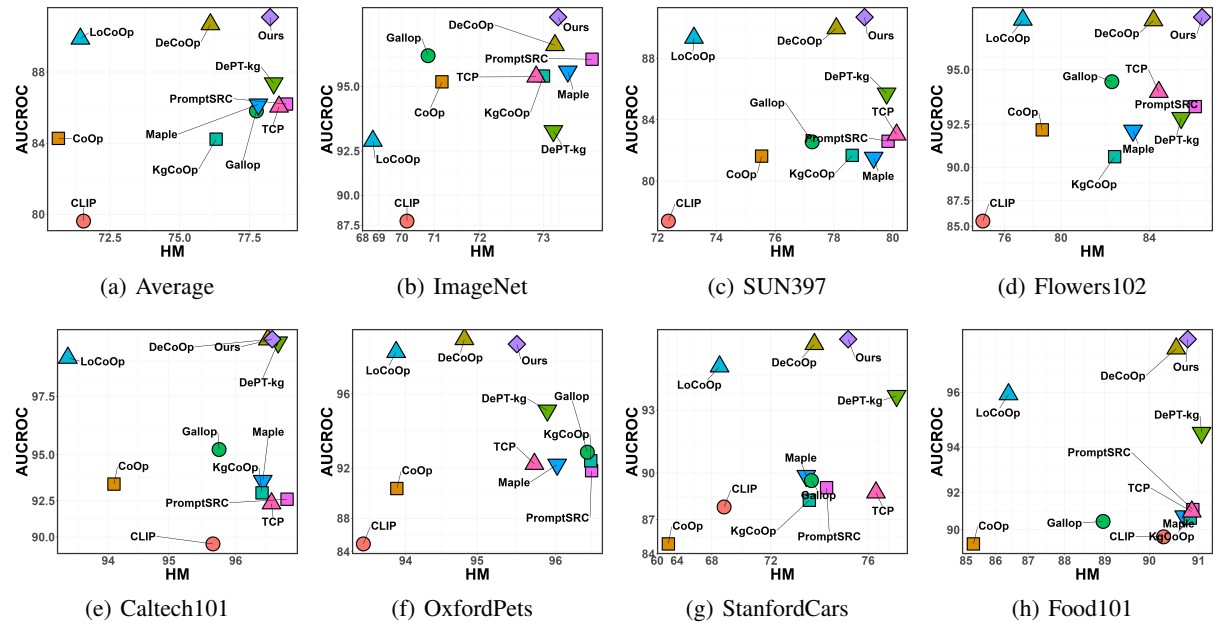

*Figure 7.* Trade-off between the first-stage AUROC metric and the second-stage HM metric on the Openworld Recognition Task.

### D.2. Additional Results for Openworld Domain Generalization Task

In this section, we present comprehensive results with standard deviation and also provide stage-wise metrics. The Tab.10 and Fig.8 once again illustrate that optimizing our can more effectively balance the trade-off between the first-stage detection performance **(P1)** and the second-stage classification performance **(P2)**. This further underscores the robustness and generalizability of across diverse domains, as well as the efficacy of our proposed approach in addressing the challenges of open-world domain generalization tasks.

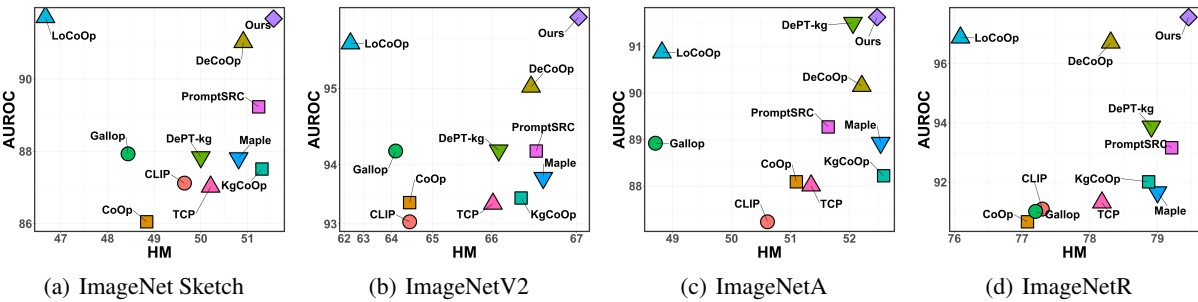

*Figure 8.* Trade-off between the first-stage AUROC metric and the second-stage HM metric on the Openworld domain generalization task.

### D.3. Sensitivity Analysis of different metrics in varying domain distributions

Tab.11 and Tab.12 show the OverallAcc can be dominated by the performance in the domain with more samples. To be specific, the varying new/base ratio is constructed by adjusting sample sizes: when the ratio exceeds 1 (i.e., more new samples), the number of new samples is held constant while reducing base samples; conversely, when the ratio is below 1, base samples remain unchanged and new samples are decreased. Under this setup, when the new/base ratio is larger than 1 meaning that there are more new samples, the OverallAcc metric is lower which is dominated by new domain. As the number of base domain samples grows, OverallAcc increases significantly as the number of base domain samples grows, while our metric OpenworldAUC and AUROC, HM remain stable. Since the prior knowledge about domain distribution is generally unknown, a distribution-sensitive metric is improper for OPT.

*Table 10.* Quantitative comparisons on ImageNetv2, ImageNet-sketch, ImageNet-A and ImageNet-R. The best and the runner-up method on each dataset are marked with blue and red, respectively.

| Method | ImageNetv2 | | | ImageNet_sketch | | |
|--------|------|-------|-------------|------|-------|-------------|
| | HM | AUROC | OpenworldAUC | HM | AUROC | OpenworldAUC |
| CLIP | 64.49±0.00 | 93.04±0.00 | 39.49±0.00 | 49.65±0.00 | 87.12±0.00 | 22.57±0.00 |
| CoOp | 64.49±0.34 | 93.39±0.32 | 39.88±0.40 | 48.84±0.40 | 86.05±0.81 | 21.69±0.40 |
| Maple | 66.65±0.15 | 93.80±0.64 | 42.44±0.23 | 50.81±0.43 | 87.81±0.71 | 23.79±0.37 |
| PromptSRC | 66.57±0.48 | 94.20±0.16 | 42.55±0.58 | 51.24±0.54 | 89.23±0.49 | 24.50±0.50 |
| LoCoOp | 62.37±0.78 | 95.49±0.16 | 38.45±0.90 | 46.67±0.50 | 91.71±0.20 | 20.88±0.47 |
| KgCoOp | 66.39±0.24 | 93.47±0.14 | 42.14±0.32 | 51.31±0.17 | 87.51±0.31 | 24.24±0.13 |
| DePT-kg | 66.10±0.21 | 94.21±0.83 | 42.45±0.39 | 50.00±0.41 | 87.84±1.68 | 23.97±0.19 |
| Gallop | 64.12±0.20 | 94.20±0.23 | 39.86±0.28 | 48.44±0.37 | 87.93±0.48 | 21.57±0.35 |
| DeCoOp | 66.51±0.21 | 95.02±0.11 | 43.01±0.22 | 50.91±0.16 | 91.02±0.20 | 24.65±0.12 |
| TCP | 66.02±0.15 | 93.37±0.08 | 41.66±0.20 | 50.21±0.30 | 87.02±0.25 | 23.15±0.31 |
| **Ours** | **67.02±0.11** | **95.75±0.10** | **43.98±0.19** | **51.56±0.14** | **91.67±0.21** | **25.64±0.16** |

| Method | ImageNet_A | | | ImageNet_R | | |
|--------|------|-------|-------------|------|-------|-------------|
| | HM | AUROC | OpenworldAUC | HM | AUROC | OpenworldAUC |
| CLIP | 50.61±0.00 | 87.23±0.00 | 23.83±0.00 | 77.30±0.00 | 91.09±0.00 | 57.04±0.00 |
| CoOp | 51.10±0.45 | 88.09±0.77 | 24.22±0.43 | 77.08±0.68 | 90.66±0.86 | 56.58±1.08 |
| Maple | 52.53±1.21 | 88.93±0.42 | 25.10±0.57 | 79.00±0.38 | 91.66±0.56 | 59.71±0.68 |
| PromptSRC | 51.64±0.95 | 89.27±0.70 | 25.08±0.91 | 79.21±0.59 | 93.16±0.40 | 60.65±0.93 |
| LoCoOp | 48.82±0.59 | 90.87±0.19 | 22.98±0.56 | 76.09±0.82 | 96.88±0.23 | 57.31±1.28 |
| KgCoOp | 52.58±0.50 | 88.22±0.36 | 25.79±0.49 | 78.87±0.08 | 92.01±0.18 | 59.75±0.10 |
| DePT-kg | 52.06±0.21 | 91.51±0.44 | 25.84±0.40 | 78.91±0.12 | 93.88±0.14 | 60.60±0.12 |
| Gallop | 48.71±0.53 | 88.92±0.36 | 22.28±0.47 | 77.20±0.33 | 91.01±0.37 | 56.78±0.61 |
| DeCoOp | 52.21±0.33 | 90.15±0.23 | 25.31±0.22 | 78.31±0.22 | 96.71±0.18 | 61.01±0.11 |
| TCP | 51.35±0.37 | 88.01±0.24 | 24.62±0.35 | 78.18±0.17 | 91.31±0.27 | 58.46±0.31 |
| **Ours** | **52.47±0.56** | **91.63±0.35** | **26.49±0.63** | **79.46±0.16** | **97.57±0.20** | **62.67±0.27** |

## D.4. Additional Results for Openworld Cross-dataset Generalization Task

We extend our evaluation to investigate the generalization performance of our optimization framework in more challenging cross-dataset open-world scenarios. The comprehensive results of this cross-domain evaluation, which rigorously tests the model's ability to handle both known and new categories across diverse visual domains, are presented in the Tab.13. The experimental results further speak to the efficiency of our method.

## D.5. Fine-grained analysis of OpenworldAUC metric

The OpenworldAUC comprehensively evaluates: 1) base-to-new detection 2) base classification 3) new classification. A high OpenworldAUC indicates the model performs well on three. To diagnose low OpenworldAUC, we can further check three sub-metrics: AUROC, BaseAcc, NewAcc. To validate this, we present fine-grained results on four datasets shown in the table below and Tab.14, Fig.9, which reveal:

- OOD-focused methods (Gallop) excel at BaseAcc-AUROC but struggle with new domain classification

- Base-to-new methods (DePT) show weaker detection performance

- Our method achieves better OpenworldAUC, indicating improved trade-offs across three, which further validates the comprehensiveness of OpenworldAUC.

## D.6. Complexity analysis

We have included the prompt complexity analysis in Fig.5 in the main text. To highlight the efficiency of our method, we present those numerical results in the Tab.15, along with additional results on inference speed in the Tab.16. The Tab.15 compares average performance on three open-world tasks and learnable parameter counts across methods. Our method

*Table 11.* Performance of TCP(Yao et al., 2024) on DTD dataset with respect to new/base ratios. It's clear that the Overall metric is sensitive to the new/base ratio.

| The new/base ratio | BaseAcc | NewAcc | HM | AUROC | OverallAcc | OpenworldAUC |
|:---:|:---:|:---:|:---:|:---:|:---:|:---:|
| 10 | 83.06 | 55.80 | 66.75 | 80.42 | 40.94 | 38.62 |
| 5 | 81.43 | 55.80 | 66.22 | 78.61 | 43.72 | 37.75 |
| 3 | 81.09 | 55.80 | 66.11 | 78.84 | 45.45 | 37.85 |
| 2 | 82.31 | 55.80 | 66.51 | 78.62 | 49.76 | 38.24 |
| 1 | 81.85 | 55.80 | 66.36 | 78.47 | 54.61 | 37.92 |
| 0.7 | 81.85 | 55.72 | 66.30 | 78.03 | 57.40 | 37.62 |
| 0.5 | 81.85 | 55.46 | 66.12 | 78.47 | 59.77 | 37.86 |
| 0.3 | 81.85 | 53.04 | 64.37 | 78.90 | 63.33 | 36.21 |
| 0.2 | 81.85 | 56.02 | 66.52 | 78.68 | 64.94 | 38.18 |
| 0.1 | 81.85 | 52.46 | 63.94 | 77.69 | 67.74 | 35.43 |
| Mean | 81.90 | 55.17 | 65.92 | 78.67 | 54.77 | 37.57 |
| Variance | 0.27 | 1.66 | 0.92 | 0.51 | **89.13** | 0.96 |

*Table 12.* Performance of TCP(Yao et al., 2024) on Flowers102 dataset with respect to new/base ratios. It's clear that the Overall metric is sensitive to the new/base ratio.

| The new/base ratio | BaseAcc | NewAcc | HM | AUROC | OverallAcc | OpenworldAUC |
|:---:|:---:|:---:|:---:|:---:|:---:|:---:|
| 10 | 96.15 | 75.76 | 84.75 | 92.84 | 72.77 | 67.78 |
| 5 | 97.47 | 75.76 | 85.25 | 94.00 | 74.25 | 69.34 |
| 3 | 96.17 | 75.76 | 84.75 | 93.53 | 75.18 | 68.39 |
| 2 | 97.67 | 75.76 | 85.33 | 94.40 | 77.59 | 69.67 |
| 1 | 97.19 | 75.76 | 85.15 | 94.05 | 80.97 | 69.19 |
| 0.7 | 97.19 | 75.86 | 85.21 | 94.09 | 83.16 | 69.34 |
| 0.5 | 97.19 | 75.47 | 84.96 | 94.17 | 84.81 | 69.01 |
| 0.3 | 97.19 | 76.07 | 85.34 | 94.67 | 87.71 | 69.97 |
| 0.2 | 97.19 | 76.26 | 85.46 | 94.39 | 89.31 | 70.08 |
| 0.1 | 97.19 | 75.52 | 85.00 | 94.78 | 91.33 | 69.59 |
| Mean | 97.06 | 75.80 | 85.12 | 94.09 | 81.71 | 69.24 |
| Var. | 0.25 | 0.05 | 0.06 | 0.32 | **43.75** | 0.50 |

outperforms SOTA methods on the average performance of these with **a smaller parameter cost**. While recent SOTA methods design **deep** prompt structures, we optimize **multiple shallow** prompts in detector and classifiers. As shown in Tab.16, while slightly slower than CoOp due to prompt mixing, our approach runs 34% faster than DeCoOp and matches DePT/Gallop in speed, which maintains practical inference speeds.

*Table 13.* The OpenworldAUC empirical results on seven benchmark for open-world cross-dataset generalization.

| Method | AC | C101 | Cars | DTD | ES | SUN | UCF | Avg |
|---|---|---|---|---|---|---|---|---|
| CLIP | 16.21 | 60.76 | 45.59 | 31.13 | 30.33 | 44.31 | 45.83 | 39.17 |
| CoOp | 12.06 | 62.50 | 44.67 | 26.75 | 25.70 | 44.72 | 45.38 | 37.40 |
| MaPLe | 16.36 | 66.29 | 47.34 | 31.89 | 31.56 | 48.72 | 48.74 | 41.56 |
| PromptSRC | 17.75 | 65.89 | 49.08 | 33.94 | 34.14 | 49.53 | **50.06** | 42.91 |
| KgCoOp | 16.15 | 64.84 | 47.75 | 31.46 | 33.39 | 48.34 | 49.40 | 41.62 |
| DePT-Kg | 17.47 | 66.54 | 49.98 | 33.96 | 35.17 | 49.38 | 49.62 | 43.16 |
| DeCoOp | 16.95 | 66.35 | 50.04 | 33.91 | 36.12 | 49.42 | 49.65 | 43.21 |
| TCP | 16.77 | 65.86 | 47.60 | 32.69 | 33.50 | 48.81 | 49.31 | 42.08 |
| Ours | **18.18** | **66.58** | **50.25** | **34.21** | **36.84** | **49.65** | 49.87 | **43.65** |

*Table 14.* Fine-grained results in terms of BaseAcc, NewAcc, HM, AUROC and OpenworldAUC metrics on Flowers102, ImageNet, ImageNet V2 and ImageNet-Sketch, comparing OOD-focused method Gallop, base-to-new method DePT, OPT method DeCoOp.

| Method | Flowers102 | | | | | ImageNet | | | | |
|---|---|---|---|---|---|---|---|---|---|---|
| | BaseAcc | NewAcc | HM | AUROC | OpenworldAUC | BaseAcc | NewAcc | HM | AUROC | OpenworldAUC |
| DePT-Kg | 97.68 | 75.38 | 85.09 | 92.83 | 69.46 | 76.75 | 69.85 | 73.14 | 93.38 | 51.35 |
| Gallop | **98.60** | 70.94 | 82.51 | 94.50 | 65.69 | **79.25** | 64.02 | 70.83 | 95.93 | 49.01 |
| DeCoOp | 93.90 | 76.24 | 84.15 | 96.84 | 70.28 | 75.55 | **70.90** | 73.15 | 96.22 | 51.98 |
| **Ours** | 96.53 | **77.16** | **85.76** | **96.95** | **72.79** | 76.13 | 70.46 | **73.19** | **96.94** | **52.64** |

| Method | ImageNet V2 | | | | | ImageNet-Sketch | | | | |
|---|---|---|---|---|---|---|---|---|---|---|
| | BaseAcc | NewAcc | HM | AUROC | OpenworldAUC | BaseAcc | NewAcc | HM | AUROC | OpenworldAUC |
| DePT-Kg | 70.25 | 62.42 | 66.10 | 90.21 | 41.45 | 46.64 | 53.88 | 50.00 | 86.84 | 23.37 |
| Gallop | **73.20** | 57.04 | 64.12 | 94.20 | 39.86 | **49.79** | 47.16 | 48.44 | 87.93 | 21.57 |
| DeCoOp | 70.55 | 62.91 | 66.51 | 95.02 | 43.01 | 48.23 | 53.90 | 50.91 | 91.02 | 24.65 |
| **Ours** | 71.04 | **63.60** | **67.02** | **95.75** | **43.98** | 48.77 | **54.69** | **51.56** | **91.67** | **25.64** |

## D.7. Effect of mutiple pseudo base-to-new Partitions.

Fig.10 illustrates the effect of using different numbers of base-to-new partitions ($K$) on the Flowers102 and SUN397 datasets. The results show that OpenworldAUC increases monotonically as more class partitions are performed. This observation agrees with Thm. 5.2. To purse the tradeoff between the efficiency and the efficacy, we choose $K = 3$ in our main experiments.

## D.8. Effect of Different Shots

Fig.11 and Fig.6 illustrate the performance comparison between different shot settings (1-shot, 2-shot, 4-shot, 8-shot and 16-shot) between the proposed method and representative competitors on the DTD and OxfordPets datasets. The results show that our method consistently outperforms existing representative approaches across all shot settings, further demonstrating its effectiveness. Additionally, OpenworldAUC improves as $N$ increases, aligning with our theoretical findings.

## D.9. Ablation Studies of the Gating Mechanism

We further explore the effectiveness of the gating mechanism in the Tab.17. Replacing the sigmoid-weighted gate with a fixed 0-1 mask ("Ours 0-1 Gate") slightly improves over removing the gate entirely ("Ours w/o Gate") but under-performs compared to the adaptive sigmoid gate. It validates the effectiveness of sparse sample selection mechanism and the gate approximation mechanism.

## D.10. Effect of $CE$ Regularization

In the practical optimization objective ($OP_{fin}$), the second term is actually an AUC-like ranking loss. However, recent AUC-based optimization studies (Yuan et al., 2022; Yang et al., 2023) show that maximizing the AUC loss from scratch can degrade feature representations. Therefore, we add a cross-entropy regularization term ($\lambda \ell_{ce}$) to improve discriminability

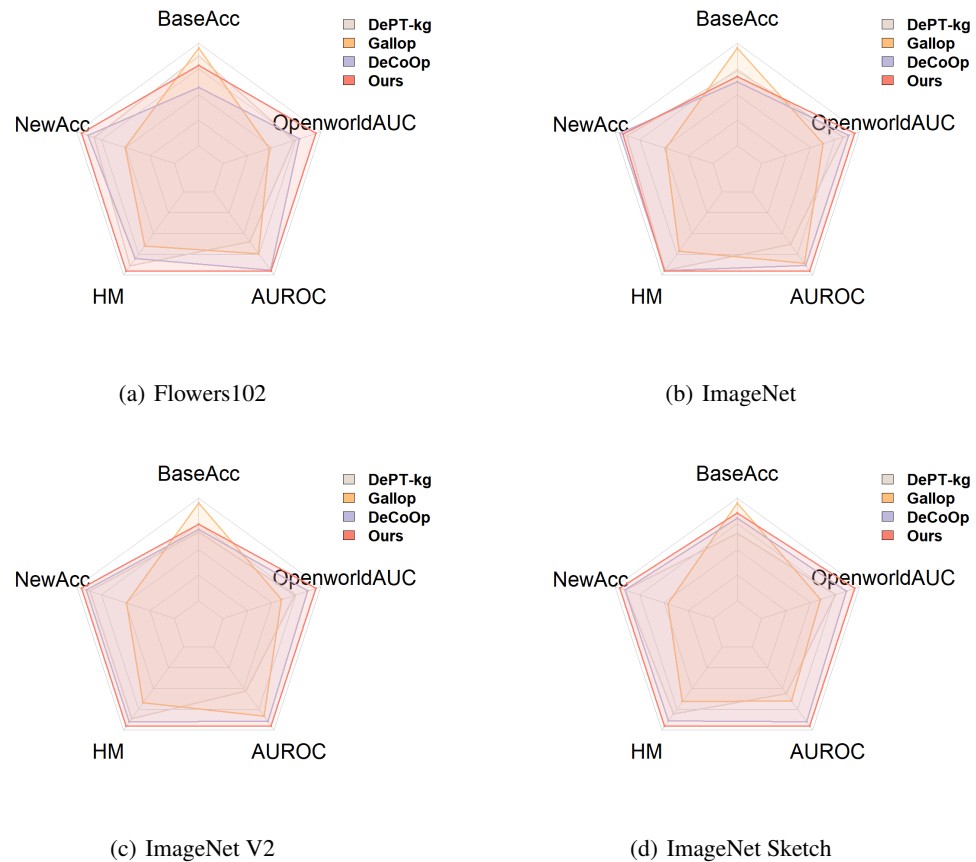

(a) Flowers102

(b) ImageNet

(c) ImageNet V2

(d) ImageNet Sketch

*Figure 9.* Fine-grained results on four datasets, Flowers102, ImageNet, ImageNetV2, ImageNet Sketch, comparing three competitive methods and ours. Our method achieves better OpenworldAUC, indicating improved trade-offs across three sub-metrics, which further validates the comprehensiveness of OpenworldAUC.

following (Yang et al., 2023). To assess the impact of $\lambda$, we carry out sensitivity tests by altering $\lambda$'s value. We present the results for Caltech101 and Food101 in the Fig.12 and Fig.6. A proper $\lambda \in [1/2, 1]$ can effectively improve the ranking performance.

### D.11. Ablation Studies of Mixture-of-Prompts

In order to show the effectiveness of our proposed mixture-of-prompts, we compare its performance with the following two variants of MoP:

- **w/o Zero Shot new domain classifier** $h$ abbreviated as w/o ZS h. This variant of our method replaces the handcrafted prompt for the new domain classifier with a tuned prompt based on the base domain.

- **Single Prompt**. This is a variant of our method where we only use one prompt to balance three component in $\hat{\mathcal{R}}(r, g, h)$.

The empircal results on ImageNet, StanfordCars, DTD, Flowers102, SUN397 and UCF101 datasets are provided in Fig.6 and Fig.13. From these results, we can see that: 1) Traing a new domain classifier on the base domain can lead to overfitting, compromising generalization on unseen new domains. Therefore, **w/o ZS** $h$ could not outperform our mixture-of-prompts. 2) Single prompt performs significantly worse than mixture prompts. This validates the challenge of optimizing the OpenworldAUC within a single prompt. This strengthens the effectiveness of our proposed mixture-of-prompts scheme.

*Table 15.* The comparisons of average performance on open-world tasks and learnable parameter counts across methods.

| Method | Recognition task | Imbalanced recognition task | Domain adaption | **#param** |
|---|---|---|---|---|
| CLIP | 47.83 | 54.59 | 47.31 | 0 |
| CoOp | 47.59 | 53.97 | 48.93 | 8.2k |
| Maple | 57.35 | 63.05 | 51.89 | 3555.1K |
| PromptSRC | 58.21 | 65.44 | 52.44 | 46.1k |
| LoCoOp | 52.62 | 60.65 | 45.12 | 8.2k |
| KgCoOp | 55.07 | 62.71 | 51.45 | 8.2k |
| DePT-Kg | 59.00 | 66.06 | 51.35 | 292.9k |
| Gallop | 56.90 | 63.67 | 49.01 | 606.2k |
| DeCoOp | 58.77 | 66.87 | 51.98 | 30.7K |
| TCP | 58.90 | 65.13 | 51.34 | 331.9k |
| **Ours** | **60.94** | **68.84** | **52.64** | 26.6k |

*Table 16.* Average per-sample inference time comparison across ten datasets (ImageNet excluded) in the open-world recognition task.

| CoOp | DePT | Gallop | DeCoOp | Ours |
|---|---|---|---|---|
| 0.00117S | 0.00167S | 0.00148S | 0.00273S | 0.00180S |

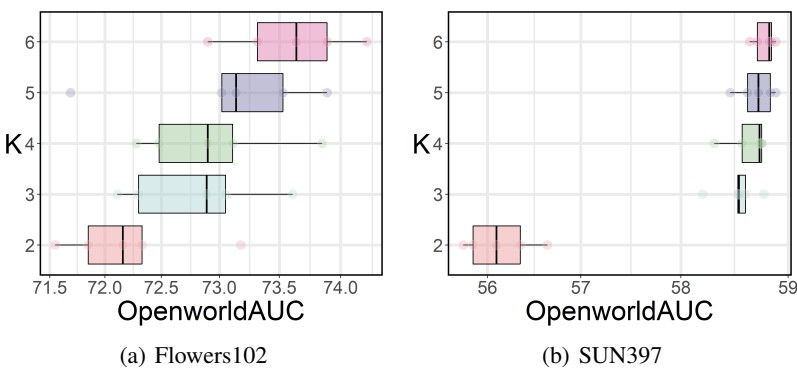

(a) Flowers102  (b) SUN397

*Figure 10.* The Effect of Using Different Number of base-to-new Partitions ($K$).

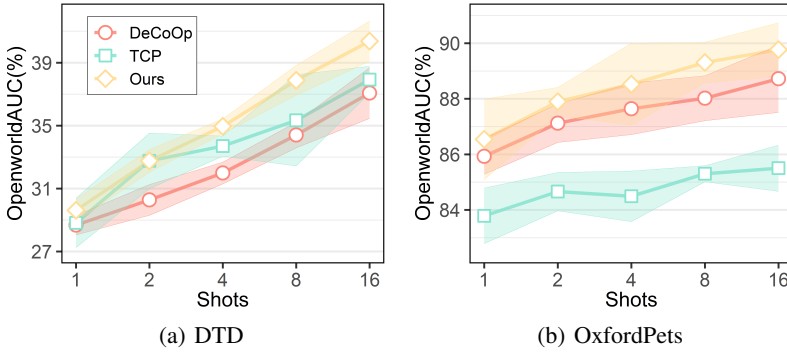

(a) DTD  (b) OxfordPets

*Figure 11.* Comparison of few-shot learning with $1, 2, 4, 8, 16$-shot samples on DTD and OxfordPets.

*Table 17.* The ablation study of the gating mechanism.

| Method | Avg. | IN | C101 | Pets | Cars | F102 | Food | AC | SUN | DTD | ES | UCF |
|---|---|---|---|---|---|---|---|---|---|---|---|---|
| Ours w/o Gate | 60.29 | 52.49 | 92.56 | 89.47 | 55.06 | 72.59 | 79.30 | 10.97 | 56.96 | 40.63 | 51.27 | 61.85 |
| Ours 0-1 Gate | 60.65 | 52.61 | 92.77 | 89.50 | 55.20 | 72.71 | 79.92 | 11.08 | 57.13 | **40.72** | 52.78 | 62.75 |
| Ours Sigmoid Gate | **60.94** | **52.64** | **92.81** | **89.77** | **55.31** | **72.79** | **81.25** | **11.42** | **58.54** | 40.37 | **53.09** | **62.39** |

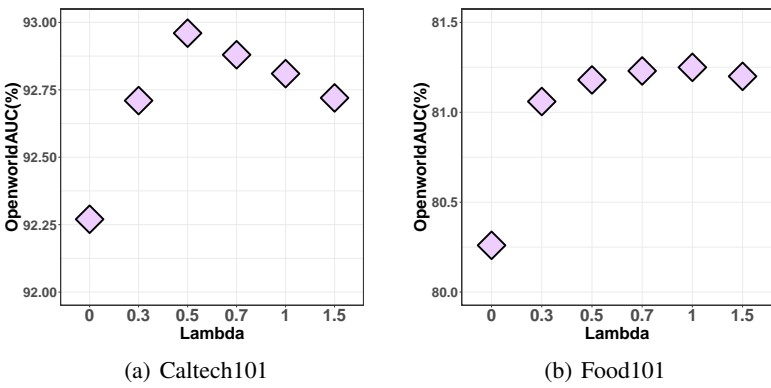

(a) Caltech101         (b) Food101

*Figure 12.* The Sensitive Analysis of $\lambda$.

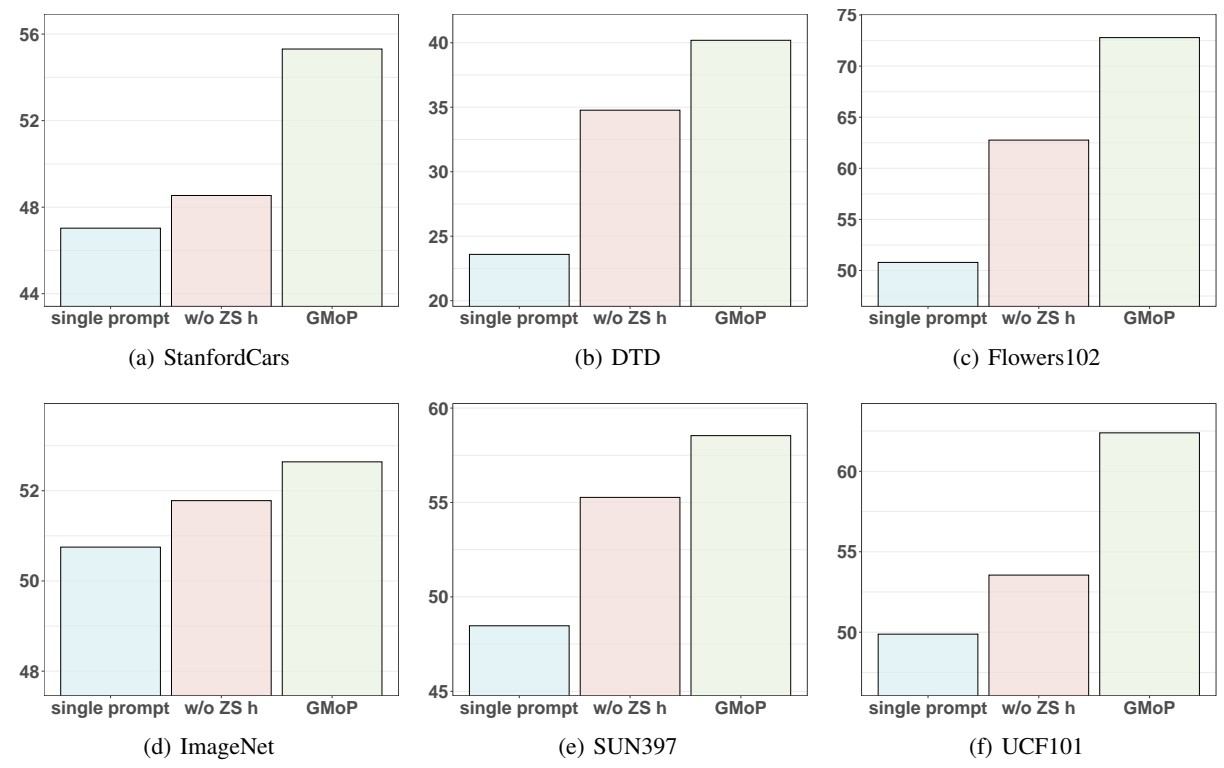

(a) StanfordCars     (b) DTD     (c) Flowers102

(d) ImageNet     (e) SUN397     (f) UCF101

*Figure 13.* The ablation study of mixture-of-prompts.

