# OpenReview forum: "OpenworldAUC: Towards Unified Evaluation and Optimization for Open-world Prompt Tuning"
_ICML.cc/2025/Conference — ICML 2025 poster_

### Official Review · Reviewer_uVwC · 2025-02-27

**Overall Recommendation:** 4

**Summary:**

The paper introduces OpenworldAUC, a unified evaluation metric for open-world prompt
tuning (OPT) that jointly assesses base-to-new detection (P1), domain-specific classification
(P2), and insensitivity to domain distribution (P3). To optimize OpenworldAUC, the authors
propose Gated Mixture-of-Prompts (GMoP), which employs domain-specific prompts and a
gating mechanism to balance detection and classification. Theoretical guarantees for
generalization are provided, and experiments on 15 benchmarks.

**Claims And Evidence:**

The authors provide a detailed analysis of the limitations of existing metrics, such as HM,
OverallAcc, and AUROC, and demonstrate how OpenworldAUC overcomes these limitations.
The effectiveness of the proposed GMoP framework is supported by extensive experimental
results on multiple benchmarks.

**Essential References Not Discussed:**

None

**Experimental Designs Or Analyses:**

The experimental designs and analyses are well-structured and valid. The authors evaluate
the proposed method on a diverse set of benchmarks, including open-world recognition and
domain generalization tasks. The experiments are designed to test the method's
performance under various conditions, such as different domain distributions and imbalance
settings.

**Methods And Evaluation Criteria:**

Methods: OpenworldAUC: The pairwise formulation (jointly ranking detection and
classification correctness) is novel and aligns well with OPT requirements. The use of
domain-specific prompts and gating mechanisms is logical for balancing conflicting subobjectives. The pseudo partition strategy and zero-shot classifier for new domains address
practical constraints (unseen classes).
Evaluation Criteria: Benchmarks cover diverse scenarios (recognition, domain generalization,
imbalance), and metrics include OpenworldAUC, HM, AUROC, and OverallAcc. However,
computational efficiency is not discussed.

**Other Comments Or Suggestions:**

The implementation details section could benefit from more specific information about the
training strategy and hyperparameters used for each competitor.

**Other Strengths And Weaknesses:**

Strengths:
The paper is well-structured and clearly presents the problem, the proposed solution, and
the experimental results.
Weaknesses:
While the paper discusses the potential impact on fairness-sensitive real scenarios, it lacks
specific examples or case studies demonstrating the application of the proposed method in
real-world settings.

**Questions For Authors:**

How does the pseudo partition strategy generalize to truly unseen domains (e.g., crossdataset evaluation)?

While the paper discusses the potential impact on fairness-sensitive real-world scenarios, it lacks specific examples or case studies demonstrating the application of the proposed method in real-world settings. Could the authors provide concrete examples or case studies that illustrate how the proposed method can be applied in such scenarios?

The Gated Mixture-of-Prompts (GMoP) framework introduces multiple prompts and a gating mechanism. Is this approach practical in terms of computational cost and implementation complexity?

How does the performance gain from GMoP compare to the increased complexity?

Given that the framework may be too complex for real-world applications, especially in scenarios with limited computational resources, could the authors provide a detailed analysis of the trade-offs between performance and complexity?

**Relation To Broader Scientific Literature:**

None

**Theoretical Claims:**

The paper presents theoretical guarantees for the generalization of the GMoP framework.
While the proofs are not provided in the main text, the authors reference the appendix for
detailed proofs.

---

> ### Author Rebuttal · Authors · 2025-04-01
>
> We deeply appreciate your time and effort in providing us with such constructive comments. We would like to respond to them as follows:
>
> > Q1: Generalization to truly unseen domains, e.g., cross-dataset evaluation.
>
> Following your insightful suggestion, we extend our evaluation to investigate the generalization performance of our optimization framework in more challenging cross-dataset open-world scenarios. The experimental settings are listed as follows:
>
> - We train our prompt exclusively on the ImageNet-500 base domain and subsequently evaluate model performance on a combined test set containing both the original ImageNet-500 classes and new categories from seven external datasets: `FGVC-Aircraft`, `Caltech-101`, `Stanford-Cars`, `DTD`, `EuroSAT`,  `SUN397`, and `UCF101`.
> - To ensure a fair evaluation of open-world generalization, we meticulously remove any categories from these external datasets that overlapped with the ImageNet-500 class space before evaluation.
>
> The comprehensive results of this cross-domain evaluation, which rigorously tests the model's ability to handle both known and new categories across diverse visual domains, are presented in table below. The experimental results further speak to the efficiency of our method.
>
> ||AC|C101|Cars|DTD|ES|SUN|UCF|Avg|
> |-|-|-|-|-|-|-|-|-|
> |CLIP|16.21|60.76|45.59|31.13|30.33|44.31|45.83|39.17|
> |CoOp|12.06|62.50|44.67|26.75|25.70|44.72|45.38|37.40|
> |MaPLe|16.36|66.29|47.34|31.89|31.56|48.72|48.74|41.56|
> |PromptSRC|17.75|65.89|49.08|33.94|34.14|49.53|**50.06**|42.91|
> |KgCoOp|16.15|64.84|47.75|31.46|33.39|48.34|49.40|41.62|
> |DePT-Kg|17.47|66.54|49.98|33.96|35.17|49.38|49.62|43.16|
> |DeCoOp|16.95|66.35|50.04|33.91|36.12|49.42|49.65|43.21|
> |TCP|16.77|65.86|47.60|32.69|33.50|48.81|49.31|42.08|
> |Ours|**18.18**|**66.58**|**50.25**|**34.21**|**36.84**|**49.65**|49.87|**43.65**|
>
> > Q2: The trade-offs between **performance** and **complexity** of mixture-of-prompts.
>
> Thank you for your constructive suggestion! In fact, we have included the prompt complexity analysis in Fig.5 in the initial submission. To highlight the efficiency of our method, we present those numerical results in the table below, along with additional results on inference speed.
>
> - The table compares average performance on three open-world tasks and learnable parameter counts (#param) across methods
>
> |Method|Recognition task | Imbalance recognition|Domain adaption | #param |
> |-|-|-|-|-|
> |CLIP|47.83|54.59|47.31|0|
> |CoOp|47.59|53.97|48.93|8.2k|
> |Maple|57.35|63.05|51.89|3555.1K|
> |PromptSRC|58.21|65.44|52.44|46.1k|
> |DePT|59.00|66.06|51.35|292.9k|
> |Gallop|56.90|63.67|49.01|606.2k|
> |DeCoOp|58.77|66.87|51.98|30.7K|
> |TCP|58.90|65.13|51.34|331.9k|
> |Ours|**60.94**|**68.84**|**52.64**|26.6k|
>
> - The table presents a comparison of per-sample inference time, averaged across ten datasets
>
> |CoOp|DePT|Gallop|DeCoOp|Ours|
> |-|-|-|-|-|
> |0.00117S|0.00167S|0.00148S|0.00273S|0.00180S|
>
> According to these empirical results, we answer the questions raised by the reviewer:
>
> - Our method outperforms SOTA methods on the average performance of three open-world tasks with **a smaller parameter cost**. While recent SOTA methods design **deep** prompt structures, we optimize **multiple shallow** prompts in the detector and classifiers.
>
> - While slightly slower than CoOp due to prompt mixing, our approach runs 34% faster than DeCoOp and matches DePT/Gallop in speed, which maintains practical inference speeds.
> - **The performance gain outweighs complexity**. Our method outperforms SOTA methods by **1.94% –13.55%** across tasks while using **≤ 9% parameters** of methods like Maple or DePT. The moderate speed trade-off (slightly slower than CoOp) is justified by significant performance improvements.
>
> > Q3: Fairness-sensitive real-world scenarios application of the proposed method
>
> The **cross-dataset task** mentioned above involves a significant **imbalance between the base and new domains**, as shown in the table below. The base domain includes 500 ImageNet categories with 25,000 testing samples.
>
> |New domain|#categories|class-imbalance|#samples|sample-imbalance|
> |-|-|-|-|-|
> |StanfordCars|196|2.6|8041|3.1|
> |UCF101|101|5.0|3783|6.6|
> |Caltech101|84|6.0|2135|11.7|
> |DTD|47|10.6|1692|14.8|
> |EuroSAT|10|50.0|8100|3.1|
> |Sun397|392|1.3|19600|1.3|
> |FGVC_aircraft|100|5.0|3333|7.5|
>
> Accuracy-driven metrics often favor the majority domain of ImageNet, creating fairness risks in scenarios with extreme data imbalances (e.g., 15× sample or 50× class ratios). These imbalances mirror real-world cases where critical minority samples are suppressed by dominant base classes. OpenworldAUC overcomes this bias by fairly evaluating all classes through pairwise ranking. Unlike traditional metrics that amplify imbalance-related errors, OpenworldAUC improves reliability in critical applications.

---

> > ### Comment · Reviewer_uVwC · 2025-04-05
> >
> > The authors have addressed my concern. I will raise my score.

---

> > > ### Author Response · Authors · 2025-04-07
> > >
> > > Thank you so much for your feedback and the improved score. Following your suggestions, we will enrich the content of our paper in the final version.

---

### Official Review · Reviewer_Df2f · 2025-03-08

**Overall Recommendation:** 4

**Summary:**

This paper explore a new evaluation metric openworldauc for the practical open-world prompt tuning (OPT) task, which jointly measure the inter-domain detection and intra-domain classification performance and remains insensitive towards the varying data distributions. Further, the mixture-of-prompt learning framework GMoP is proposed to optimize openworldauc during training. Besides, generalization analysis are conduced to support the method theoretically. Comprehensive experiments on various benchmark show the  advantages of the proposed metric and the effectiveness of the proposed framework.

**Claims And Evidence:**

The submission provides well-substantiated claims supported by both theoretical and empirical evidence.

- The paper provides sufficient arguments to analyze the limitations of existing metrics and demonstrates the superior properties of the proposed OpenworldAUC metric from both theoretical and empirical perspectives.
- The paper also demonstrates the effectiveness of the proposed GMoP method through a novel generalization bound and comprehensive empirical studies.

**Essential References Not Discussed:**

The paper covers most of the essential related works in the field.

**Experimental Designs Or Analyses:**

I review the experimental setup, results, and analysis in both the main text and appendix. In my view, the experimental setup in this paper is well-justified and the results are convincing. Notably, the authors first incorporate the additional experiments on distribution imbalance compared to previous research.  The experiments further validate the comprehensiveness of OpenworldAUC and its insensitivity to imbalanced data distributions. The lightweight learning framework outperforms competitors in OpenworldAUC and achieves a superior balance between the detection metric AUC and the classification metric HM, confirming its effectiveness.

However, some details regarding the calculation of the detection score  $r(\cdot)$  in the main text could be improved. The authors mention in the appendix that the calculation of  $r(\cdot)$ can utilize new class names in this experimental setting for all competitors. I recommend that the authors emphasize this implementation detail in the main text.

**Methods And Evaluation Criteria:**

The proposed evaluation metric and the proposed learning framework make sense for the open-world learning at hand. To be specific,

- Theoretical results show that the proposed openworldauc is consistent with the goal of OPT while escape from all identified limitations. In light of this, this metric can evaluate models more robustly and guide model optimization more effectively, thereby providing valuable insights to the open-world learning community and advancing the field.

- To purist of this, the proposed mixture-of-prompt framework  co-optimizes the prompts for the detector and classifier using a sparse gating mechanism. This innovative approach offers new insights and design principles for optimizing models in open-world scenarios.

**Other Comments Or Suggestions:**

Please see the weakness part above.

**Other Strengths And Weaknesses:**

The strengths of this paper are as follows:

- This paper provides a systematic analysis of the limitations of existing metrics for the OPT task. The proposed OpenworldAUC metric, which features a concise formulation, effectively addresses these limitations and demonstrates desirable properties from both theoretical and empirical perspectives.
- The paper introduces a novel and unified learning framework designed to optimize the OpenworldAUC metric. This framework dynamically balances multiple prompts targeting specific goals through a sparse gating mechanism. To support this approach, the authors derive a novel and informative bound, a contribution rarely explored in prior literature.
- The empirical results are convincing. They highlight the limitations of the existing accuracy metric and underscore the advantages of OpenworldAUC. Additionally, the results validate the effectiveness of the proposed learning framework.

My minor concerns are as follows:

- The authors should provide a more detailed explanation of the challenges associated with generalization analysis in the main text.
- The authors should elaborate further on the calculation of the base-domain confidence score \( $r$​ \).
- The authors should supplement the ablation studies by investigating the performance of the gating mechanism. Specifically, it would be valuable to examine how the model performs when the gating mechanism is removed and replaced with a simple binary 0-1 mask for selecting correctly classified samples.

Overall, I hold a clear positive view of the proposed novel metric and the corresponding empirical learning framework. I believe this paper makes significant contributions to the open-world learning and prompt-tuning community and meets the standards of ICML.

**Questions For Authors:**

Please see the weakness part above.

**Relation To Broader Scientific Literature:**

Towards the practical and challenging Openworld prompt tuning (OPT) task, prior methods adopt the HM metric , AUROC metric and the Overall-accuracy metric to evaluate the model, here, the authors argue that these metrics suffer from three types of limitations. To this end, this paper proposed the novel metric openworldauc and the corresponding empirical learning framework, which are the key contributions to the openworld learning and prompt tuning communities.

**Theoretical Claims:**

I generally review the correctness of the proofs for several theoretical claims within the manuscript, which include:

- The proofs related to metric analysis, corresponding to Proposition 4.1, Proposition 4.2, and Proposition 4.3.
- The proof for Proposition 5.1.
- The proof related to the generalization bound.

Overall, the theoretical analysis is both correct and intuitively sound. In particular, the proof of the generalization bound demonstrates ingenuity, as the authors decompose the complex generalization gap into several parts and analyze the source of each error term, resulting in an informative and illustrative generalization bound.

In the context of the generalization bound for detector optimization, the authors use covering number and ϵ*ϵ*-net arguments to derive the generalization bound. My minor concern is the challenge of this analysis. The authors should provide more explanation for why traditional Rademacher Complexity-based theoretical analysis cannot be directly applied here.

---

> ### Author Rebuttal · Authors · 2025-03-31
>
> We deeply appreciate your time and effort in providing us with such constructive comments. We would like to respond to them as follows:
>
> > **Q1:** More detailed explanation of the challenges associated with generalization analysis.
>
> - Our main theoretical findings focus on how well our optimization method generalizes to **unseen** test data distributions with **new classes**. According to standard learning theory, we measure generalization error by comparing: (a) the expected error across the joint distribution of the real new domain $\mathcal{Y}$ and test data $\mathcal{D}$, and (b) the empirical average across the pseudo new domain $\hat{\mathcal{Y}}$ and training data $\mathcal{S}$. To construct the pseudo new domain, we perform $K$ pseudo base-to-new partitions $\hat{\mathcal{Y}}^{(k)}, k \in [1, K]$, which incorporates a hierarchical sampling approach, class sampling, and data sampling. This leads to a **hierarchical** structure of stochastic errors, presenting a significant challenge in our theoretical analysis.
> - The another major challenge arises from the AUC-type risk $\ell_{sq}(r(x_b;\theta_r) - r(x_n;\theta_r))$ in optimizing the detector $r$. The standard generalization analysis techniques, such as Rademacher complexity-based theoretical arguments[1,2], require the loss function to be expressed as the sum of independent terms. Unfortunately, the pairwise AUC-type risk can not satisfy this assumption. For instance, the optimization functions for the detector, $\ell_{sq}(r(x_b^i;\theta_r) - r(x_n^j;\theta_r))$ and $\ell_{sq}(r(\tilde{x}_b^{i};\theta_r) - r(\tilde{x}_n^j;\theta_r))$, are interdependent if any term is shared (e.g., $\tilde{x}_n^j = x_n^j$ or $\tilde{x}_b^i = x_b^i$). To this end, in this study, we use **covering numbers** and **$\epsilon$-net arguments** [3] in the subsequent proof to derive the generalization bound.
>
> > **Q2:** The authors should elaborate further on the calculation of the base-domain confidence score  $r$.
>
> Following the setting in [4], the new domain class names are known during testing in the OPT task and the base-domain confidence score is derived from the maximum probability over the base domain. A high value of $r$ means the high probability of the sample belonging to the base domain. Given the `image_features` [1,512] and `text_features` [512,C] where 512 means  the dimension of the latent feature and $C$ is the number of all classes, we calculate the  `prob = Softmax(image_features·text_features,dim=1)` and then base domain confidence score can be obtained via  $\max_{1\le j\le C_b}\textbf{prob}[:,j]$, where $C_b$ is the number of base classes.
>
> > **Q3:** The ablation studies of  the gating mechanism.
>
> Following your suggestion, we further explore the effectiveness of the gating mechanism. Replacing the sigmoid-weighted gate with a fixed 0-1 mask ("Ours 0-1 Gate") slightly improves over removing the gate entirely ("Ours w/o Gate") but under-performs compared to the adaptive sigmoid gate. It validates the effectiveness of sparse sample selection mechanism and the gate approximation mechanism.
>
> | Method | Avg.  | IN      | C101      | Pets  | Cars  | F102  | Food  | AC    | SUN   | DTD   | ES    | UCF   |
> |-|-|-|-|-|-|-|-|-|-|-|-|-|
> | Ours w/o Gate     | 60.29     | 52.49     | 92.56     | 89.47 | 55.06 | 72.59 | 79.3  | 10.97 | 56.96 | 40.63 | 51.27 | 61.85 |
> | Ours 0-1 Gate     | 60.65     | 52.61     | 92.77     | 89.50 | 55.20 | 72.71 | 79.92 | 11.08 | 57.13 | **40.72** | 52.78 | 62.75 |
> | Ours Sigmoid Gate | **60.94** | **52.64** | **92.81** | **89.77** | **55.31** | **72.79** | **81.25** | **11.42** | **58.54** | 40.37 | **53.09** | **62.39** |
>
> > [1]Rademacher and Gaussian Complexities: Risk bounds and structural results. COLT, vol. 2111, pp. 224–240, 2001.
> >
> > [2]Foundations of Machine Learning. MIT Press, 2012.
> >
> > [3]Probability in Banach Spaces: Isoperimetry and processes. 1991.
> >
> > [4]DECOOP: Robust Prompt Tuning with Out-of-Distribution Detection. ICML 2024.

---

### Official Review · Reviewer_PcZP · 2025-03-13

**Overall Recommendation:** 5

**Summary:**

This paper addresses the open-world prompt tuning problem and uncovers the fundamental limitations of current evaluation metrics in this field. To tackle this challenge, the authors propose a novel, unified metric, OpenWorldAUC, which jointly evaluates the model’s detection and classification performance without requiring prior domain knowledge. On top of this, a Gated Mixture-of-Prompts (GMoPs) approach is introduced to optimize OpenWorldAUC directly. Both theoretical analyses and empirical studies consistently demonstrate the effectiveness of the proposed method.

**Claims And Evidence:**

The claims made in the work are supported by clear and convincing evidence.

**Essential References Not Discussed:**

No. The literature included in this paper is satisfactory.

**Experimental Designs Or Analyses:**

Yes. The experiment designs of this paper follow the standard evaluation setups in this area and the authors compare the proposed method/metric with 10 recent state-of-the-art methods. Overall, the empirical studies are convincing and comprehensive.

**Methods And Evaluation Criteria:**

This study presents a well-motivated and rigorous framework to open-world prompt tuning.

**Other Comments Or Suggestions:**

N/A

**Other Strengths And Weaknesses:**

Novel Open-world metric for reliable evaluations: Existing methods assess model performance separately on base and unseen domains, resulting in inconsistent evaluations. This work provides a systematic analysis to reveal the limitations of existing metrics and proposes a novel metric, OpenworldAUC, to address these shortcomings. Furthermore, an end-to-end algorithm is developed that enables the model to learn directly from OpenworldAUC. This approach establishes a more comprehensive and reliable evaluation framework, thereby facilitating more effective model optimization in open-world scenarios.

Sufficient theoretical guarantee: This paper conducts theoretical analysis of the proposed learning framework, which provides an interesting perspective to understand the work mechanism for OpenworldAUC. The results show that optimizing OpenworldAUC could lead to a satisfactory generalization performance.

Comprehensive experiments: The authors perform extensive experiments to showcase the advantages of the proposed method, including 11 benchmarks and 10 competitors. Additionally, ablation studies are presented to compare the effectiveness of OpenWorldAUC against previous counterpart metrics. Empirical results consistently show that maximizing OpenWorldAUC leads to superior performance in both open-world recognition and open-world domain generalization tasks.


However, I have the following minor concerns:

1. The necessity of introducing the Gated Mixture-of-Prompts remains unclear. Why are the inputs to each component distinct? A more detailed explanation is needed to justify this design choice.

2. How is \( OP_0 \) derived from Proposition 5.1? Additionally, why is it valid to approximate the 0-1 loss using the square loss? I believe the authors should provide a clearer explanation, along with relevant references, to aid readers who may be less familiar with this topic.

3. The motivation behind the zero-shot new domain classifier is not well articulated. Why is \theta_{h}^* necessary, and what is the underlying intuition for its inclusion? A more thorough discussion would help clarify its role in the overall framework.

**Questions For Authors:**

Please refer to the Weaknesses part.

**Relation To Broader Scientific Literature:**

This paper borrows the ideas from open-world recognition, prompt tuning, and mixture-of-experts models, contributing a novel perspective on evaluation and optimization in open-world prompt learning. The proposed method has broad application potential in various AI-driven scenarios.

**Theoretical Claims:**

I have carefully checked the proofs of theoretical arguments including Prop.4.2, Prop.4.3, Prop.5.1 and Thm.5.2, and confirmed their correctness.

---

> ### Author Rebuttal · Authors · 2025-03-31
>
> Thanks for your constructive comments, and we would like to make the following response.
>
> > **Q1:** The necessity of introducing the Gated Mixture-of-Prompts remains unclear. A more detailed explanation is needed to justify this design choice.
>
> The three components $g$, $h$, and $r$ have **conflicting** objectives: $g$ classifies base samples (its objective function relies on base samples), $h$ classifies new samples (its objective function uses new samples), and $r$ ranks base-new pairs (its objective function uses base-new sample pairs). Using a single prompt leads to **mutual interference**. To address this, our design assigns distinct prompts to each component, separating their optimization processes. A gate mechanism, which uses sigmoid-weighted confidence scores, adaptively combines their outputs. This ensures that the $r$-prompt focuses on ranking correctly classified pairs. By doing so, we prevent conflicts and allow each prompt to specialize for its task. Empirical results from six datasets confirm the difficulty of optimizing OpenworldAUC with a single prompt and demonstrate the effectiveness of our mixture-of-prompts strategy.
>
> ||ImageNet|SUN397|DTD|Cars|UCF101|Flowers102|
> |:-:|:-:|:-:|:-:|:-:|:-:|:-:|
> |Single Prompt|50.75|48.47|23.59|47.03|49.88|50.79|
> |GMoP|52.64|58.54|40.19|55.31|62.39|72.79|
>
> > **Q2:** How is ( OP_0 ) derived from Proposition 5.1? Additionally, why is it valid to approximate the 0-1 loss using the square loss? I believe the authors should provide a clearer explanation, along with relevant references, to aid readers who may be less familiar with this topic.
>
> ($OP_0$) is derived by replacing the population risk in Proposition 5.1 with its **empirical approximation** (since $\mathcal{D}$ is unknown). Besides, following the framework of surrogate loss, we replace the non-differential 0-1 loss with a convex loss function $\ell$, such that $\ell(t)$ is an upper bound of $\ell_{0,1}(t)$. Note that if the scores live in $[0,1]$, standard loss functions such as $\ell_{sq}(t) = (1-t)^2$ often satisfy this constraint. This smooth approximation enables gradient-based optimization while preserving ranking semantics. The details can also be found in the recent survey [1,2,3].
>
> > **Q3:** The motivation behind the zero-shot new domain classifier is not well articulated. A more thorough discussion would help clarify its role in the overall framework.
>
> Thank you for your constructive suggestion!  The motivation behind the zero-shot new domain classifier falls into the following two aspects.
>
> **Balance efficiency and accuracy:** Recent prompt-tuning methods show that learnable prompts, optimized on the base domain with only CE loss, may hurt new-domain classification performance compared to zero-shot CLIP with fixed prompts. The results on the NewAcc of the Zeroshot CLIP and the PT baseline CoOp confirm this point, as shown in the table below.
>
> |  | ImageNet | SUN397 |  DTD  | Cars | UCF101 | Oxford_flowers |
> | :-----------: | :------: | :----: | :---: | :----------: | :----: | :------------: |
> | Zeroshot-CLIP | **68.10** | **75.62** | **60.51** |    **75.02** | **78.64** |     **77.19**  |
> |   CoOp    |  67.03   | 71.28  | 40.19 |    56.37     | 52.96  |     66.57      |
>
> To improve generalization, many recent methods focus on **maintaining alignment** (loss design) with zero-shot CLIP while using **structured prompts** (structure design) to preserve zero-shot knowledge. These methods slightly outperform zero-shot CLIP in term of new domain accuracy but introduce additional computational and storage overhead. To purse a tradeoff between the, we just design a hand-crafted prompt.
>
> **Decoupling Classification in different domains:** Since the base-to-new detector effectively distinguishes base and new samples in open-world scenarios, we can **decouple** the learnable prompt $\theta_g$ (for base-domain classification) and the fixed prompt  $\theta_{h}^*$(for new-domain classification) and achieve promising performance. In other words, using different prompt parameters for classification task in different domains is feasible and effective in practical scenarios.
>
> To further validate the effectiveness of zero-shot new domain classifier, we replace $\theta_{h}^*$ with $\theta_g$ and observe the overall OpenworldAUC performance drops, confirming the necessity of this decoupling and the effectiveness of the zero-shot new domain classifier.
>
> ||ImageNet|SUN397|DTD|StanfordCars|UCF101|Oxford_flowers|
> |:-:|:-:|:-:|:-:|:-:|:-:|:-:|
> |Ours w/o ZS h|51.78|55.27|34.77|48.54|53.55|62.76|
> |Ours|**52.64**|**58.54**|**40.19**|**55.31**|**62.39**|**72.79**|
>
> > [1] Auc maximization in the era of big data and ai: A survey. ACMComputing Surveys (CSUR), 2022.
> >
> > [2] On the consistency of AUC pairwise optimization. IJCAI, 2015.
> >
> > [3] Stochastic auc maximization with deep neural networks. ICLR,2019.

---

> > ### Comment · Reviewer_PcZP · 2025-04-04
> >
> > The authors have addressed my concern on Gated Mixture-of-Prompts, 0-1 loss approximation, and new domain classifier. I'll raise my score!

---

> > > ### Author Response · Authors · 2025-04-07
> > >
> > > We are grateful for your comments and the improved rating. We will incorporate all your suggestions to strengthen our paper's content in the revision.

---

### Official Review · Reviewer_fECn · 2025-03-13

**Overall Recommendation:** 2

**Summary:**

Since existing evaluation metrics cannot comprehensively assess performance in open-world prompt tuning, this paper proposes a unified evaluation metric called OpenworldAUC. This metric not only measures the detection capability of base/new samples (P1) and classification accuracy (P2), but also ensures robustness against changes in the proportion of base to new samples. Additionally, a multi-prompt combination method based on a gating mechanism is proposed to optimize the OpenworldAUC metric.

**Claims And Evidence:**

Yes, the claims made in the submission are supported by clear and convincing evidence.

**Essential References Not Discussed:**

No, the paper cites all essential references necessary to understand its key contributions. The authors have adequately discussed prior related findings and provided a comprehensive context for their work.

**Experimental Designs Or Analyses:**

Yes, I checked the soundness and validity of all experimental designs and analyses, and they are appropriate and well-executed.

**Methods And Evaluation Criteria:**

Yes, the proposed methods and/or evaluation criteria are sensible for the problem at hand.

**Other Comments Or Suggestions:**

No.

**Other Strengths And Weaknesses:**

Strengths:
Innovation: This paper proposes a new evaluation metric, OpenworldAUC, to overcome the limitations of existing evaluation methods.
Method effectiveness: The GMoP method introduced in this paper optimizes detection and classification through multi-prompt combinations, achieving outstanding performance on multiple state-of-the-art tasks.
Theoretical support: Through theoretical reasoning, the paper demonstrates that OpenworldAUC offers more stable evaluations and that the GMoP training objective exhibits strong generalization.
Experimental comprehensiveness: The results from testing on 15 datasets and various ablation experiments validate the robustness of this approach.

Weaknesses:
Data distribution sensitivity: Imbalanced sampling of base/new data may affect detection optimization.
Limited generalization ability: Unseen new class data may lead to overfitting on the base class, impacting detector performance.
High optimization difficulty: Manual adjustment of loss weights is required, and the approach is sensitive to hyperparameters.
High computational complexity: Multiple prompts are still needed during inference, making it unsuitable for low-compute devices.

**Questions For Authors:**

Based on the "Weaknesses" section, I have the following concerns:
1. Establishing a fair and objective metric can reflect the true performance of a model and reveal its current limitations. However, can the proposed OpenworldAUC in this paper provide a fine-grained analysis of the model's various capabilities and effectively uncover its limitations?
2. The quality of the prompt determines the model's classification performance. If the prompt is poorly designed, classification performance may degrade. How does the author ensure the quality of the prompts?
3. The GMoP method still requires multiple prompts during the inference stage. Does this introduce additional computational and storage overhead?
4. GMoP requires manual tuning of multiple loss function weights. How does the author ensure the optimal weights across different datasets?
5. Since the "new" class is unknown during training, the paper simulates this through a pseudo base-to-new partitioning approach. How does the author ensure the accuracy of this partitioning method?

**Relation To Broader Scientific Literature:**

The key contributions of this paper are closely related to the broader scientific literature on evaluation metrics for open-world recognition tasks. Existing metrics, such as HM, Overall Accuracy, and AUROC, have been widely used but suffer from specific limitations. These metrics fail to simultaneously address three critical requirements: (P1) distinguishing between base and new classes, (P2) ensuring correct classification, and (P3) adapting to varying class distributions. Building on these prior findings, the authors propose a novel evaluation metric, OPEB, which aims to address these limitations and provide a more comprehensive framework for evaluating open-world recognition systems.

**Theoretical Claims:**

Yes, I checked the correctness of all proofs for the theoretical claims, and they are correct.

---

> ### Author Rebuttal · Authors · 2025-03-31
>
> Thanks for your valuable comments! Due to space constraints, we include full tables and figures https://anonymous.4open.science/r/R-4D06. References to Tab.X, Fig.X  correspond to those provided in this link.
>
> > Q1: OpenworldAUC for fine-grained model capability analysis
>
> The OpenworldAUC comprehensively evaluates: 1) base-to-new detection 2) base classification 3) new classification. A high OpenworldAUC indicates the model performs well on three. To diagnose low OpenworldAUC, we can further check three sub-metrics: AUROC, BaseAcc, NewAcc. To validate this, we present fine-grained results on four datasets shown in the table below and Tab.1, Fig.1, which reveal:
> - OOD-focused methods (Gallop) excel at BaseAcc-AUROC but struggle with new domain classification
> - Base-to-new methods (DePT) show weaker detection performance
> - Our method achieves better OpenworldAUC, indicating improved trade-offs across three, which further validates the comprehensiveness of OpenworldAUC.
> |F102|Baseacc|Newacc|AUROC|OpenworldAUC|
> |-|-|-|-|-|
> |DePT|97.68|75.38|92.83|69.46|
> |Gallop|**98.60**|70.94|94.50|65.69|
> |DeCoOp|93.90|76.24|96.84|70.28|
> |Ours|96.53|**77.16**|**96.95**|**72.79**|
>
> > Q2: How to ensure the quality of prompts
>
> The prompts of base classification and base-to-new detection can be **automatically optimized** via our carefully designed loss function, which is a paradigm widely validated in existing prompt tuning research. Additionally, the prompt of new classification is hand-crafted to alleviate overfitting on the base training set. Unlike the naive prompt template "a photo of {}", in our initial submission, we choose a more informative prompt template based on the base domain, further ensuring the new-domain classification performance, shown in the table below and Tab.2.
> |Prompt for ImageNet|NewAcc|OpenworldAUC|
> |-|-|-|
> |a photo of {}|68.12|51.43|
> |Prompt ensemble:a {} in a video game.art of the {}.a photo of the small {}...|70.46|52.64|
>
> > Q3:  Computational complexity of GMoP
>
> In fact, we have included the prompt complexity analysis in Fig.5 in the initial version. To highlight the efficiency of our method, we present those numerical results in the table below and Tab.3, along with additional results on inference speed.
>
> - Our method outperforms SOTA methods on the average performance of three open-world tasks with a smaller parameter cost. While recent SOTA methods design **deep** prompt structures, we optimize **multiple shallow** prompts in detector and classifiers.
>
> ||Recognition task|Domain adaption|#param(k)|
> |-|-|-|-|
> |CoOp|47.59|48.93|8.2|
> |Gallop|56.90|49.01|606.2|
> |Maple|57.35|51.89|3555.1|
> |DeCoOp|58.77|51.98|30.7|
> |DePT|59.00|51.35|292.9|
> |Ours|**60.94**|**52.64**|26.6|
>
> - We measure inference speed by comparing average processing times per sample across ten datasets, testing representative methods and ours. While slightly slower than CoOp due to prompt mixing, our approach runs 34% faster than DeCoOp and matches DePT/Gallop in speed(S)
> |CoOp|DePT|Gallop|DeCoOp|Ours|
> |-|-|-|-|-|
> |0.00117|0.00167|0.00148|0.00273|0.00180|
>
> >Q4: The choice of multiple loss function weights across different datasets
>
> It's a pity that our method is not fully understood. In fact, our optimization framework involves only one loss weight λ of the CE regularization added to the AUC loss, which has been discussed in App.E.6 in the initial version. Sensitivity tests across four datasets show consistent performance when λ∈[1/2,1], with SOTA results in this range, shown in the table below and Tab.4
> ||C101|SUN|
> |-|-|-|
> |DePT|92.74|56.42|
> |DeCoOp|92.72|57.00|
> |λ=1/4|92.64|57.59|
> |λ=1/2|92.96|58.92|
> |λ=3/4|92.88|58.86|
> |λ=1|92.81|58.54|
>
> > Q5: The generalization of pseudo base-new partition
>
> The foundation model itself has strong base-to-new generalization ability. Our task is to enhance such ability by leveraging base training data. To this end,  we adopt the following partition strategy to simulate new class detection.
>
> - We perform **multiple** (K) base-to-new partitions to ensure statistical stability of the new class simulation
>
> - We ensure K pseudo base classes **fully cover** the base class
> - Thm.5.2 suggests increasing K reduces such approximation error. Experiments on two datasets confirm this shown in the table below and Tab.5. We usually set K=3 to balance performance and efficiency
> |SUN397|AUROC|OpenworldAUC|
> |-|-|-|
> |K=1|84.45|53.16|
> |K=2|87.57|56.14|
> |K=3|90.70|58.54|
> |K=4|91.02|58.63|
> |K=5|91.25|58.71|
>
> > Q6: Imbalanced sampling of base/new data may affect detection optimization
>
> Test Imbalance: Since true new-class data is unavailable during training, test imbalance doesn't impact optimization
>
> Pseudo Imbalance: We sample pseudo base/new pairs from true base for AUC loss training. AUC's distribution insensitivity ensures robustness to pseudo base-to-new imbalance, validated by stable performance with varying ratios on DTD
> |b/n ratio|AUROC|OpenworldAUC|
> |-|-|-|
> |2:1|78.58|40.37|
> |3:1|78.30|40.32|
> |5:1|78.45|40.39|

---

### Decision · Program_Chairs · 2025-05-01

**Decision:**

Accept (poster)

**Comment:**

A new unified evaluation metric, OpenWorldAUC, is proposed for open-world prompt tuning, which combines detection ability, classification accuracy and robustness to varying proportions of base and new samples. The paper received three detailed reviews and an author rebuttal was submitted. The ability of the proposed metric to yield fine-grained insights is questioned by fECn, which the author rebuttal suggests can be addressed by consideration of other traditional metrics in conjunction with the proposed one, which is a reasonable approach. PcZP is strongly supportive of acceptance based on evaluation of technical contributions, theoretical justifications and comprehensive experimentation. Clarification questions on the motivations for design choices like the gated mixture of prompts and the zero-shot domain classifier are well-addressed by the rebuttal. Df2f also recommends acceptance based on the method and analysis, with the further ablation study of the gating mechanism satisfactorily provided by the rebuttal. In balance, the ACs agree with the reviewer majority that the paper brings new insights and the proposed new metric has been sufficiently justified with theoretical and empirical analysis to indicate potential broad usage in future works. So, the paper is recommended for acceptance at ICML. The authors are encouraged to include the further clarifications and experiments from the rebuttal in the final version of the paper.